# Single-Agent Poisoning Attacks Suffice to Ruin Multi-Agent Learning

**Fan Yao** [*]
Department of Computer Science
University of Virginia
Charlottesville, VA 22904, USA
fy4bc@virginia.edu

**Yuwei Cheng** [*]
Department of Statistics
University of Chicago
Chicago, IL 60637, USA
yuweicheng@uchicago.edu

**Ermin Wei**
Electrical and Computer Engineering
Industrial Engineering and Management Sciences
Northwestern University
ermin.wei@northwestern.edu

**Haifeng Xu**
Department of Computer Science
University of Chicago
haifengxu@uchicago.edu

## Abstract

We investigate the robustness of multi-agent learning in strongly monotone games with bandit feedback. While previous research has developed learning algorithms that achieve last-iterate convergence to the unique Nash equilibrium (NE) at a polynomial rate, we demonstrate that all such algorithms are vulnerable to adversaries capable of poisoning even a single agent's utility observations. Specifically, we propose an attacking strategy such that for any given time horizon $T$, the adversary can mislead any multi-agent learning algorithm to converge to a point other than the unique NE with a corruption budget that grows sublinearly in $T$. To further understand the inherent robustness of these algorithms, we characterize the fundamental trade-off between convergence speed and the maximum tolerable total utility corruptions for two example algorithms, including the state-of-the-art one. Our theoretical and empirical results reveal an intrinsic efficiency-robustness trade-off: the faster an algorithm converges, the more vulnerable it becomes to utility poisoning attacks. To the best of our knowledge, this is the first work to identify and characterize such a trade-off in the context of multi-agent learning.

## 1 Introduction

In recent years, multi-agent learning (MAL) systems have become increasingly prevalent, finding applications in diverse fields such as autonomous systems, distributed optimization, and economic markets (Leo et al., 2014; Shalev-Shwartz et al., 2016; Jin et al., 2018; Qiu et al., 2021; Zhou et al., 2021; Wu et al., 2022). These systems, characterized by agents acting independently to maximize their own utilities in a non-cooperative environment, offer significant potential but also introduce unique vulnerabilities to potential adversarial attacks. Consider, for example, autonomous vehicles that communicate to coordinate traffic flow or financial trading algorithms that interact to optimize individual profits (Sharif & Marijan, 2022; Ataiefard & Hemmati, 2023; Shah et al., 2024). In both cases, each agent (vehicle or trader) aims to maximize its own utility without direct collaboration. However, these systems can be highly susceptible to manipulation, as a single compromised or malicious agent can significantly influence the overall outcome. While it may be difficult for an adversary to directly attack or control multiple agents, targeting a single vulnerable or insider agent can be more feasible (Lin et al., 2020). This raises an intriguing question of whether an adversary can still create a disruptive outcome by simply poisoning the utility of one agent.

---

[*]Equal contribution.

This scenario—where a single agent in an MAL system is targeted for a utility poisoning attack—presents a new and critical challenge. Prior research has primarily focused on two related but distinct areas: adversarial attacks on individual agents in online learning environments (Bogunovic et al., 2021a; Jun et al., 2018; Garcelon et al., 2020), and many-agent attacks on MAL systems (McMahan et al., 2024; Guo et al., 2021; Wu et al., 2023; Ma et al., 2021; Zhang et al., 2023; Liu & Lai, 2023). However, the impact of single-agent attacks specifically in the multi-agent learning context remains underexplored. In this work, we address this gap by investigating the robustness of MAL dynamics against single-agent utility poisoning attacks. We show that even when one agent is compromised, the system can be disrupted, revealing a stark vulnerability that has been overlooked in prior studies.

To study the effect of single-agent poisoning in multi-agent systems, we use NE-finding dynamics in monotone games as our benchmark. Monotone games (Rosen, 1965), which include well-known models like Cournot competition (Cournot, 1838), Kelly auctions (Kelly et al., 1998), and Tullock contests (Tullock, 2008), represent a broad class of multi-agent strategic systems. In these games, agents' utility functions are concave, and many well-established MAL dynamics converge to their unique NE (Even-Dar et al., 2009; Farina et al., 2022; Bravo et al., 2018; Ba et al., 2024; Tatarenko & Kamgarpour, 2020; Cai & Zheng, 2023), making them ideal benchmarks for studying the effects of single-agent poisoning on system-wide behavior.

Our theoretical results demonstrate that single-agent poisoning attacks can drive the dynamics away from the original NE, using an imperceptible, sublinear budget with respect to the time horizon. This holds for all MAL algorithms in this setting. We also investigate the robustness of two representatives of such MAL algorithms (Ba et al., 2024; Bravo et al., 2018) and show that by sacrificing convergence speed, one can increase the system's tolerance to corruption, revealing a fundamental trade-off between efficiency and robustness.

Our contributions can be summarized as follows: (1) We propose a utility poisoning strategy that can mislead any MAL dynamics in strongly monotone games to converge to a new NE with a sublinear corruption budget, even when only targeting a single agent. (2) We analyze the robustness of MAL algorithms against general utility poisoning attacks, showing that adjusting the learning rate introduces inherent robustness. (3) Our findings uncover an efficiency-robustness trade-off in MAL dynamics, where faster-converging dynamics are more vulnerable to utility poisoning. To the best of our knowledge, this is the first work to reveal this trade-off in multi-agent learning systems.

**Related Work.** Adversarial attacks on multi-agent learning—often termed "steering" or "policy teaching"—have been extensively studied under various assumptions about the attacker's knowledge, objectives, and strategies. For instance, Liu & Lai (2023) show that in certain Markov games, neither action poisoning nor reward poisoning alone can be both efficient and successful, even with complete knowledge of the environment ("white-box" setting), highlighting challenges in designing effective attacks. Similarly, Wu et al. (2023) investigate reward poisoning attacks on offline multi-agent reinforcement learners, demonstrating how adversaries can manipulate outcomes in offline settings.

Another line of research focuses on designing incentives to guide multi-agent dynamics toward a desirable equilibrium, known as "equilibrium steering" (Zhang et al., 2023; Canyakmaz et al., 2024; Huang et al., 2024). Ma et al. (2021) consider game redesign where a designer, operating in a white-box setting, aims to induce players to choose targeted actions in a normal-form game using a sublinear cost, assuming the players employ no-regret learning. The most relevant work to ours is Zhang et al. (2023), which shows that a sublinear total incentive can induce a predetermined equilibrium against no-regret agents. However, their approach can only steer the game to an existing equilibrium and cannot achieve a non-equilibrium outcome without incurring a cost of $\Omega(T)$.

## 2 PRELIMINARIES

Our study focuses on strategic games with a finite number of agents $[n] = \{1, \cdots, n\}$ and continuous action sets $\{\mathcal{X}_i\}_{i=1}^n$ associated with agents $i \in [n]$. During play, each agent $i$ chooses an action (i.e., a pure strategy) $\boldsymbol{x}_i \in \mathcal{X}_i \subset \mathbb{R}^{d_i}$, where $d_i$ is the dimension of action space of agent $i$, and forms the joint action profile $\boldsymbol{x} = (\boldsymbol{x}_i; \boldsymbol{x}_{-i}) \equiv (\boldsymbol{x}_1, \cdots, \boldsymbol{x}_n)$. We

denote $d := \max_{i \in [n]} d_i$. Let $\mathcal{X} = \prod_{i=1}^{n} \mathcal{X}_i$ be the game's strategy space, and each agent's reward is determined by a utility function $u_i : \mathcal{X} \to \mathbb{R}$. Without loss of generality we assume the range of each $u_i$ is a bounded region $[0, 1]$. We do not require agents know their utility functions and only assumes that they can observe the point-wise feedback $u_i(\boldsymbol{x})$, also known as *bandit* feedback in online learning literature. Such a game is denoted by a tuple $\mathcal{G} \equiv \mathcal{G}(n, \mathcal{X}, \{u_i\}_{i=1}^{n})$. The commonly adopted solution concept for such strategic games is Nash equilibrium (NE), an action profile where no agent can deviate unilaterally to improve her utility. The formal definition of NE is given by the following:

**Definition 1.** *A joint strategy profile $\boldsymbol{x}^* = (\boldsymbol{x}_1^*, \cdots, \boldsymbol{x}_n^*)$ forms a Nash equilibrium (NE) of a game $\mathcal{G}$, if for every agent $i$, $\boldsymbol{x}_i^*$ is a best response to the opponents' strategies $\boldsymbol{x}_{-i}^*$; formally,*

$$u_i(\boldsymbol{x}_i^*, \boldsymbol{x}_{-i}^*) \geq u_i(\boldsymbol{x}_i, \boldsymbol{x}_{-i}^*) \quad \text{for every } \boldsymbol{x}_i \in \mathcal{X}_i. \tag{1}$$

## 2.1 Monotone Games

Much of this paper's theoretical development focuses on an important class of games called *strictly monotone games* (strictly MG). This focus is due to at least two reasons. First, from computational perspective, strictly monotone games are one of the most general classes of continuous games that admit efficient algorithms for NE; it includes most popular continuous games such as Cournot competition (Cournot, 1838), Tullock contest (Tullock, 2008; He et al., 2024) and its variants that model content creation competitions (Yao et al., 2024a;b;c). Additionally, our framework encompasses all strictly convex-concave zero-sum games and any game with a strictly concave potential. Second, strictly monotone games turn out to admit fast converging (to NE) multi-agent learning algorithms (Bravo et al., 2018). Both conditions are important for our theoretical analysis. The concept of "monotonicity", first introduced by Rosen (1965), refers to games where each agent has a concave utility function, along with a global condition known as diagonal strict concavity (DSC). A stronger version of the DSC condition defines a sub-class known as strongly monotone games. Formal definitions of these are provided below.

**Definition 2** (Strictly/strongly monotone games). *A continuous multi-agent game $\mathcal{G}(n, \{\mathcal{X}_i\}_{i=1}^{n}, \{u_i\}_{i=1}^{n})$ is strictly monotone if for any $\boldsymbol{x} = (\boldsymbol{x}_1, \cdots, \boldsymbol{x}_n), \boldsymbol{x}' = (\boldsymbol{x}_1', \cdots, \boldsymbol{x}_n') \in \mathcal{X}$, it satisfies the diagonal strict concavity (DSC) property:*

$$\sum_{i=1}^{n} (\boldsymbol{x}_i' - \boldsymbol{x}_i)^\top (v_i(\boldsymbol{x}_i', \boldsymbol{x}_{-i}') - v_i(\boldsymbol{x}_i, \boldsymbol{x}_{-i})) < 0, \tag{2}$$

*where $v_i(\boldsymbol{x}) = \nabla_{\boldsymbol{x}_i} u_i(\boldsymbol{x})$ is the first-order derivative of $u_i$ with respect to $x_i$. In addition, the game $\mathcal{G}$ is $\beta$-strongly monotone if $\sum_{i=1}^{n} (\boldsymbol{x}_i' - \boldsymbol{x}_i)^\top (v_i(\boldsymbol{x}_i', \boldsymbol{x}_{-i}') - v_i(\boldsymbol{x}_i, \boldsymbol{x}_{-i})) \leq -\beta \|\boldsymbol{x}' - \boldsymbol{x}\|_2^2$.*

When discussing strongly monotone game in this paper, we treat $\beta$ as a small constant and sometimes omit it in bounds unless there is a need to emphasize the dependence on $\beta$. Notably, the DSC property readily implies the strict concavity of $u_k$ with respect to $\boldsymbol{x}_k$ for any fixed $\boldsymbol{x}_{-k}$. This can be observed by taking $\boldsymbol{x}' = (\boldsymbol{x}_1, \cdots, \boldsymbol{x}_{k-1}, \boldsymbol{x}_k', \boldsymbol{x}_{k+1}, \cdots, \boldsymbol{x}_n)$ in Eq. (2). As a result, the following best response mapping for each agent-$k$ is well-defined[1]:

$$\text{Best response mapping:} \qquad BR_k(\boldsymbol{x}_{-k}) \triangleq \arg \max_{\boldsymbol{x}_k \in \mathcal{X}_k} u_k(\boldsymbol{x}_k, \boldsymbol{x}_{-k}). \tag{3}$$

Moreover, the following notion of game Hessian matrix will be useful throughout the paper:

$$\text{Game Hessian:} \qquad H_{\mathcal{G}} = [\nabla_j \nabla_i u_i(\boldsymbol{x})]_{i,j}. \tag{4}$$

The following smoothness property of agents' utility functions will be useful for our analysis.

**Definition 3** (Second-order $L$-smooth utilities). *Given any game $\mathcal{G}$, we say an agent $k$'s utility function $u_k$ is second-order $L$-smooth if there exists a feasible set $\mathcal{F} \subseteq \mathbb{R}^d$ such that for any $\boldsymbol{\delta} \in \mathcal{F}$, the following function $h_k(\cdot, \boldsymbol{\delta}) : \mathbb{R}^{nd} \to \mathbb{R}^d$ defined as*

$$h_k(\boldsymbol{x}; \boldsymbol{\delta}) \triangleq \frac{v_k(\boldsymbol{x}_k + \boldsymbol{\delta}, \boldsymbol{x}_{-k}) - v_k(\boldsymbol{x}_k, \boldsymbol{x}_{-k})}{\|\boldsymbol{\delta}\|}, \tag{5}$$

*is $L$-Lipschitz continuous; that is, for any $\boldsymbol{x}, \boldsymbol{x}' \in \mathcal{X}$, it holds that*

$$\|h_k(\boldsymbol{x}; \boldsymbol{\delta}) - h_k(\boldsymbol{x}'; \boldsymbol{\delta})\| \leq L \|\boldsymbol{x} - \boldsymbol{x}'\|. \tag{6}$$

---

[1]Because the maximizer of a strictly concave function always exists and is unique.

Second-order smoothness requires that, for a given agent $k$, while holding other agents' strategies fixed, the marginal change in $k$'s utility with respect to her own strategy $\boldsymbol{x}_k$ is Lipschitz continuous with respect to the joint strategies $\boldsymbol{x}$ of all agents. This is a relatively mild assumption, as it essentially demands the smoothness of the second-order derivatives of the agent's utility. To see this connection, by letting $\boldsymbol{\delta} \to \boldsymbol{0}$, we have $h_k(\boldsymbol{x}; \boldsymbol{\delta}) \to \frac{\partial^2 u_k}{\partial \boldsymbol{x}_k^2} \cdot \frac{\boldsymbol{\delta}}{\|\boldsymbol{\delta}\|}$. Hence, in this case Condition Eq. (6) becomes the Lipschitz continuity on the Hessian matrix $\frac{\partial^2 u_k}{\partial \boldsymbol{x}_k^2}$ of $h_k$, which is why it is called second-order smoothness.

A few remarks are worth mentioning. First, our attack poison's only a single agent and its success is guaranteed so long as this single agent's utility is second-order smooth. Second, the smooth parameter $L$ in Definition 3 turns out to affect how "controllable" the attack outcome is. Our results will show that clever adversaries would tend to poison an agent with small $L$ parameter. Third, many well-known strongly monotone games exhibit this smoothness property. For example, the $n$-agent Cournot competition is second-order 0-smooth, where $n$-agent Tullock contest is second-order $\mathcal{O}(\sqrt{n})$-smooth (see Appendix A for full descriptions of these games and proofs of these claims in Proposition 2 and 3).

## 2.2 THE $(\alpha, p)$-MULTI-AGENT LEARNING DYNAMICS UNDER BANDIT FEEDBACK

Monotone games are known to have a unique Nash equilibrium, hence a substantial body of recent work develops equilibrium-converging dynamics, including (Bravo et al., 2018; Tatarenko & Kamgarpour, 2020; Cai & Zheng, 2023; Ba et al., 2024). These studies demonstrate that in a strictly monotone game if each agent independently follows such an algorithm, the joint strategies of all agents converge in last iterate to the NE at a polynomial rate. Among these, we are particularly interested in algorithms that operate in a bandit feedback environment, where agents do not have access to their utility functions neither utility gradients, and can only observe the utilities (possibly noisy) resultant from the actions they take at that round. This feedback setting is challenging yet realistic, as it reflects un-coupled learning situations where agents are unaware of their opponents' existence and simply optimize their own utilities selfishly. The following notion is crucial to our analysis.

**Definition 4** $((\alpha, p)$-MAL Dynamics$)$. *We say a (possibly randomized) multi-agent learning (MAL) dynamics $\mathcal{A} = (\mathcal{A}_1, \cdots, \mathcal{A}_n)$ for an $n$-agent game $\mathcal{G}$ with bandit feedback is an $(\alpha, p)$-MAL dynamics if when each agent $i$ uses algorithm $\mathcal{A}_i$ to determine her strategy $x_i^{(t)}$ at time $t$, then their joint strategy sequence $\{\boldsymbol{x}^{(t)}\}_{t=1}^{\infty}$ converges to a Nash Equilibrium $\boldsymbol{x}^*$ in the following sense: there exists positive constants $C, T_0$ such that for any $t > T_0$ we have*

$$\mathbb{E}[\|\boldsymbol{x}^{(t)} - \boldsymbol{x}^*\|_2^p] \leq C^p \cdot t^{-p\alpha}. \tag{7}$$

In other words, an $(\alpha, p)$-MAL dynamics leads to last iterate convergence to some NE of the game $\mathcal{G}$ in the sense that the $p$-th power of the strategy profile difference, evaluated under the $L_2$ norm, has a polynomial convergence rate $\alpha > 0$. In the following section, we shall see that the $\alpha, p$ parameter in $(\alpha, p)$-MAL algorithms fundamentally affects the algorithm's robustness to adversarial corruptions. In the literature of multi-agent learning for monotone games, various $(\alpha, p)$-MAL dynamics have been developed. For instance, the **M**ulti-**A**gent **M**irror **D**escent with Bandit Feedback (MAMD) proposed by Bravo et al. (2018) leads to a $(\frac{1}{6}, 2)$-MAL algorithm, while the Multi-Agent **M**irror **D**escent with **S**elf-**C**oncordant **B**arrier Bandit Learning (MD-SCB) introduced in Ba et al. (2024) leads to a $(\frac{1}{4}, 2)$-MAL dynamics. Notably, while these designs often let every agent use the same learning algorithm, neither our definition of the $(\alpha, p)$-MAL dynamics nor our designed attack later require this restriction — all we assume is that the learning dynamics converge.

## 3 VULNERABILITY OF MAL TO UTILITY POISONING IN MG

In this section, we examine the vulnerability of multi-agent learning (MAL) dynamics. Our main result is the design of a single-agent utility poisoning attack, which is provably successful to any $(\alpha, p)$-MAL algorithm for $\beta$-strongly monotone games. Our results also quantitatively characterize how the attack costs and outcome depend on parameters $\beta, \alpha, p$. The risk of having a poisoned agent within a community is very realistic, hence our attack raises serious concerns regarding naively applying MAL dynamics to setting with potential

adversaries. We conclude this section by showing how the attack can lead to worse outcomes when the adversary can poison even more agents.

## 3.1 Poisoning a Single agent Suffices to Steer the Equilibrium Away

We design an explicit implementation of such a utility poisoning attack, referred to as the **S**ingle-agent **U**tility **S**hifting **A**ttack (SUSA). SUSA first selects a victim agent $k$ to poison, and computes a corrupted utility function $\tilde{u}_k$ for this agent. Then during MAL process, at each round SUSA simply shift the realized utility of agent $k$ from $u_k$ to the $\tilde{u}_k$ with attacking cost $c = |\tilde{u}_k - u_k|$ (hence the name "utility shifting"). The formal definition of SUSA is provided below.

**Definition 5** (Single-agent Utility Shifting Attack)**.** *A Single-agent Utility Shifting Attack (SUSA) against a strategic game instance $\mathcal{G}$ is specified by a tuple $(k, \boldsymbol{\delta}, \Delta)$, which is implemented by the following two stages:*

1. *[**Preparation Stage**] Pre-compute a corrupted utility function for "victim agent" $k$:*
$$\tilde{u}_k(\boldsymbol{x}_k, \boldsymbol{x}_{-k}) = u_k(\boldsymbol{x}_k + \boldsymbol{\delta}, \boldsymbol{x}_{-k}) + \Delta(\boldsymbol{x}_{-k}, \boldsymbol{\delta}), \tag{8}$$
    *where $\Delta : \mathcal{X} \to \mathbb{R}$ is a function depending on the joint strategy $\boldsymbol{x}_{-k}$ and $\boldsymbol{\delta} \in \mathcal{X}_k{}^2$.*

2. *[**Attacking Stage**] During the execution of an MAL algorithm at round $t$, add a corruption*
$$c_t = \tilde{u}_k(\boldsymbol{x}_k^{(t)}, \boldsymbol{x}_{-k}^{(t)}) - u_k(\boldsymbol{x}_k^{(t)}, \boldsymbol{x}_{-k}^{(t)})$$
    *to agent-$k$'s utility observation.*

*In addition, we denote the corrupted game instance as $\tilde{\mathcal{G}}(k, \boldsymbol{\delta}, \Delta)$, in which the $k$-th agent's utility function is replaced with $\tilde{u}_k$.*

A few remarks are worth clarifying. First, $c_t$ is introduced after observing the current round action. This type of corruption is often referred as "strong" corruption in the literature (Liu & Shroff, 2019; Jun et al., 2018; Bogunovic et al., 2021b). Second, SUSA is applicable to any MAL learning dynamics in any games with continuous utility functions, though below we only show that the success of such attacks can be guaranteed for strongly monotone games under certain conditions. Third, SUSA only poisons agent $k$'s utility observations and does not interfere with her actions, though the poisoning amount depends on an action shifting term $\boldsymbol{\delta}$. Essentially, it makes $k$ believe her utilities are drawn from a modified function $\tilde{u}_k$.

Notably, SUSA does *not* attack any other agents, but will nevertheless steer their equilibrium behaviors through influencing the victim $k$'s behaviors. Since agents do not know their true utility functions but only observe the bandit feedback about utilities, this form of corruption not only disrupts the sequential strategy updates of the victim agent $k$, but also influences the dynamical behaviors of all other agents involved.

The following Lemma 1 reveals a nice property of SUSA. That is, if the target game $\mathcal{G}$ is strongly monotone, SUSA preserves the strongly monotone property as long as the $l_2$-norm of the deviation $\|\boldsymbol{\delta}\|_2$ is upper bounded by a constant depending on the game parameters.

**Lemma 1.** *Consider any $\beta$-strongly monotone game $\mathcal{G}(n, \{\mathcal{X}_i\}_{i=1}^n, \{u_i\}_{i=1}^n)$. If $u_k$ is second-order $L$-smooth, then under SUSA the corrupted game $\tilde{G}(k, \boldsymbol{\delta}, \Delta)$ with $\|\boldsymbol{\delta}\|_2 < \beta/L$ remains strongly monotone.*

Lemma 1 indicates that the corrupted game preserves strong monotonicity as long as the norm of the shifting offset $\boldsymbol{\delta}$ is reasonably bounded. This is useful because MAL dynamics retains convergence in the corrupted game (still strongly monotone). As mentioned in Section 2.1, Cournot competition is $\mathcal{O}(1)$-strongly monotone and 0-smooth, so a SUSA using any $\boldsymbol{\delta}$ preserves its monotonicity. In contrast, the Tullock contest is $\mathcal{O}(1)$-strongly monotone but $\mathcal{O}(\sqrt{n})$-smooth, hence the adversary is limited to using $\|\boldsymbol{\delta}\| < \mathcal{O}(1/\sqrt{n})$. This means that as the number of agents increases, it becomes more difficult for SUSA to preserve the monotone property. The proof of Lemma 1 can be found in Appendix B.1.

Our following main theorem of this section, which formally characterizes SUSA's capability.

---

[2]Rigorously, we need to extend the domain $\mathcal{X}_k$ to ensure $u_k(\boldsymbol{x}_k + \boldsymbol{\delta}, \boldsymbol{x}_{-k})$ is well-defined. In fact, any concave extension would suffice, and it does not affect the validity of our theoretical results.

**Theorem 1.** *Let $\mathcal{G}(n, \{\mathcal{X}_i\}_{i=1}^n, \{u_i\}_{i=1}^n)$ be any $\beta$-strongly monotone game (with unique NE $\boldsymbol{x}^*$) and $\mathcal{A}$ be any $(\alpha, p)$-MAL dynamics. Suppose a victim agent-$k$'s utility function $u_k$ is second-order $L$-smooth. Consider $SUSA(k, \boldsymbol{\delta}, \Delta)$ constructed as*

$$\Delta(\boldsymbol{x}_{-k}, \boldsymbol{\delta}) = -u_k(BR_k(\boldsymbol{x}_{-k}), \boldsymbol{x}_{-k}) + u_k(BR_k(\boldsymbol{x}_{-k}) - \boldsymbol{\delta}, \boldsymbol{x}_{-k}) \tag{9}$$

*where $\boldsymbol{\delta}$ is any vector satisfying $\|\boldsymbol{\delta}\|_2 < \beta/L$ and $BR_k(\boldsymbol{x}_{-k}) \triangleq \arg\max_{\boldsymbol{x}_k \in \mathcal{X}_k} u_k(\boldsymbol{x}_k, \boldsymbol{x}_{-k})$ is the best response mapping of agent-$k$. Then the resulting dynamics induced by $\mathcal{A}$ and $SUSA(k, \boldsymbol{\delta}, \Delta)$ converges to some $\tilde{\boldsymbol{x}}^*$, such that*

1. *[**Attack Success**] The deviation to the original NE satisfies*

$$\|\tilde{\boldsymbol{x}}^* - \boldsymbol{x}^*\|_2 \geq \beta \|\boldsymbol{\delta}\|_2 / \sup\{\rho(H_{\mathcal{G}}(\boldsymbol{x})) : \boldsymbol{x} \in [\boldsymbol{x}^*, \tilde{\boldsymbol{x}}^*]\}, \tag{10}$$

   *where $\rho(H_{\mathcal{G}}(\boldsymbol{x}))$ represents the spectral norm of the game's Hessian $H_{\mathcal{G}}$ at $\boldsymbol{x}$, and $[\boldsymbol{x}^*, \tilde{\boldsymbol{x}}^*]$ denotes the segment (with a slight abuse of notation) $\lambda\tilde{\boldsymbol{x}}^* + (1-\lambda)\boldsymbol{x}^*, \lambda \in [0, 1]$.*

2. *[**Sublinear Attack Costs**] The expected total budget satisfies*

$$\mathbb{E}\left[\sum_{t=1}^T |c_t|\right] \leq C_0 \cdot T^{1 - \frac{p\alpha}{p+1}}, \tag{11}$$

   *where the constant $C_0 = CL_1(4L_2 + 5) + 2$, $L_1$ is the Lipschitz constant of all $u_i$ w.r.t. $\boldsymbol{x}_i$ and $L_2$ is the Lipschitz constant of victim agent $k$'s best response $BR_k$ w.r.t. $\boldsymbol{x}_{-k}$.*

As described in Eq. (10) and Eq. (11), the success of a SUSA relies on achieving two goals simultaneously: (1) steering the convergence to a joint strategy profile that is at least a constant distance away from the original NE, and (2) ensuring that the total budget required is $o(T)$. Theorem 1 guarantees both even when the adversary only targets a single agent.

One might wonder, beyond the NE shifting distance guarantee Eq. (10), whether an adversary can have any control over the shifting direction, especially if it aims to influence specific agents' strategies. Our answer is affirmative: although rigorously characterizing the direction $\tilde{\boldsymbol{x}}^* - \boldsymbol{x}^*$ is challenging, we can approximate it as $H_{\mathcal{G}}^{-1}[:, k][\nabla_{kk}u_k]\boldsymbol{\delta}$ (see Appendix B.3), where $H_{\mathcal{G}}^{-1}[:, k]$ is the $k$-th block column of the inverse of $H_{\mathcal{G}}$, and $\nabla_{kk}u_k$ is the Hessian of $u_k$ w.r.t. $\boldsymbol{x}_k$. This suggests that if an adversary possesses some additional global knowledge about the game's Hessian $H_{\mathcal{G}}$, it can further steer the NE deviation in a desired direction.

To get a better sense of the attack success guarantee, we derive constants in Theorem 1 for two examples as follows. The proof can be found in Proposition 2 and 3 in Appendix A.

**Remark 1.** *For $n$-person Cournot competition, the corresponding parameters $L_0, L_1, L_2, \beta$ and $H_{\mathcal{G}}$ specified in Theorem 1 satisfy*

$$L_0 = 0, L_1 = \mathcal{O}(1), L_2 = \mathcal{O}(\sqrt{n}), \beta = \mathcal{O}(1), \sup_{\boldsymbol{x} \in \mathcal{X}}\{\rho(H_{\mathcal{G}}(\boldsymbol{x}))\} = \mathcal{O}(n),$$

*therefore, for an arbitrary $\boldsymbol{\delta}$, SUSA can induce an NE shift $\|\tilde{\boldsymbol{x}}^* - \boldsymbol{x}^*\|_2 \geq \mathcal{O}(n^{-1})$. For $n$-person Tullock contest, we have*

$$L_0 = \mathcal{O}(\sqrt{n}), L_1 = \mathcal{O}(1), L_2 = \mathcal{O}(\sqrt{n}), \beta = \mathcal{O}(1/\sqrt{n}), \sup_{\boldsymbol{x} \in \mathcal{X}}\{\rho(H_{\mathcal{G}}(\boldsymbol{x}))\} = \mathcal{O}(n),$$

*and thus for $\|\boldsymbol{\delta}\| < \mathcal{O}(n^{-1})$, SUSA can induce $\|\tilde{\boldsymbol{x}}^* - \boldsymbol{x}^*\|_2 \geq \mathcal{O}(n^{-\frac{5}{2}})$. Both results are obtained within a total budget $\mathbb{E}\left[\sum_{t=1}^T |c_t|\right] \leq \mathcal{O}(\sqrt{n}T^{1 - \frac{p\alpha}{p+\alpha}})$.*

We observe that while some of the constants in the bounds are $\mathcal{O}(1)$, some of them inevitably depend on $n$ and the specific guarantees vary across different game structures. In our two examples, Cournot games are clearly more vulnerable to attacks compared to Tullock games. One might also wonder the induced deviation from the original NE may be too small when $n$ gets large. As we will discuss in Section 3.2, the adversary can improve the deviation in Eq. (10) by removing the factor $\sup\{\rho(H_{\mathcal{G}})\}$ and may even force the NE deviation to an arbitrary direction if allowed to poison multiple agents.

Next, we discuss the implications of the necessary requirements in Theorem 1, which provide insight into why SUSA can be successful. The first condition demands that the $l_2$-norm of

the deviation $\|\boldsymbol{\delta}\|$, must be bounded by a constant. This is to ensure that the corrupted game remains strongly monotone and thus allows $\mathcal{A}$ to converge to $\tilde{\boldsymbol{x}}^*$, the unique NE of the corrupted game, as indicated by Lemma 1. The second condition provides an explicit construction of the offset function, which is essential for maintaining a sub-linear total budget. In fact, the rationale behind the design of $\Delta$ is to satisfy that for the target agent-$k$ and any $\boldsymbol{x}_{-k} \in \mathcal{X}_{-k}$, $\max_{\boldsymbol{x}_k \in \mathcal{X}_k} \tilde{u}_k(\boldsymbol{x}_k, \boldsymbol{x}_{-k}) = u_k(\tilde{BR}_k(\boldsymbol{x}_{-k}), \boldsymbol{x}_{-k})$, where $\tilde{BR}_k(\boldsymbol{x}_{-k}) = \arg\max_{\boldsymbol{x} \in \mathcal{X}_k} \tilde{u}_k(\boldsymbol{x}, \boldsymbol{x}_{-k}) = BR_k(\boldsymbol{x}_{-k}) - \boldsymbol{\delta}$ is the best response function for agent-$k$ under the corrupted utility $u_k$. In other words, regardless of the opponents' strategies, the best response of any agent under their corrupted utility $\tilde{u}_k$ aligns with the intersection of $\tilde{u}_k$ and the original utility $u_k$. This means that as an agent gradually learns the best response to their opponents' strategies–particularly when converging to $\tilde{\boldsymbol{x}}^*$–the adversary's budget decreases to zero, ultimately resulting in a total budget that is sub-linear in $T$. The proof for Eq. (10) requires establishing a connection between the Jacobian of best response mapping and the game's Hessian and applying Lagrange mean-value theorem for vector-valued functions (Hall & Newell, 1979). The proof for Eq. (11) follows from a careful analysis of the expected budget $\mathbb{E}[\|c_t\|]$ at each round. In fact, we can show that the same convergence rate applies to the corrupted game $\tilde{G}$ and as the joint strategy approaches $\tilde{\boldsymbol{x}}$ at rate $t^{-\alpha}$, $\mathbb{E}[\|c_t\|]$ decreases at rate $t^{-\frac{p\alpha}{p+1}}$. For detailed proof, please refer to Appendix B.2.

## 3.2 Potential Power of Poisoning Many Agents

Theorem 1 demonstrates the significant impact of attacking even a single agent. A natural follow-up question is whether an adversary can gain additional power by poisoning multiple agents. While this is not the primary focus of our work—since it is unclear how realistic or feasible it would be for an adversary to poison many agents—we provide preliminary evidence that such an adversary would indeed have significantly more influence. We illustrate this through an example in the Cournot competition, as detailed below.

**Proposition 1.** *Under the same assumptions stated in Theorem 1, consider an adversary performing $SUSA(k, \boldsymbol{\delta}_k, \Delta_k)$ against all agents $k \in [n]$ in an n-person Cournot competition. Then, the required attacking budget for each agent is still bounded as Eq. (11), and the shifted NE under attack satisfies $\tilde{\boldsymbol{x}}^* - \boldsymbol{x}^* = H_{\mathcal{G}}^{-1} D\boldsymbol{\delta}$, where $H_{\mathcal{G}}$ is the game Hessian defined in Eq. (4), and D is a diagonal block matrix with the i-th diagonal block being the Hessian of $u_i$ w.r.t. $\boldsymbol{x}_i$. As a result,*

1. *picking $\boldsymbol{\delta} = D^{-1} H_{\mathcal{G}} \boldsymbol{v}$ can induce an arbitrary NE deviation direction $\boldsymbol{v}$,*

2. *picking $\boldsymbol{\delta}$ aligning with the direction associated with the largest eigenvalue of $H_{\mathcal{G}}^{-1} D$ induces the largest possible NE deviation distance $\|\tilde{\boldsymbol{x}}^* - \boldsymbol{x}^*\|$, which is at least $\|\boldsymbol{\delta}\|$.*

Although limited to Cournot games, Proposition 1 offers a preview of the additional advantages of attacking multiple agents, namely more refined control over both the magnitude and direction of NE deviation. Compared to Remark 1, attacking multiple agents increases the magnitude of the NE deviation from $\mathcal{O}(1/n)$ to $\mathcal{O}(1)$, and allows for arbitrary directional deviation. We are able to derive a closed-form solution of $\tilde{\boldsymbol{x}}^* - \boldsymbol{x}^*$ thanks to the quadratic utility function in Cournot games which yields a constant game Hessian and a linear best response mapping. For more general game structures, we can show that similar results hold, albeit in an approximate sense, as a strongly convex function can always be approximated by a quadratic form, and the best response mapping can be linearized near $\boldsymbol{x}^*$. However, a rigorous exploration of this is beyond the scope of our current work.

## 4 Intrinsic Trade-Off Between Efficiency and Robustness

Our main result in previous sections (i.e. Theorem 1) reveals an intriguing trade-off between the efficiency and robustness of MAL dynamics: the faster an algorithm converges (i.e., a larger $\alpha$), the smaller corruption needed for inducing an NE shift. For simplicity, we name such outcome as NE Shifting Attack (NSA). While such a trade-off is well-documented in single-agent online learning (Cheng et al., 2024), we are the first to identify it in the multi-agent context. In this section, we explore whether sacrificing the convergence rate of specific MAL dynamics can improve robustness against utility poisoning attacks. We focus on two dynamics: MD-SCB (Ba et al., 2024), the fastest-converging state-of-the-art MAL

algorithm, and MAMD (Bravo et al., 2018), the foundational MAL algorithm for strongly concave games with bandit feedback.

Essentially, our key insight is that the robustness of these dynamics can be improved (i.e., become more resilient to NSA) by simply adjusting the learning rate. Theorem 2 shows how the learning rate of MD-SCB (see Appendix C.1 for details) influences its robustness against general utility poisoning attacks. Originally proposed by Ba et al. (2024), MD-SCB achieves an optimal convergence rate of $\mathcal{O}(d/\sqrt{T})$ without adversarial attacks, if choosing the learning rate $\eta_t \propto t^{-\frac{1}{2}}$ At each round $t$, we consider a general adversary who selects a set of agents $\tilde{\mathcal{I}}_t$ to attack: any agent $\tilde{i} \in \tilde{\mathcal{I}}_t$ will receive an altered utility observation $\tilde{\mu}_{\tilde{i}}^{(t)} \leftarrow \mu_{\tilde{i}}(\boldsymbol{x}_t) + c_{t,\tilde{i}}(\boldsymbol{x}_t)$. Theorem 2 highlights how the algorithm's convergence under such a generic utility poisoning attack depends on both the learning rate $\eta_t$ and the total attack budget $\rho$.

**Theorem 2.** *Consider an adversary that attacks a set of agents $\tilde{\mathcal{I}}_t$ at round $t$ with a total budget satisfying $\sum_{j=1}^{t} \sum_{\tilde{i} \in \tilde{\mathcal{I}}_j} |c_{j,\tilde{i}}| \leq \mathcal{O}(t^\rho)$ for all $t$ and some $\rho \in [0,1]$. By choosing a learning rate sequence $\eta_t = \frac{1}{2d} t^{-\phi}$, for sufficiently large $T$ the last-iterate convergence rate of MD-SCB satisfies $\mathbb{E}\left[\|\hat{\boldsymbol{x}}_T - \boldsymbol{x}^*\|_2^2\right] \leq \max\left\{\mathcal{O}(dT^{\phi-1}), \mathcal{O}(dT^{-\phi}), \mathcal{O}(dT^{\rho-\frac{1}{2}(\phi+1)})\right\}$. Specifically, $\mathbb{E}\left[\|\hat{\boldsymbol{x}}_T - \boldsymbol{x}^*\|_2^2\right] \leq \mathcal{O}\left(dT^{\frac{2(\rho-1)}{3}}\right)$ can be achieved by choosing $\phi = \frac{2}{3}(\rho + \frac{1}{2})$.*

Theorem 2 delivers a strong message: for a given sublinear total budget $\mathcal{O}(T^\rho)$, even as $\rho \to 1$, we can always adjust the learning rate decay to recover last-iterate convergence, making MD-SCB absolutely resilient to NSA[3], although at the cost of potentially slower convergence speed. The proof of Theorem 2 quantifies the bias introduced by attacks in the utility observations of the gradient estimator $\tilde{\boldsymbol{v}}_i^{(t)}$, and tracks the propagation of this bias throughout the convergence analysis (see Appendix C). We will later use a similar approach to establish an analogous result for Algorithm 3 as well.

Theorem 2 provides a potential defense strategy for an algorithm designer, assuming they can estimate the upper bound of the total corruption. To clarify the inherent trade-off between efficiency and robustness in MAL dynamics, we compare this result with our findings in Section 3 from a unified perspective. For this comparison, let $ADV$ represent all possible utility poisoning strategies, and define the following quantity:

$$\rho(\mathcal{C}, \mathcal{A}) = \inf\{\rho \in [0,1] | \mathcal{C} \text{ with budget } \mathcal{O}(T^\rho) \text{ } always \text{ achieve NSA against } \mathcal{A}.\},$$
$$= \sup\{\rho \in [0,1] | \mathcal{C} \text{ with budget } \mathcal{O}(T^\rho) \text{ } cannot \text{ achieve NSA against } \mathcal{A}.\},$$

where $\mathcal{C} \in ADV$ represents a generic utility poisoning adversary. Now, a fundamental question from both the attacker and defender's perspectives is to quantify the following two functions (with $p = 2$ fixed):

$$\rho_-(\alpha) = \inf_{\mathcal{C} \in ADV} \sup_{\mathcal{A} \in MAL(\alpha,2)} \rho(\mathcal{C}, \mathcal{A}), \quad \rho_+(\alpha) = \sup_{\mathcal{A} \in MAL(\alpha,2)} \inf_{\mathcal{C} \in ADV} \rho(\mathcal{C}, \mathcal{A}). \quad (12)$$

Function $\rho_-(\alpha)$ represents the optimal strategy of a utility poisoning adversary: for any MAL dynamics known to enjoy convergence rate $\alpha$, $\rho_-(\alpha)$ denotes the minimum budget level $\rho$ required for NSA. In contrast, $\rho_+(\alpha)$ describes the optimal defense from the perspective of the algorithm designer: given a required convergence rate $\alpha$, $\rho_+(\alpha)$ identifies the most robust algorithm that can withstand NSA of level $\rho$ from any utility poisoning adversary. The well-known max-min inequality also implies the following basic fact.

**Fact 1.** $\rho_-(\alpha) \geq \rho_+(\alpha)$ *for any $\alpha \in [0, 1/4]$.*

Note that we restrict $\alpha \in [0, \frac{1}{4}]$ because $\alpha = \frac{1}{4}$ is proven to be the optimal convergence rate under $p = 2$ in the non-corrupted setting (Ba et al., 2024). In general, one should expect strict inequality in this context (i.e., $\rho_-(\alpha) > \rho_+(\alpha)$), unless $\rho(\mathcal{C}, \mathcal{A})$ has particular special properties. For instance, if $\rho(\mathcal{C}, \mathcal{A})$ is convex in $\mathcal{C}$ and concave in $\mathcal{A}$, the equality must hold, as seen in the well-known minimax theorem from v. Neumann (1928); Sion (1958). In our problem, however, both $\mathcal{C}, \mathcal{A}$ represent algorithms and it is unclear how $\rho$ changes w.r.t.

---

[3]As NSA necessarily leads to divergence, last-iterate convergence must imply resilience to NSA.

$\mathcal{C}$ and $\mathcal{A}$. Thus, an intriguing open question remains: does the equality hold in Fact 1? In other words, does the adversary gain a strict advantage (i.e., $\rho_-(\alpha) > \rho_+(\alpha)$) by first observing the dynamics $\mathcal{A}$ and then designing the attack $\mathcal{C}$?

To better understand the strength of the adversary, our main results in Theorems 1 and 2 provide insights by establishing upper and lower bounds of $\rho_-(\alpha), \rho_+(\alpha)$, respectively, as summarized in Corollary 1.

**Corollary 1.** *Given the definitions of $\rho_-(\alpha)$ and $\rho_+(\alpha)$ in Eq. (12), for some particular MAL algorithm $\mathcal{A}_0$ and adversary $\mathcal{C}_0 \in ADV$, we further define*

$$\rho(\mathcal{A}_0(\alpha)) = \inf_{\mathcal{C} \in ADV} \rho(\mathcal{C}, \mathcal{A} = \mathcal{A}_0(\alpha)), \quad \rho(\alpha, \mathcal{C}_0) = \sup_{\mathcal{A} \in MAL(\alpha, 2)} \rho(\mathcal{C} = \mathcal{C}_0, \mathcal{A}),$$

*where $\mathcal{A}_0(\alpha)$ denotes algorithm $\mathcal{A}_0$ with a set of hyper-parameters that guarantees $MAL(\alpha, 2)$ convergence. Then for any $\alpha \in [0, \frac{1}{4}]$, it holds that*

$$1 - \alpha \leq \rho(\text{MD-SCB}(\alpha)) \leq \rho_+(\alpha) \leq \rho_-(\alpha) \leq \rho(\alpha; SUSA) \leq 1 - \frac{2\alpha}{3}. \tag{13}$$

We defer the proof of Corollary 1 to Appendix D.1 as it is straightforward: the three middle inequalities follow directly from the definitions of $\rho_-, \rho_+$ and Fact 1. The two outer inequalities are derived from a restatement of Theorem 1 and 2. In fact, Theorem 1 quantifies the capability of a particular adversary (i.e., SUSA) and thus establish the upper bounds of $\rho(\alpha; \text{SUSA})$. Theorem 2, on the other hand, characterizes the robustness of a particular algorithm (i.e. MD-SCB), thus gives the lower bounds of $\rho(\text{MD-SCB}(\alpha))$.

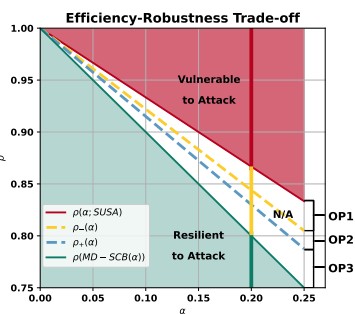

Figure 1: Illustration of the four quantities in Corollary 1, revealing the intrinsic efficiency-robustness trade-off under NSA.

Although the bounds in Corollary 1 are not tight, they highlight an intrinsic trade-off between efficiency and robustness: MAL dynamics with higher convergence rates become more vulnerable—they can withstand less corruption, and adversaries need a smaller budget to attack them. Figure 1 illustrates this trade-off. The yellow dashed line represents $\rho_-(\alpha)$; above it, a budget of $\mathcal{O}(T^\rho)$ suffices for a powerful adversary to induce NSA in any $MAL(\alpha, 2)$ dynamics. The red region, a subset of this area, corresponds to the actual budget required for NSA using our specific attack strategy, SUSA. Conversely, the blue dashed line represents $\rho_+(\alpha)$; below it, the most robust $MAL(\alpha, 2)$ dynamics can withstand corruption of $\mathcal{O}(T^\rho)$ against any adversary. For a specific MAL dynamics like MD-SCB, the green region shows the budget within which it remains resilient to NSA. We only present $\rho \in [0.75, 1]$ because the MAL dynamics with the best convergence rate in such a setting $(\alpha = 0.25)$(Ba et al., 2024) can withstand a corruption level of $\rho = 0.75$ (see Theorem 2). The three gaps in Figure 1 correspond to three open problems: OP1. what is the most budget-efficient adversary for any MAL dynamics; OP2. does an adversary have a strict advantage by observing the dynamics $\mathcal{A}$ first and then design the attack $\mathcal{C}$; and OP3. what is the most robust MAL dynamics resilient to any adversary? We left these for future research.

Such an efficiency-robustness trade-off extends to more general contexts. For example, the MAL algorithm MAMD (detailed in Algorithm 3 in Appendix C.4) exhibits a similar phenomenon. We establish a corresponding result in analogous to Theorem 2 in Proposition 4 in Appendix C.4, showing how to adjust MAMD's learning rate to enhance its robustness. Moreover, this trade-off applies to a broader notion of robustness beyond just resilience to NSA attacks. In Appendix D, we present similar results for a stronger form of robustness that not only ensures resilience to NE shifting but also prevents any possible dampening of the convergence speed.

## 5 EXPERIMENTS

In this section, we validate our theoretical results from Theorem 1 and Theorem 2 by implementing SUSA against MD-SCB on $n$-person Cournot games, whose formal definition is given in Appendix A. Additional results for Algorithm 3 are provided in Appendix E.

**Experimental Setup:** For the $n$-person Cournot game instance, we set $a = 10, b = 0.05$, and the cost $c_i = 1$ for all agents $i \in [n]$. The action space is set to $\mathcal{X}_i = [0, 50]$ and the unique NE of such games can be verified as $\boldsymbol{x}_i^* = \frac{180}{n+1}, i \in [n]$. We let each agent run MD-SCB with tha game size $n \in \{10, 50, 100\}$. The learning rate schedule of MD-SCB is set to $(\eta_t \propto t^{-\phi}, \phi = 0.5, 0.7, 0.9)$, corresponding to convergence rates $\alpha = 0.25, 0.15, 0.05, p = 2$ when no attack is present. We implement SUSA against agent 1, with a fixed $\delta = 10.0$. For each game instance specified by $n$ and the attacked algorithm specified by the convergence rate $\alpha$, we compare the dynamics' behavior both without attack and under SUSA, and report the actual total budget used.

**Result:** The left panel of Figure 2 plots the convergence curve of the $L_2$ square error for $n = 10$, serving as a sanity check to confirm that different learning rate schedules induce different convergence rates for MD-SCB in absence of an adversary. The middle panel shows the outcome of SUSA against MD-SCB with $\alpha = 0.25$, with the $y$-axis representing $L_2$ distance between $\boldsymbol{x}_t$ and NE. The dashed lines display the convergence curve without the adversary, while the solid lines represent the dynamics under SUSA, with different colors indicating results for various game sizes $n$. As shown, for different values of $n$, the attacked dynamics exhibit divergence, as the solid lines saturate and stop decreasing, indicating that the dynamics are being steered to converge to a new point. Additionally, we observe that the induced NE deviation decreases with respect to $n$, which aligns with our theoretical predictions in Theorem1 and Remark 1. The right panel shows the cumulative attack budget $C_t = \sum_{\tau=1}^{t} |c_\tau|$ at each time step $t$ against MD-SCB with different values of $\alpha$. As the results indicate, although all exhibit a sublinear trend, a faster-converging dynamics ($\alpha = 0.25$) requires a smaller attack budget, corroborating our findings in Theorem 1 and 2. Results for additional parameter settings are provided in Appendix E.

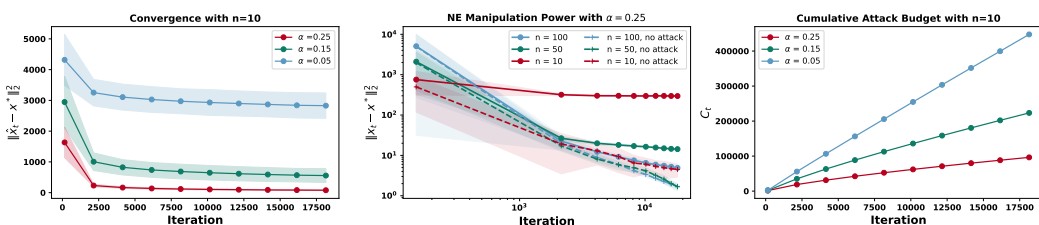

Figure 2: Left: square error of MD-SCB with varying convergence rates. Middle: square error of MD-SCB on different sizes of game instances against SUSA. Right: cumulative attack budget used by SUSA against MD-SCB with varying convergence rates. Error bars represent the 1-$\sigma$ region from 20 independent simulations.

## 6 DISCUSSION

In addition to the three open problems regarding the fundamental limits of attack efficiency and defense robustness raised in Section 4, we discuss more potential limitations and future directions here. One limitation of our setting is that, although SUSA does not require the adversary to have full knowledge of the entire game (such as knowing every agent's utility function, as assumed in some prior literature (Ma et al., 2021)), it does rely on the assumption that the adversary has complete knowledge of the victim agent's utility function. We believe this assumption can be further relaxed. A promising direction for future work is to enhance the applicability and practicality of SUSA by developing methods that assume the adversary has no prior knowledge of the victim agent's utility function and must instead learn it from observed trajectories. Another key question we did not address is the societal impact of utility poisoning attacks, particularly its potential for "steering for social good": as although SUSA could be used by adversaries to induce a harmful NE, it also opens up opportunities for a benevolent social planner to guide the system toward a better social outcome, simply by incentivizing a single agent. These intriguing open problems offer valuable avenues for future exploration.

**Acknowledgment.** This work is supported by Army Research Office Award W911NF23-1-0030, ONR Award N00014-23-1-2802 and NSF Awards CCF-2303372, CNS-2030251,

ECCS-2216970, CMMI-2024774, IIS-2128019. Fan Yao conducted this research while he is visiting the University of Chicago.

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

# Appendix to *Single-Agent Poisoning Attacks Suffice to Ruin Multi-Agent Learning*

## A    EXAMPLES OF SECOND-ORDER SMOOTH MONOTONE GAMES

In this section, we provide two explicit examples of second-order $L$-smooth and $\beta$-strongly monotone games introduced in Definition 3. These examples offer a concrete sense of what we can expect regarding the outcome of a utility poisoning attack in such games. The two cases we will discuss are the $n$-person Cournot Competition Cournot (1838) and the Tullock contest Tullock (2008) (a.k.a. resource allocation auctions with costs).

Cournot competition is an economic model describing a number of firms independently competing on the amount of output they will produce. Each agent-$i$'s pure strategy $x_i$ is the quantity of product, and the payoff is determined by the marginal return of a unit of product which decreases w.r.t. the opponents' total production ($a - b\sum_{i=1}^{n}$), and a marginal cost $c_i$. The utility of agent-$i$ is given by

$$u_i(x_i, x_{-i}) = x_i \left( a - b\sum_{j=1}^{n} x_j \right) - c_i x_i. \tag{14}$$

Tullock contest involves $n$ agents competing for a unit amount of prize in a "winner-takes-all" framework. Each agent $i$'s pure strategy is to choose an effort level $x_i \in [0,1]$, and the winning probability is proportional to each agent's effort level. The prize is awarded to the contestant with the highest relative effort, and the payoff of each agent is the expected reward minus a cost given by some function of the invested effort. Specifically, the payoff functions of agent-$i$ is given by

$$u_i(x_i, x_{-i}) = \frac{x_i}{a + \sum_{j=1}^{n} x_j} - c_i(x_i), \tag{15}$$

where $a > 0$ models the exogenous factor that affects the outcome of the contest, and $c_i : \mathbb{R}_{\geq 0} \to \mathbb{R}_{\geq 0}$ is an increasing cost function.

According to Rosen (1965), a sufficient condition to verify whether a game is $\beta$-strongly monotone (a.k.a. $\beta$-diagonal strict concavity, i.e., $\beta$-DSC) can be summarized in the following Lemma 2.

**Lemma 2.** *A sufficient condition for a game $\mathcal{G}(n, \{\mathcal{X}_i\}, \{u_i\})$ to be $\beta$-DSC is that*

1. *Each $\mathcal{X}_i$ is compact and convex, and $u_i(\boldsymbol{x}_i, \boldsymbol{x}_{-i})$ is concave in $\boldsymbol{x}_i$ and convex in $\boldsymbol{x}_{-i}$.*

2. *$\mathcal{G}$'s negative symmetric game Hessian defined as $-H(\boldsymbol{x}) = -\frac{1}{2}\left( H_{\mathcal{G}}(\boldsymbol{x}) + H_{\mathcal{G}}^{\top}(\boldsymbol{x}) \right)$ is negative definite and its smallest eigenvalue is at least $\beta$.*

And to verify the second-order $L$-smooth condition, we only need to directly apply Definition 3. The following two propositions formalize our claims.

**Proposition 2.** *An $n$-person Cournot competition $\mathcal{G}$ with payoff functions defined by Eq. (14) satisfies:*

1. *each agent's utility function $u_i$ is $(a + b + c_i)$-Lipschitz in $x_i$;*

2. *$u_k$ is $\frac{\sqrt{n-1}}{2}$-Lipschitz in $\boldsymbol{x}_{-k}$;*

3. *$\mathcal{G}$ is $2b$-strongly monotone;*

4. *the spectral norm of the $\mathcal{G}$'s Hessian (defined in Eq. (4)) satisfies $\rho(H_{\mathcal{G}}) \leq (n+1)b$;*

5. *all agents' utility functions are second-order 0-smooth;*

*Proof.* Because $u_i$ is continuous in $x_i$, an Lipschitz constant can be $\max_{\boldsymbol{x} \in \mathcal{X}} |\nabla_{x_i} u_i(x_i, \boldsymbol{x}_{-i})| \leq |a - b\sum_{j=1}^{n} x_j - bx_i - c_i|$. Since $a - b\sum_{j=1}^{n} x_j \geq 0$ and $bx_i \geq 0, c_i \geq 0$, we have $|a - b\sum_{j=1}^{n} x_j - bx_i - c_i| \leq a + b + c_i$.

The best response mapping has a closed form $BR_k(\boldsymbol{x}_{-k}) = \frac{a - b\sum_{j\neq k} x_j - c_k}{2b}$ and we can verify

$$|BR_k(\boldsymbol{x}_{-k}) - BR_k(\boldsymbol{x}'_{-k})| = \frac{1}{2b}\left|b\sum_{j\neq k}(x'_j - x_j)\right| \leq \frac{\sqrt{n-1}}{2}\|\boldsymbol{x}_{-k} - \boldsymbol{x}'_{-k}\|.$$

Therefore, $u_k$ is $\frac{\sqrt{n-1}}{2}$-Lipschitz in $\boldsymbol{x}_{-k}$.

For strongly monotonicity, simple calculation shows the Hessian of $n$-person Cournot game has the following form

$$-H_{ij}(\boldsymbol{x}) = b(1 + \delta_{ij}), \tag{16}$$

where $\delta_{ij} = \mathbb{I}[i = j]$. We can easily verify that $H = b(I + \mathbf{1}\mathbf{1}^\top)$ and the smallest and largest eigenvalues of $H$ are $2b$ and $(n+1)b$. Therefore, the game is $2b$-strongly monotone, and $\rho(H_{\mathcal{G}}) = \rho(H) = (n+1)b$.

To verify second-order smoothness, we directly compute function $h_i(\boldsymbol{x};\delta)$ for agent-$i$ from Eq. (5), which gives

$$\frac{v_i(x_i, \boldsymbol{x}_{-i}) - v_i(x_i + \delta_i, \boldsymbol{x}_{-i})}{\delta}$$
$$= \frac{(a - b\sum_{j=1}^n x_j - c_i - bx_i) - (a - b\sum_{j=1}^n x_j - b\delta - c_i - bx_i - b\delta)}{\delta} = 2b,$$

which means for any $\boldsymbol{x} \in \mathcal{X}$ and $\delta \neq 0$, $h_i(\boldsymbol{x};\boldsymbol{\delta})$ is a constant function. Therefore, the game is second-order smooth for any $L \geq 0$.

$\square$

**Proposition 3.** *Consider an $n$-person Tullock contest $\mathcal{G}$ with payoff functions defined by Eq. (15) in which each agent's cost function $c_i$ is $\beta_i$-strongly convex with its $r$-th order derivative bounded in $[-M_r, M_r], 1 \leq r \leq 3$. Then $\mathcal{G}$ satisfies:*

1. *each agent's utility function $u_i$ is $\left(\frac{1}{a} + M_1\right)$-Lipschitz in $x_i$;*

2. *$u_k$ is $\sqrt{n-1}\left(\frac{2}{a^3\beta_k} + \frac{1}{a^2\beta_k}\right)$-Lipschitz in $\boldsymbol{x}_{-k}$;*

3. *$\mathcal{G}$ is $\left(\min_{i\in[n]}\{\beta_i\} + \frac{a}{(a+n)^3}\right)$-strongly monotone;*

4. *the spectral norm of the $\mathcal{G}$'s Hessian (defined in Eq. (4)) satisfies $\rho(H_{\mathcal{G}}) \leq \frac{n+1}{a^2} + \max_{i\in[n]}\{\beta_i\}$;*

5. *all agents' utility functions are second-order $\left(\frac{4\sqrt{n}}{a^3} + \frac{6\sqrt{n}}{a^4} + \frac{2}{a^3} + M_3\right)$-smooth;*

*Proof.* We prove these claims one by one.

1. Because $u_i$ is continuous in $x_i$, an Lipschitz constant can be

$$\max_{\boldsymbol{x}\in\mathcal{X}} |\nabla_{x_i} u_i(x_i, \boldsymbol{x}_{-i})| = \max_{\boldsymbol{x}\in\mathcal{X}}\left|\frac{a + \sum_{j\neq i} x_j}{(a + \sum_{j=1}^n x_j)^2} - \nabla c_i\right|$$
$$\leq \frac{1}{|a + \sum_{j=1}^n x_j|} + |\nabla c_i| \leq \frac{1}{a} + M_1.$$

2. From the implicit function theorem,

$$\nabla_{\boldsymbol{x}_{-k}} BR_k(\boldsymbol{x}_{-k}) = \frac{\partial \arg\max_t u_k(t, \boldsymbol{x}_{-k})}{\partial \boldsymbol{x}_{-k}} = -\left(\frac{\partial^2 u_k(x_k, \boldsymbol{x}_{-k})}{\partial x_k^2}\right)^{-1} \cdot \frac{\partial^2 u_k(x_k, \boldsymbol{x}_{-k})}{\partial x_k \partial \boldsymbol{x}_{-k}}. \tag{17}$$

From Lagrange mean value theorem for multi-variable functions, there exists some $\boldsymbol{y}_{-k} \in \mathcal{X}_{-k}$ such that

$$BR_k(\boldsymbol{x}_{-k}) - BR_k(\boldsymbol{x}'_{-k}) = \nabla_{\boldsymbol{x}_{-k}}^\top BR_k(\boldsymbol{y}_{-k})(\boldsymbol{x}_{-k} - \boldsymbol{x}'_{-k}).$$

As a result, a Lipschitz constant for function $BR_k(\boldsymbol{x}_{-k})$ can be $\max_{\boldsymbol{y}_{-k} \in \mathcal{X}_{-k}} |\nabla_{\boldsymbol{x}_{-k}} BR_k(\boldsymbol{y}_{-k})|$. Direct calculation shows that for any $i, j$,

$$\nabla_{ii} u_i(\boldsymbol{x}) = -2 \left( a + \sum_{j \neq i} x_j \right) \left( a + \sum_{i=1}^n x_i \right)^{-3} - \nabla^2 c_i(x_i), \tag{18}$$

$$\nabla_j \nabla_i u_i(\boldsymbol{x}) = \left( \sum_{i=1}^n x_i - 2 \sum_{j \neq i}^n x_j - a \right) \left( a + \sum_{i=1}^n x_i \right)^{-3}. \tag{19}$$

Since $u_k$ is $\beta_k$-strongly concave, we have $|\nabla_{ii} u_i(x_i, \boldsymbol{x}_{-i})| \geq \beta_i$, and $|\nabla_j \nabla_i u_i(\boldsymbol{x})| \leq \frac{2}{a^3} + \frac{1}{a^2}$. Substitute them into Eq. (17) we obtain

$$|\nabla_{\boldsymbol{x}_{-k}} BR_k(\boldsymbol{x}_{-k})| \leq \sqrt{n-1} \left( \frac{2}{a^3 \beta_k} + \frac{1}{a^2 \beta_k} \right).$$

3. To see why $\mathcal{G}$ is $\left( \min_{i \in [n]} \{\beta_i\} + \frac{a}{(a+n)^3} \right)$-strongly monotone, we need to show $-H_{\mathcal{G}}$ is positive definite and then pin down its smallest eigenvalue. From Eq. (18) and Eq. (19) we can derive

$$-H_{\mathcal{G}} = \left( a + \sum_{i=1}^n x_i \right)^{-3} \cdot \begin{bmatrix} 2(a + \sum_{i \neq 1} x_i) & a + \sum_{i \notin \{1,2\}} x_i & \cdots \\ a + \sum_{i \notin \{1,2\}} x_i & 2(a + \sum_{i \neq 2} x_i) & \cdots \\ \vdots & \vdots & \ddots \end{bmatrix} + \begin{bmatrix} \nabla^2 c_1(x_1) & 0 & \cdots \\ 0 & \nabla^2 c_2(x_2) & \cdots \\ \vdots & \vdots & \ddots \end{bmatrix}$$

$$\triangleq \left( a + \sum_{i=1}^n x_i \right)^{-3} \cdot M(a, \boldsymbol{x}) + \text{diag}(\nabla^2 c_1(x_1), \cdots, \nabla^2 c_n(x_n)). \tag{20}$$

For any $\boldsymbol{y} = (y_1, \cdots, y_n) \in \mathbb{R}^n$, we have

$$-\boldsymbol{y}^\top M(a, \boldsymbol{x}) \boldsymbol{y} = 2 \sum_{i=1}^n y_i^2 \left( a + \sum_{j \neq i} x_j \right) + 2 \sum_{i<j} y_i y_j \left( a + \sum_{k \notin \{i,j\}} x_k \right)$$

$$= \sum_{i=1}^n y_i^2 \sum_{j \neq i} x_j + \sum_{i=1}^n x_i \left( \sum_{j \neq i} y_j \right)^2 + a \left( \sum_{i=1}^n y_i^2 + \left( \sum_{i=1}^n y_i \right)^2 \right) \tag{21}$$

$$\geq a \|\boldsymbol{y}\|_2^2.$$

Therefore, $\lambda_{\min}(-M(a, \boldsymbol{x})) \geq a$ and

$$\lambda_{\min}(-H_{\mathcal{G}}) \geq \frac{a}{(a + \sum_{i=1}^n x_i)^3} + \beta_0 \geq \frac{a}{(a+n)^3} + \beta_0, \tag{22}$$

where $\beta_0 = \min_{i \in [n]} \{\beta_i\}$.

4. $\mathcal{G}$ is second-order $\left( \frac{4\sqrt{n}}{a^3} + \frac{3\sqrt{n}}{a^4} \right)$-smooth:

Direct calculation shows that

$$h_i(\boldsymbol{x}; \delta) = \frac{v_i(x_i, \boldsymbol{x}_{-i}) - v_i(x_i + \delta, \boldsymbol{x}_{-i})}{\delta}$$

$$= \frac{1}{(A+\delta)^2} + \frac{1}{A(A+\delta)} - \frac{x_i}{A(A+\delta)^2} - \frac{x_i}{A^2(A+\delta)} + \frac{\nabla c_i(x_i + \delta) - \nabla c_i(x_i)}{\delta},$$

where $A = a + \sum_{i=1}^{n} x_i$. From the mean value theorem for vector-valued functions Hall & Newell (1979), there exists $\boldsymbol{y} = \boldsymbol{x}' + \lambda(\boldsymbol{x} - \boldsymbol{x}')$ such that

$$\|h_i(\boldsymbol{x}; \delta) - h_i(\boldsymbol{x}'; \delta)\|_2 \leq \|J(\boldsymbol{y})\| \cdot \|\boldsymbol{x} - \boldsymbol{x}'\|_2,$$

where $J(\boldsymbol{y})$ is the Jacobian matrix of $h_i$ at $\boldsymbol{y}$, and $\|\cdot\|$ denotes the spectral norm of a matrix, defined as

$$\|M\| = \sup_{\|\boldsymbol{x}\|_2 = 1} \|Mx\|_2 = \sqrt{\sigma_{\max}(M^T M)}. \tag{23}$$

Therefore, the smallest constant $L$ such that $\mathcal{G}$ is second-order smooth is

$$\|J(\boldsymbol{y})\| \leq \|J_0\| + \|J_1\| + \|J_2\| + \|J_3\| + \|J_4\|, \tag{24}$$

where $J_0, J_1, J_2, J_3, J_4$ are the Jacobian matrices of functions $f(\boldsymbol{x}) = \frac{1}{(A+\delta)^2}, f(\boldsymbol{x}) = \frac{1}{A(A+\delta)}, f(\boldsymbol{x}) = \frac{x_i}{A(A+\delta)^2}, f(\boldsymbol{x}) = \frac{x_i}{A^2(A+\delta)}, f(\boldsymbol{x}) = \frac{\nabla c_i(x_i+\delta) - \nabla c_i(x_i)}{\delta}$. For a single-valued functions $f$, $J(f) = \nabla f$ and thus $\|J(f)\| = \|\nabla f^\top \nabla f\|$. From straightforward calculation we obtain

$$\|J_0\| = \frac{2}{(A+\delta)^3}\|\boldsymbol{x}\| \leq \frac{2\sqrt{n}}{a^3},$$

$$\|J_1\| = \frac{2A+\delta}{A^2(A+\delta)^2}\|\boldsymbol{x}\| \leq \frac{2\sqrt{n}}{a^3},$$

$$\|J_2\| = \left\| -x_i \frac{3A^2 + 4A\delta + 2\delta^2}{A^2(A+\delta)^4}\boldsymbol{x} + \frac{1}{A(A+\delta)^2}\boldsymbol{e}_i \right\| \leq \frac{3\sqrt{n}}{a^4} + \frac{1}{a^3},$$

$$\|J_3\| = \left\| -x_i \frac{3A^2 + 2A\delta}{A^4(A+\delta)^2}\boldsymbol{x} + \frac{1}{A^2(A+\delta)}\boldsymbol{e}_i \right\| \leq \frac{3\sqrt{n}}{a^4} + \frac{1}{a^3},$$

$$\|J_4\| \leq \|\nabla^3 c_i\| \leq M_3.$$

Hence, the game is second order $\left( \frac{4\sqrt{n}}{a^3} + \frac{6\sqrt{n}}{a^4} + \frac{2}{a^3} + M_3 \right)$-smooth.

5. Upperbound of $\rho(H_\mathcal{G})$.

From Eq. (21) it holds that

$$-\boldsymbol{y}^\top M(a,\boldsymbol{x})\boldsymbol{y} = \sum_{i=1}^{n} y_i^2 \sum_{j \neq i} x_j + \sum_{i=1}^{n} x_i \left( \sum_{j \neq i} y_j \right)^2 + a \left( \sum_{i=1}^{n} y_i^2 + \left( \sum_{i=1}^{n} y_i \right)^2 \right)$$

$$\leq \left[ (n+1)\sum_{i=1}^{n} x_i \right] \|\boldsymbol{y}\|_2^2 + a(n+1)\|\boldsymbol{y}\|_2^2$$

$$= \left[ (n+1)\left( a + \sum_{i=1}^{n} x_i \right) \right] \|\boldsymbol{y}\|_2^2,$$

therefore, by Eq. (20) we have

$$\|\boldsymbol{y}^\top H_\mathcal{G} \boldsymbol{y}\| \leq \frac{n+1}{(a + \sum_{i=1}^{n} x_i)^2}\|\boldsymbol{y}\|_2^2 + \max_{i \in [n]}\{\beta_i\}\|\boldsymbol{y}\|_2^2$$

$$\leq \left( \frac{n+1}{a^2} + \max_{i \in [n]}\{\beta_i\} \right) \|y\|_2^2.$$

Hence, an upperbound of $\rho(H_\mathcal{G})$ is $\frac{n+1}{a^2} + \max_{i \in [n]}\{\beta_i\}$.

$\square$

# B  Proofs in Section 3

In this section, we present the missing proofs from Section 3, including the proofs of Lemma 1 and Theorem 1 which fully characterize the theoretical guarantees of SUSA, and the proof of Proposition 1 along with an extended argument regarding an additional attack success guarantee of SUSA.

Notably, we establish these results under a broader setting where the adversary can attack multiple agents, rather than being restricted to a single agent. Specifically, we consider an adversary targeting a subset of agents $\mathcal{V} \subseteq [n]$, and for each victim agent-$k \in \mathcal{V}$, the adversary performs SUSA$(k, \boldsymbol{\delta}_k, \Delta_k)$ on agent-$k$. We specify such an utility poisoning attack as the tuple $(\mathcal{V}, \{\boldsymbol{\delta}_k\}_{k\in\mathcal{V}}, \{\Delta_k\}_{k\in\mathcal{V}})$, and we denote the corresponding adversary as SUSA$(\mathcal{V}, \{\boldsymbol{\delta}_k\}_{k\in\mathcal{V}}, \{\Delta_k\}_{k\in\mathcal{V}})$.

The proofs provided in this section apply to any SUSA$(\mathcal{V}, \{\boldsymbol{\delta}_k\}_{k\in\mathcal{V}}, \{\Delta_k\}_{k\in\mathcal{V}})$, and thus also apply to the special case when $\mathcal{V} = \{k\}$, which is the setting considered in the main paper. For simplicity of notations, we denote the vector $\boldsymbol{\delta} = (\boldsymbol{\delta}_1, \cdots, \boldsymbol{\delta}_n) \in \mathbb{R}^{nd}$, where $\boldsymbol{\delta}_i = \boldsymbol{\delta}_i$ if $i \in \mathcal{V}$, and $\boldsymbol{\delta}_i = \mathbf{0}$ if $i \notin \mathcal{V}$. In the following, we restate the generalized versions of our results in Section 3 under SUSA$(\mathcal{V}, \{\boldsymbol{\delta}_k\}_{k\in\mathcal{V}}, \{\Delta_k\}_{k\in\mathcal{V}})$ and then present their proofs.

## B.1  Proof of Lemma 1

We prove the following generalized version of Lemma 1 as follows:

**Lemma 3.** *For any $\beta$-strongly monotone and second-order $L$-smooth game $\mathcal{G}(n, \{\mathcal{X}_i\}_{i=1}^n, \{u_i\}_{i=1}^n)$ under SUSA$(\mathcal{V}, \{\boldsymbol{\delta}_k\}_{k\in\mathcal{V}}, \{\Delta_k\}_{k\in\mathcal{V}})$, the corrupted game $\tilde{G}(\mathcal{V}, \{\boldsymbol{\delta}_k\}_{k\in\mathcal{V}}, \{\Delta_k\}_{k\in\mathcal{V}})$ with $\|\boldsymbol{\delta}\|_2 < \beta/L$ remains strongly monotone.*

*Proof.* By the definition of strongly monotonicity, for any $\boldsymbol{x} = (\boldsymbol{x}_1, \cdots, \boldsymbol{x}_n), \boldsymbol{x}' = (\boldsymbol{x}_1', \cdots, \boldsymbol{x}_n') \in \mathcal{X}$, it holds that

$$\sum_{i=1}^n (\boldsymbol{x}_i' - \boldsymbol{x}_i)^\top (v_i(\boldsymbol{x}_i', \boldsymbol{x}_{-i}') - v_i(\boldsymbol{x}_i, \boldsymbol{x}_{-i})) \leq -\beta \|\boldsymbol{x}' - \boldsymbol{x}\|_2^2, \tag{25}$$

where $v_i(\boldsymbol{x}) = \nabla_{\boldsymbol{x}_i} u_i(\boldsymbol{x})$. Without loss of generality let's assume the SUSA targets at all agents (for a agent $i \notin \mathcal{V}$, simply let $\boldsymbol{\delta}_i = \mathbf{0}$ and $\Delta_i \equiv 0$). Then for agent-1's corrupted utility function $\tilde{u}_1$, it holds that $\tilde{v}_1(\boldsymbol{x}) = \nabla_{\boldsymbol{x}_1} \tilde{u}_1(\boldsymbol{x}) = v_1(\boldsymbol{x}_1 + \boldsymbol{\delta}_1, \boldsymbol{x}_{-1})$. By the definition of strictly monotonicity, what we need to show is that there exists a $\beta_0 > 0$ such that

$$\sum_{i=1}^n (\boldsymbol{x}_i' - \boldsymbol{x}_i)^\top (v_i(\boldsymbol{x}_i' + \boldsymbol{\delta}_i, \boldsymbol{x}_{-i}') - v_i(\boldsymbol{x}_i + \boldsymbol{\delta}_i, \boldsymbol{x}_{-i})) < -\beta_0 \|\boldsymbol{x}' - \boldsymbol{x}\|_2^2. \tag{26}$$

According to the second-order $L$-smooth condition, for any $i \in [n]$ and $\boldsymbol{\delta}_i \in \mathbb{R}^d$, the function $h_i(\boldsymbol{x}) : \mathbb{R}^{nd} \to \mathbb{R}^d$ defined as

$$h_i(\boldsymbol{x}) = \frac{v_i(\boldsymbol{x}_i, \boldsymbol{x}_{-i}) - v_i(\boldsymbol{x}_i + \boldsymbol{\delta}_i, \boldsymbol{x}_{-i})}{\|\boldsymbol{\delta}_i\|} \tag{27}$$

is $L$-Lipschitz in $\boldsymbol{x}$. As a result, we have

$$|\text{LHS of Eq. (25)} - \text{LHS of Eq. (26)}|$$

$$\leq \sum_{i=1}^{n} \left| (\boldsymbol{x}_i' - \boldsymbol{x}_i)^\top (v_i(\boldsymbol{x}_i', \boldsymbol{x}_{-i}') - v_i(\boldsymbol{x}_i' + \boldsymbol{\delta}_i, \boldsymbol{x}_{-i}') + v_i(\boldsymbol{x}_i + \boldsymbol{\delta}_i, \boldsymbol{x}_{-i}) - v_i(\boldsymbol{x}_i, \boldsymbol{x}_{-i})) \right|$$

$$= \sum_{i=1}^{n} \|\boldsymbol{\delta}_i\| \cdot \left| (\boldsymbol{x}_i' - \boldsymbol{x}_i)^\top (h_i(\boldsymbol{x}') - h_i(\boldsymbol{x})) \right|$$

$$\leq \sum_{i=1}^{n} \|\boldsymbol{\delta}_i\| \cdot \|\boldsymbol{x}_i' - \boldsymbol{x}_i\| \cdot \|h_i(\boldsymbol{x}') - h_i(\boldsymbol{x})\|$$

$$\leq \sum_{i=1}^{n} \|\boldsymbol{\delta}_i\| \cdot \|\boldsymbol{x}_i' - \boldsymbol{x}_i\| \cdot L\|\boldsymbol{x}' - \boldsymbol{x}\|$$

$$= L\|\boldsymbol{x}' - \boldsymbol{x}\| \cdot \sum_{i=1}^{n} \|\boldsymbol{\delta}_i\| \cdot \|\boldsymbol{x}_i' - \boldsymbol{x}_i\|$$

$$\leq L\|\boldsymbol{x}' - \boldsymbol{x}\| \cdot \left( \sum_{i=1}^{n} \|\boldsymbol{\delta}_i\|^2 \right)^{\frac{1}{2}} \left( \sum_{i=1}^{n} \|\boldsymbol{x}_i' - \boldsymbol{x}_i\|^2 \right)^{\frac{1}{2}}$$

$$\leq L\|\boldsymbol{x}' - \boldsymbol{x}\|^2 \cdot \left( \sum_{i=1}^{n} \|\boldsymbol{\delta}_i\|^2 \right)^{\frac{1}{2}}.$$

Since $\left( \sum_{i=1}^{n} \|\boldsymbol{\delta}_i\|^2 \right)^{\frac{1}{2}} < \frac{\beta}{L}$, there must exist a $\beta_0 \in (0, \beta]$ such that $\left( \sum_{i=1}^{n} \|\boldsymbol{\delta}_i\|^2 \right)^{\frac{1}{2}} \leq \frac{\beta - \beta_0}{L}$. As a result, the difference between the LHS of Eq. (25) and the LHS of Eq. (26) does not exceed $(\beta - \beta_0)\|\boldsymbol{x}' - \boldsymbol{x}\|^2$. Hence, Eq. (26) holds and the corrupted game $\tilde{\mathcal{G}}$ remains $\beta_0$-strongly monotone.

$\square$

### B.2 PROOF OF THEOREM 1

We prove the following generalized version of Theorem 1:

**Theorem 3** (Vulnerability of MAL algorithms for strongly monotone games under $SUSA(\mathcal{V}, \{\boldsymbol{\delta}_k\}_{k\in\mathcal{V}}, \{\Delta_k\}_{k\in\mathcal{V}})$). *Consider a $SUSA(\mathcal{V}, \{\boldsymbol{\delta}_k\}_{k\in\mathcal{V}}, \{\Delta_k\}_{k\in\mathcal{V}})$ to a $\beta$-strongly monotone game $\mathcal{G}(n, \{\mathcal{X}_i\}_{i=1}^{n}, \{u_i\}_{i=1}^{n})$ (with unique NE $\boldsymbol{x}^*$). Suppose the victim agent-$k$'s utility function $u_k$ is second-order $L_0$-smooth for all $k \in \mathcal{V}$. All agents run an $(\alpha, p)$-MAL algorithm $\mathcal{A}$ for $T$ rounds. If the following conditions are satisfied:*

1. *$\|\boldsymbol{\delta}\|_2 < \beta/L_0$,*

2. *$\Delta(\boldsymbol{x}_{-k}, \boldsymbol{\delta}) = -u_k(BR_k(\boldsymbol{x}_{-k}), \boldsymbol{x}_{-k}) + u_k(BR_k(\boldsymbol{x}_{-k}) - \boldsymbol{\delta}, \boldsymbol{x}_{-k}), \forall k \in \mathcal{V}$, where $BR_k(\boldsymbol{x}_{-k}) \triangleq \arg\max_{\boldsymbol{x}_k \in \mathcal{X}_k} u_k(\boldsymbol{x}_k, \boldsymbol{x}_{-k})$ is the best response mapping of agent-$k$,*

3. *$u_i$ is $L_1$-Lipschitz in $\boldsymbol{x}_i, \forall i \in [n]$ and $BR_k$ is $L_2$-Lipschitz in $\boldsymbol{x}_{-k}, \forall k \in \mathcal{V}$.*

*Then, the resulting dynamics induced by $\mathcal{A}$ converges to some $\tilde{\boldsymbol{x}}^*$, such that*

1. *The deviation to the original NE satisfies*

$$\|\tilde{\boldsymbol{x}}^* - \boldsymbol{x}^*\|_2 \geq \beta\|\boldsymbol{\delta}\|_2 / \sup_{\boldsymbol{x} \in [\boldsymbol{x}^*, \tilde{\boldsymbol{x}}^*]} \{\rho(H_{\mathcal{G}}(\boldsymbol{x}))\}, \tag{28}$$

   *where $\rho(H_{\mathcal{G}}(\boldsymbol{x}))$ represents the spectral norm of the game's Hessian $H_{\mathcal{G}}$ at $\boldsymbol{x}$.*

2. *The expected total budget satisfies*

$$\mathbb{E}\left[ \sum_{t=1}^{T} |c_t| \right] \leq C_0 \cdot T^{1 - \frac{p\alpha}{p+1}}, \tag{29}$$

where the constant $C_0 = [CL_1(4L_2 + 5) + 2] \cdot |\mathcal{V}|$.

*Proof.* From Lemma 3 we know that the corrupted game $\tilde{G}$ is strongly monotone and thus also has a unique NE $\tilde{\boldsymbol{x}}^*$. We first give an estimation of the distance between $\boldsymbol{x}^*$ and $\tilde{\boldsymbol{x}}^*$.

Define the best response mapping $f_i(\boldsymbol{x}_{-i}) = \arg\max_t u_i(t, \boldsymbol{x}_{-i})$ for any $i \in [n]$. Then by the definition of NE, $\boldsymbol{x}^* = (\boldsymbol{x}_1^*, \cdots, \boldsymbol{x}_n^*)$ is the unique stationary point of the system

$$\begin{cases} \boldsymbol{x}_1 = f_1(\boldsymbol{x}_2, \boldsymbol{x}_3, \cdots, \boldsymbol{x}_n), \\ \boldsymbol{x}_2 = f_2(\boldsymbol{x}_1, \boldsymbol{x}_3, \cdots, \boldsymbol{x}_n), \\ \vdots \\ \boldsymbol{x}_n = f_n(\boldsymbol{x}_1, \boldsymbol{x}_2, \cdots, \boldsymbol{x}_{n-1}), \end{cases} \tag{30}$$

while $\tilde{\boldsymbol{x}}^* = (\tilde{\boldsymbol{x}}_1^*, \cdots, \tilde{\boldsymbol{x}}_n^*)$ is the unique stationary point of the system

$$\begin{cases} \boldsymbol{x}_1 = f_1(\boldsymbol{x}_2, \boldsymbol{x}_3, \cdots, \boldsymbol{x}_n) - \boldsymbol{\delta}_1, \\ \boldsymbol{x}_2 = f_2(\boldsymbol{x}_1, \boldsymbol{x}_3, \cdots, \boldsymbol{x}_n) - \boldsymbol{\delta}_2, \\ \vdots \\ \boldsymbol{x}_n = f_n(\boldsymbol{x}_1, \boldsymbol{x}_2, \cdots, \boldsymbol{x}_{n-1}) - \boldsymbol{\delta}_n, \end{cases} \tag{31}$$

where $\boldsymbol{\delta}_i = \boldsymbol{0}$ if $i \notin \mathcal{V}$. If we let $F(\boldsymbol{x}) = (f_1(\boldsymbol{x}_2, \boldsymbol{x}_3, \cdots, \boldsymbol{x}_n), f_2(\boldsymbol{x}_1, \boldsymbol{x}_3, \cdots, \boldsymbol{x}_n), \cdots, f_n(\boldsymbol{x}_1, \boldsymbol{x}_2, \cdots, \boldsymbol{x}_{n-1}))$ as an $n$-valued function and $G(\boldsymbol{x}) = F(\boldsymbol{x}) - \boldsymbol{x}$, Eq. (30) and Eq. (31) can be further expressed as

$$G(\boldsymbol{x}^*) = \boldsymbol{0},$$
$$G(\tilde{\boldsymbol{x}}^*) = (\boldsymbol{\delta}_1, \boldsymbol{\delta}_2, \cdots, \boldsymbol{\delta}_n).$$

When $\mathcal{G}$ is twice differentiable, from the mean value theorem for vector valued functions (Hall & Newell, 1979), there exists $\lambda \in [0, 1]$ such that

$$\|G(\boldsymbol{x}^*) - G(\tilde{\boldsymbol{x}}^*)\| \leq \max_{\lambda \in [0,1]} \rho(J(\lambda \tilde{\boldsymbol{x}}^* + (1-\lambda)\boldsymbol{x}^*)) \cdot \|\boldsymbol{x}^* - \tilde{\boldsymbol{x}}^*\|, \tag{32}$$

where $J(\boldsymbol{x})$ is the Jacobian of $\mathcal{G}$ at $\boldsymbol{x}$, and $\rho(J) \triangleq \sup_{\|\boldsymbol{u}\|_2=1} \|J\boldsymbol{u}\|_2$ is the spectral norm of $J$. Let $\rho_{\max}(J) \triangleq \max_{\lambda \in [0,1]}(\rho(J(\lambda \tilde{\boldsymbol{x}}^* + (1-\lambda)\boldsymbol{x}^*)))$, we immediately obtain

$$\|\boldsymbol{x}^* - \tilde{\boldsymbol{x}}^*\| \geq \|\boldsymbol{\delta}\|/\rho_{\max}(J). \tag{33}$$

Next we estimate the upper bound of $\rho(J)$. In fact, from implicit function theorem, we can derive the $(i,j)$-th element of $J$ as

$$J_{i,i} = -1, 1 \leq i \leq n,$$
$$J_{i,j} = \frac{\partial f_i(\boldsymbol{x}_{-i})}{\partial \boldsymbol{x}_j} = \frac{\partial \arg\max_t u_i(t, \boldsymbol{x}_{-i})}{\partial \boldsymbol{x}_j} = -\left(\frac{\partial^2 u_i(\boldsymbol{x}_i, \boldsymbol{x}_{-i})}{\partial \boldsymbol{x}_i^2}\right)^{-1} \cdot \frac{\partial^2 u_i(\boldsymbol{x}_i, \boldsymbol{x}_{-i})}{\partial \boldsymbol{x}_i \partial \boldsymbol{x}_j}.$$

Note that the $n$-by-$n$ matrix $H = \left[\frac{\partial^2 u_i(\boldsymbol{x}_i, \boldsymbol{x}_{-i})}{\partial \boldsymbol{x}_i \partial \boldsymbol{x}_j}\right]$ is exactly the Hessian of $\mathcal{G}$ (i.e., $H_{\mathcal{G}}$), and the diagonal matrix $D = \text{diag}(H_{\mathcal{G}})$. Thus, the Jacobian $J$ can be represented as

$$J = D^{-1} H_{\mathcal{G}}. \tag{34}$$

Since $\mathcal{G}$ is $\beta$-strongly monotone, each diagonal element of $D$ is lower bounded by $\beta > 0$. Hence, the spectral norm of $J$ can be upper bounded by

$$\rho_{\max}(J) \leq \frac{\rho_{\max}(H_{\mathcal{G}})}{\beta}, \tag{35}$$

and

$$\|\boldsymbol{x}^* - \tilde{\boldsymbol{x}}^*\| \geq \beta\|\boldsymbol{\delta}\|/\rho_{\max}(H_{\mathcal{G}}),$$

where $\rho_{\max}(H_{\mathcal{G}}) \triangleq \max_{\lambda \in [0,1]}(\rho(H_{\mathcal{G}}(\lambda \tilde{\boldsymbol{x}}^* + (1-\lambda)\boldsymbol{x}^*)))$.

Next, we derive a upper bound of the total corruption budget needed by an SUSA adversary. Since $\mathcal{A}$ is an $(\alpha, p)$-MAL algorithm and $\tilde{\mathcal{G}}$ is strictly monotone, the resulting playing sequence $\{\boldsymbol{x}_t\}$ converges to the unique NE of $\tilde{\mathcal{G}}$ with a polynomial rate specified by

$$\mathbb{E}[\|\boldsymbol{x}_t - \tilde{\boldsymbol{x}}^*\|_2^p] \leq C^p \cdot t^{-p\alpha}.$$

Let $\gamma$ be any number such that $0 < \gamma < p\alpha$. From Chebyshev's inequality, with probability at least $1 - t^{-p\alpha+\gamma}$, $\|\boldsymbol{x}_t - \tilde{\boldsymbol{x}}^*\|_2^p \leq C^p \cdot t^{-\gamma}$, and the following bounds also hold:

$$\|\boldsymbol{x}_{t,i} - \tilde{\boldsymbol{x}}_i^*\|_2^p \leq C^p \cdot t^{-\gamma}, \forall i \in [n], t \in [T]. \tag{36}$$

Denote the attacking cost at round $t$ for a agent $i \in \mathcal{V}$ as $c_{t,i}$, which is given by $c_{t,i} = |\tilde{u}_i(\boldsymbol{x}_{t,i}, \boldsymbol{x}_{t,-i}) - u_i(\boldsymbol{x}_{t,i}, \boldsymbol{x}_{t,-i})|$. By the choice of $\Delta_i$, it holds that

$$\max_{\boldsymbol{x}_i \in \mathcal{X}_i} \tilde{u}_i(\boldsymbol{x}_i, \boldsymbol{x}_{-i}) = u_i(\tilde{BR}_i(\boldsymbol{x}_{-i}), \boldsymbol{x}_{-i}), \forall \boldsymbol{x}_{-i} \in \mathcal{X}_{-i}. \tag{37}$$

Hence, we can estimate an upper bounded of $c_{t,i}$ as the following:

$$c_{t,i} = |\tilde{u}_i(\boldsymbol{x}_{t,i}, \boldsymbol{x}_{t,-i}) - u_i(\boldsymbol{x}_{t,i}, \boldsymbol{x}_{t,-i})|$$
$$\leq |u_i(\boldsymbol{x}_{t,i}, \boldsymbol{x}_{t,-i}) - u_i(\tilde{BR}_i(\boldsymbol{x}_{t,-i}), \boldsymbol{x}_{t,-i})| + |\tilde{u}_i(\boldsymbol{x}_{t,i}, \boldsymbol{x}_{t,-i}) - u_i(\tilde{BR}_i(\boldsymbol{x}_{t,-i}), \boldsymbol{x}_{t,-i})|$$
$$= |u_i(\boldsymbol{x}_{t,i}, \boldsymbol{x}_{t,-i}) - u_i(\tilde{BR}_i(\boldsymbol{x}_{t,-i}), \boldsymbol{x}_{t,-i})| + |\tilde{u}_i(\boldsymbol{x}_{t,i}, \boldsymbol{x}_{t,-i}) - \max_{\boldsymbol{x}_i \in \mathcal{X}_i} \tilde{u}_i(\boldsymbol{x}_{t,i}, \boldsymbol{x}_{t,-i})|. \tag{38}$$

Since $u_i$ is $L_1$-Lipschitz in $\boldsymbol{x}_i$, and $\tilde{BR}_i(\cdot) = BR(\cdot) - \boldsymbol{\delta}_i$ is $L_2$-Lipschitz, the first part of of RHS of Eq. (38) can be upper bounded by

$$|u_i(\boldsymbol{x}_{t,i}, \boldsymbol{x}_{t,-i}) - u_i(\tilde{BR}_i(\boldsymbol{x}_{t,-i}), \boldsymbol{x}_{t,-i})| \leq L_1 \|\boldsymbol{x}_{t,i} - \tilde{BR}_i(\boldsymbol{x}_{t,-i})\|$$
$$= L_1 \|\boldsymbol{x}_{t,i} - \tilde{\boldsymbol{x}}_i^*\| + L_1 \|\tilde{BR}_i(\boldsymbol{x}_{-i}^*) - \tilde{BR}_i(\boldsymbol{x}_{t,-i})\|$$
$$\leq L_1 \|\boldsymbol{x}_{t,i} - \tilde{\boldsymbol{x}}_i^*\| + L_1 L_2 \|\boldsymbol{x}_{-i}^* - \boldsymbol{x}_{t,-i}\|$$
$$\leq CL_1(1 + L_2)t^{-\gamma/p}. \tag{39}$$

For the second part, we have

$$|\tilde{u}_i(\boldsymbol{x}_{t,i}, \boldsymbol{x}_{t,-i}) - \max_{\boldsymbol{x}_i \in \mathcal{X}_i} \tilde{u}_i(\boldsymbol{x}_{t,i}, \boldsymbol{x}_{t,-i})|$$
$$\leq |\tilde{u}_i(\boldsymbol{x}_{t,i}, \boldsymbol{x}_{t,-i}) - \tilde{u}_i(\boldsymbol{x}_i^*, \boldsymbol{x}_{-i}^*)| + |\tilde{u}_i(\boldsymbol{x}_i^*, \boldsymbol{x}_{-i}^*) - \max_{\boldsymbol{x}_i \in \mathcal{X}_i} \tilde{u}_i(\boldsymbol{x}_{t,i}, \boldsymbol{x}_{t,-i})|. \tag{40}$$

Because $\tilde{u}_i(\boldsymbol{x}_i, \boldsymbol{x}_{-i}) = u_i(\boldsymbol{x}_i + \boldsymbol{\delta}_i, \boldsymbol{x}_{-i}) - u_i(BR_i(\boldsymbol{x}_{-i}), \boldsymbol{x}_{-i}) + u_i(BR_i(\boldsymbol{x}_{-i}) - \boldsymbol{\delta}_i, \boldsymbol{x}_{-i})$ and $u_i$, $BR_i$ are $L_1, L_2$-Lipschitz continuous, $\tilde{u}_i$ is $(L_1 + 2L_1(1 + L2)) = L_1(2L_2 + 3)$-Lipschitz continuous. Therefore, the first part of the RHS of Eq. (40) can be upper bounded by

$$|\tilde{u}_i(\boldsymbol{x}_{t,i}, \boldsymbol{x}_{t,-i}) - \tilde{u}_i(\boldsymbol{x}_i^*, \boldsymbol{x}_{-i}^*)| \leq CL_1(2L_2 + 3)t^{-\gamma/p}. \tag{41}$$

To upper bound the second term of the RHS of Eq. (40), just observe that

$$\max_{\boldsymbol{x}_i \in \mathcal{X}_i} \tilde{u}_i(\boldsymbol{x}_i, \boldsymbol{x}_{-i}) = \max_{\boldsymbol{x}_i \in \mathcal{X}_i} u_i(\boldsymbol{x}_i + \boldsymbol{\delta}_i, \boldsymbol{x}_{-i}) - u_i(BR_i(\boldsymbol{x}_{-i}), \boldsymbol{x}_{-i}) + u_i(BR_i(\boldsymbol{x}_{-i}) - \boldsymbol{\delta}_i, \boldsymbol{x}_{-i})$$
$$= u_i(BR_i(\boldsymbol{x}_{-i}), \boldsymbol{x}_{-i}) - u_i(BR_i(\boldsymbol{x}_{-i}), \boldsymbol{x}_{-i}) + u_i(BR_i(\boldsymbol{x}_{-i}) - \boldsymbol{\delta}_i, \boldsymbol{x}_{-i})$$
$$= u_i(BR_i(\boldsymbol{x}_{-i}) - \boldsymbol{\delta}_i, \boldsymbol{x}_{-i}). \tag{42}$$

Using Eq. (42), we can obtain

$$\left| \tilde{u}_i(\boldsymbol{x}_i^*, \boldsymbol{x}_{-i}^*) - \max_{\boldsymbol{x}_i \in \mathcal{X}_i} \tilde{u}_i(\boldsymbol{x}_{t,i}, \boldsymbol{x}_{t,-i}) \right|$$
$$= \left| \max_{\boldsymbol{x}_i \in \mathcal{X}_i} \tilde{u}_i(\boldsymbol{x}_i, \boldsymbol{x}_{-i}^*) - \max_{\boldsymbol{x}_i \in \mathcal{X}_i} \tilde{u}_i(\boldsymbol{x}_{t,i}, \boldsymbol{x}_{t,-i}) \right|$$
$$= \left| u_i(BR_i(\boldsymbol{x}_{-i}^*) - \boldsymbol{\delta}_i, \boldsymbol{x}_{-i}^*) - u_i(BR_i(\boldsymbol{x}_{t,-i}) - \boldsymbol{\delta}_i, \boldsymbol{x}_{t,-i}) \right|$$
$$\leq L_1(\|BR_i(\boldsymbol{x}_{-i}^*) - BR_i(\boldsymbol{x}_{t,-i})\| + \|\boldsymbol{x}_{-i}^* - \boldsymbol{x}_{t,-i}\|)$$
$$\leq CL_1(L_2 + 1)t^{-\gamma/p}. \tag{43}$$

Assembling Eq. (39), Eq. (41), Eq. (40) and Eq. (43) together, from Eq. (38) we conclude that with probability at least $1 - t^{-p\alpha+\gamma}$, it holds that

$$c_{t,i} \leq CL_1(1 + L_2)t^{-\gamma/p} + CL_1(2L_2 + 3)t^{-\gamma/p} + CL_1(L_2 + 1)t^{-\gamma/p}$$
$$= CL_1(4L_2 + 5)t^{-\gamma/p}. \tag{44}$$

On the other hand, in the event (with a probability at most $t^{-p\alpha+\gamma}$) that Eq. Eq. (36) do not hold, we can use the fact that $u_i \in [0, M]$ to give a trivial upper bound of $c_{t,i}$ as follows:

$$\begin{aligned}
c_{t,i} &= |\tilde{u}_1(\boldsymbol{x}_{t,i}, \boldsymbol{x}_{t,-i}) - u_1(\boldsymbol{x}_{t,i}, \boldsymbol{x}_{t,-i})| \\
&= |u_i(\boldsymbol{x}_{t,i} + \boldsymbol{\delta}_i, \boldsymbol{x}_{t,-i}) - u_i(BR_i(\boldsymbol{x}_{t,-i}), \boldsymbol{x}_{t,-i}) + u_i(BR_i(\boldsymbol{x}_{t,-i}) - \boldsymbol{\delta}_i, \boldsymbol{x}_{t,-i}) - u_i(\boldsymbol{x}_{t,i}, \boldsymbol{x}_{t,-i})| \\
&\leq |u_i(\boldsymbol{x}_{t,i} + \boldsymbol{\delta}_i, \boldsymbol{x}_{t,-i}) - u_i(BR_i(\boldsymbol{x}_{t,-i}), \boldsymbol{x}_{t,-i})| + |u_i(BR_i(\boldsymbol{x}_{t,-i}) - \boldsymbol{\delta}_i, \boldsymbol{x}_{t,-i}) - u_i(\boldsymbol{x}_{t,i}, \boldsymbol{x}_{t,-i})| \\
&\leq 2M. \tag{45}
\end{aligned}$$

Putting Eq. (44) and Eq. (45) together, the expected corruption on agent-$i$ the adversary needs at round $t$ can be thus upper bounded by

$$\begin{aligned}
\mathbb{E}[c_{t,i}] &\leq (1 - t^{-p\alpha+\gamma}) \cdot CL_1(4L_2 + 5)t^{-\gamma/p} + t^{-p\alpha+\gamma} \cdot 2M \\
&\leq [CL_1(4L_2 + 5) + 2M]t^{-\frac{p\alpha}{p+1}},
\end{aligned}$$

and the optimal order is achieved when $\gamma = \frac{p^2}{p+1} \cdot \alpha$. As a result, the total expected corruption is upper bounded by

$$\mathbb{E}\left[\sum_{i=1}^{n}\sum_{t=1}^{T} c_{t,i}\right] \leq [CL_1(4L_2 + 5) + 2M]|\mathcal{V}| \cdot T^{1 - \frac{p\alpha}{p+1}}. \tag{46}$$

$\square$

### B.3 A General Argument of Attacking Ability and Proof of Proposition 1

We first provide an argument regarding the relationship between $\boldsymbol{\delta}$ and $\tilde{\boldsymbol{x}}^* - \boldsymbol{x}^*$ in general games, and then present the proof of Proposition 1, which rigorously establishes the corresponding results for the case of Cournot competition.

As pointed out in the proof of Theorem 1, the converged point $\tilde{\boldsymbol{x}}^* = (\tilde{\boldsymbol{x}}_1^*, \cdots, \tilde{\boldsymbol{x}}_n^*)$ under $\text{SUSA}(\mathcal{V}, \{\boldsymbol{\delta}_k\}_{k\in[n]}, \{\Delta_k\}_{k\in[n]})$ can be described by the solution to the following system:

$$\begin{cases}
\boldsymbol{x}_1 = f_1(\boldsymbol{x}_2, \boldsymbol{x}_3, \cdots, \boldsymbol{x}_n) - \boldsymbol{\delta}_1, \\
\boldsymbol{x}_2 = f_2(\boldsymbol{x}_1, \boldsymbol{x}_3, \cdots, \boldsymbol{x}_n) - \boldsymbol{\delta}_2, \\
\vdots \\
\boldsymbol{x}_n = f_n(\boldsymbol{x}_1, \boldsymbol{x}_2, \cdots, \boldsymbol{x}_{n-1}) - \boldsymbol{\delta}_n,
\end{cases} \tag{47}$$

where the function $f_i(\boldsymbol{x}_{-i}) = \arg\max_t u_i(t, \boldsymbol{x}_{-i})$ is the best response mapping for agent $i$. And $\boldsymbol{x}^* = (\boldsymbol{x}_1^*, \cdots, \boldsymbol{x}_n^*)$ is the solution to

$$\begin{cases}
\boldsymbol{x}_1 = f_1(\boldsymbol{x}_2, \boldsymbol{x}_3, \cdots, \boldsymbol{x}_n), \\
\boldsymbol{x}_2 = f_2(\boldsymbol{x}_1, \boldsymbol{x}_3, \cdots, \boldsymbol{x}_n), \\
\vdots \\
\boldsymbol{x}_n = f_n(\boldsymbol{x}_1, \boldsymbol{x}_2, \cdots, \boldsymbol{x}_{n-1}).
\end{cases} \tag{48}$$

Let $F(\boldsymbol{x}) = (f_1(\boldsymbol{x}_2, \boldsymbol{x}_3, \cdots, \boldsymbol{x}_n), f_2(\boldsymbol{x}_1, \boldsymbol{x}_3, \cdots, \boldsymbol{x}_n), \cdots, f_n(\boldsymbol{x}_1, \boldsymbol{x}_2, \cdots, \boldsymbol{x}_{n-1}))$ as an $n$-valued function and $G(\boldsymbol{x}) = F(\boldsymbol{x}) - \boldsymbol{x}$, we have

$$\begin{aligned}
G(\boldsymbol{x}^*) &= \boldsymbol{0}, \\
G(\tilde{\boldsymbol{x}}^*) &= (\boldsymbol{\delta}_1, \boldsymbol{\delta}_2, \cdots, \boldsymbol{\delta}_n).
\end{aligned}$$

For a small perturbation $\Delta_{\boldsymbol{x}}$, we can approximate the system $G$ around $\boldsymbol{x}^*$ with a linearized version:

$$G(\boldsymbol{x}^* + \Delta_{\boldsymbol{x}}) = G(\boldsymbol{x}^*) + J(G(\boldsymbol{x}^*))\Delta_{\boldsymbol{x}} + \mathcal{O}(\|\Delta_{\boldsymbol{x}}\|^2). \tag{49}$$

If we view the target point to steer $\tilde{\boldsymbol{x}}^*$ as $\boldsymbol{x}^* + \Delta_{\boldsymbol{x}}$, it holds that

$$\boldsymbol{\delta} = G(\tilde{\boldsymbol{x}}^*) = G(\boldsymbol{x}^*) + J(G(\boldsymbol{x}^*))(\tilde{\boldsymbol{x}}^* - \boldsymbol{x}^*) + \mathcal{O}(\|\Delta_{\boldsymbol{x}}\|^2) = J(G(\boldsymbol{x}^*))(\tilde{\boldsymbol{x}}^* - \boldsymbol{x}^*) + \mathcal{O}(\|(\tilde{\boldsymbol{x}}^* - \boldsymbol{x}^*)\|^2). \tag{50}$$

From Eq. (34) we know that $J(G(\boldsymbol{x}^*)) = D^{-1}(\boldsymbol{x}^*)H_{\mathcal{G}}(\boldsymbol{x}^*)$, where $H_{\mathcal{G}}(\boldsymbol{x}^*)$ is the game Hessian at $\boldsymbol{x} = \boldsymbol{x}^*$, and $D^{-1}(\boldsymbol{x}^*)$ is the block diagonal matrix with each block element being the inverse Hessian of $u_i$ w.r.t. $\boldsymbol{x}_i^*$. Therefore, we argue the following in the sense of approximation:

1. if the adversary wants to induce a particular deviation direction $\boldsymbol{v}$, it can simply pick $\boldsymbol{\delta}$ to be $D^{-1}(\boldsymbol{x}^*)H_{\mathcal{G}}(\boldsymbol{x}^*)\boldsymbol{v}$;

2. if the goal is to induce largest deviation as possible, it can pick $\delta$ to align with the direction corresponding to the smallest eigenvalues of $D^{-1}(\boldsymbol{x}^*)H_{\mathcal{G}}(\boldsymbol{x}^*)$.

And for the special case when SUSA only has one victim agent $k$, it need to pick

$$\boldsymbol{\delta}_k = \text{diag}([\nabla_{ii}^{-1} u_i]_{i=1}^n) H_{\mathcal{G}}[:, k]\boldsymbol{v} \tag{51}$$

in order to induce an NE shift $\boldsymbol{v} = \tilde{\boldsymbol{x}}^* - \boldsymbol{x}^*$, and for any $\boldsymbol{\delta}$, it will cause an NE shifting characterized by

$$\boldsymbol{v} = H_{\mathcal{G}}^{-1}[:, k][\nabla_{kk} u_k]\boldsymbol{\delta}, \tag{52}$$

which echoes our argument after Theorem 1. In addition, since $\beta$-strongly monotonicity implies that each diagonal element of $D$ is lower bounded by $\beta > 0$, we know the smallest eigenvalue of $J$ can be upper bounded by

$$\rho_{\min}(J) \leq \frac{\rho_{\min}(H_{\mathcal{G}})}{\min_{k \in [n]} \beta_k} = \frac{\beta}{\min_{k \in [n]} \beta_k} \leq 1, \tag{53}$$

where $\rho_{\min}(H_{\mathcal{G}}) = \beta$ because $\mathcal{G}$ is $\beta$-strongly monotone, and $\beta_k$ is a constant such that $u_k$ is $\beta_k$-strongly concave, meaning $\beta_k \geq \beta$. As a result, we have

$$\|\tilde{\boldsymbol{x}}^* - \boldsymbol{x}^*\| = \rho_{\min}(J(G(\boldsymbol{x}^*)))^{-1}\|\boldsymbol{\delta}\| \geq \|\boldsymbol{\delta}\|. \tag{54}$$

We note that such an argument is not rigorous and only holds in the sense of approximation, as we omit the high-order term $\mathcal{O}(\|\Delta_{\boldsymbol{x}}\|^2)$ in Eq. (49). However, this argument becomes a rigorous proof for Cournot competition, as the Jacobian of system $G$ becomes a constant matrix and the high-order term $\mathcal{O}(\|\Delta_{\boldsymbol{x}}\|^2)$ in Eq. (49) disappears.

*Proof.* For Cournot competition, since the Jacobian of $G$ is a constant matrix and we do not have the high-order term $\mathcal{O}(\|\Delta_{\boldsymbol{x}}\|^2)$ in Eq. (49). Hence, from Eq. (49) we have

$$G(\boldsymbol{x}^* + \Delta_{\boldsymbol{x}}) = G(\boldsymbol{x}^*) + D^{-1}H_{\mathcal{G}}\Delta_{\boldsymbol{x}}, \tag{55}$$

which means that for any particular $\boldsymbol{\delta}$, it will induce

$$\tilde{\boldsymbol{x}}^* - \boldsymbol{x}^* = H_{\mathcal{G}}^{-1}D\boldsymbol{\delta}. \tag{56}$$

As a result, to induce a particular NE deviation direction $\boldsymbol{v}$, we only need to pick $\boldsymbol{\delta} = D^{-1}H_{\mathcal{G}}\boldsymbol{v}$. To induce the largest possible deviation distance $\|\tilde{\boldsymbol{x}}^* - \boldsymbol{x}^*\|$, it need to pick $\boldsymbol{\delta}$ such that $\boldsymbol{\delta}$ aligns with the direction corresponding to the largest eigenvalues of $H_{\mathcal{G}}^{-1}D$, which is also the direction corresponding to the smallest eigenvalues of $D^{-1}H_{\mathcal{G}}$. □

## C  TECHNICAL DETAILS FOR SECTION 4

In this section, we provide the technical details related to Section 4 that we do not have space to include in the main paper. These include essential details for Algorithm 1 and 3, as well as the proofs of Theorem 2, Proposition 4, and Corollary 1.

## C.1 Essential Details for Algorithm 1

In this subsection, we outline all the necessary details for Algorithm 1. The algorithm requires selecting a self-concordant barrier $\mathcal{R}_i$ (see Definition 6) and using a specific prox-mapping, $\mathcal{P}_{\mathcal{R}_i}(\boldsymbol{x}_i^{(t)}, \tilde{\boldsymbol{v}}_i^{(t)}, \eta_t, \lambda_i, \beta)$ (see Definition 7), to update the action for the subsequent round.

---

**Algorithm 1:** MD-SCB under Utility Poisoning Attack (Ba et al., 2024)

---

**Input:** step-size $\eta_t = \frac{t^{-\phi}}{2d}$, weight $\lambda_i > 0$, game's strongly concave parameter $\beta > 0$,
and self-concordant barrier $\mathcal{R}_i : \text{int}(\mathcal{X}_i) \to \mathbb{R}$

1   Initialize: Let $\boldsymbol{x}_i^{(t)} = \arg\min_{\boldsymbol{x}_i \in \mathcal{X}_i} \mathcal{R}_i(\boldsymbol{x}_i)$ for all $i \in [n]$
2   **for** $t \in [T]$ **do**
3     **for** $i \in [n]$ **do**
4       Set $A_i^{(t)} \leftarrow (\nabla^2 \mathcal{R}_i(\boldsymbol{x}_i^{(t)}) + \frac{\eta_t \beta(t+1)}{\lambda_i} I_{d_i})^{-\frac{1}{2}}$
5       Draw direction $\boldsymbol{z}_i^{(t)}$ uniformly from unit sphere $\mathbb{S}^{d_i}$
6       Play action $\hat{\boldsymbol{x}}_i^{(t)} \leftarrow \boldsymbol{x}_i^{(t)} + A_i^{(t)} \boldsymbol{z}_i^{(t)}$
7     Receive $\tilde{\mu}_{\tilde{i}}^{(t)} \leftarrow \mu_{\tilde{i}}(\hat{\boldsymbol{x}}_t) + c_{t,\tilde{i}}(\hat{\boldsymbol{x}}_t)$ for $\tilde{i} \in \tilde{\mathcal{I}}_t$, $\tilde{\mu}_i^{(t)} \leftarrow \mu_i(\hat{\boldsymbol{x}}_t)$ for $i \in [n] \setminus \tilde{\mathcal{I}}_t$
8     **for** $i \in [n]$ **do**
9       Compute gradient $\tilde{\boldsymbol{v}}_i^{(t)} \leftarrow d_i \tilde{\mu}_i^{(t)} (A_i^{(t)})^{-1} \boldsymbol{z}_i^{(t)}$
10      Update action $\boldsymbol{x}_i^{(t+1)} \leftarrow \mathcal{P}_{\mathcal{R}_i}(\boldsymbol{x}_i^{(t)}, \tilde{\boldsymbol{v}}_i^{(t)}, \eta_t, \lambda_i, \beta)$

---

**Definition 6** (Self-concordance). *A function $\mathcal{R} : int(\mathcal{A}) \to \mathbb{R}$ is self-concordant if it satisfies*

1. *$\mathcal{R}$ is three times continuously differentiable, convex, and approaches infinity along any sequence of points approaching the boundary of $int(\mathcal{A})$.*

2. *For every $h \in \mathbb{R}^d$ and $x \in int(\mathcal{A})$, $|\nabla^3 \mathcal{R}(x)[h,h,h]| \leq 2\left(h^\top \nabla^2 \mathcal{R}(x)h\right)^{\frac{3}{2}}$ holds, where $|\nabla^3 \mathcal{R}(x)[h,h,h] := \frac{\partial^3 \mathcal{R}}{\partial t_1 \partial t_2 \partial t_3}(x + t_1 h + t_2 h + t_3 h)|_{t_1=t_2=t_3=0}$.*

*In addition to these two conditions, if for every $h \in \mathbb{R}^d$ and $x \in int(\mathcal{A})$, $|\nabla \mathcal{R}(x)^\top h \leq \nu^{\frac{1}{2}}(h^\top \nabla^2 \mathcal{R}(x)h)^{\frac{1}{2}}|$ for a positive real number $\nu$, $\mathcal{R}$ is $\nu$-self-concordant.*

**Definition 7** (Specific Prox-mapping). *The prox-mapping exploited in Algorithm 1 is defined as*

$$\mathcal{P}_{\mathcal{R}}(x, \hat{v}, \lambda, \eta) = \arg\min_{x' \in x} \langle \hat{v}, x - x' \rangle + \frac{\eta \beta(t+1)}{2\lambda}\|x - x'\|_2^2 + D_{\mathcal{R}}(x', x),$$

*where $D_{\mathcal{R}}(x', x)$ is the Bregman divergence that measures the difference between the values of a convex $\mathcal{R}$ at two points, $x$ and $x'$, combined with the local linear approximation of $\mathcal{R}$ around $x$, described by the following equation*

$$D_{\mathcal{R}}(x', x) = \mathcal{R}(x') - \mathcal{R}(x) - \langle \nabla \mathcal{R}(x), x' - x \rangle.$$

## C.2 Proof of Theorem 2

**Theorem** (Theorem 2 restated). *Consider an adversary that attacks a set of agents $\tilde{\mathcal{I}}_t$ at round $t$ with a total budget satisfying $\sum_{j=1}^t \sum_{\tilde{i} \in \tilde{\mathcal{I}}_j} |c_{j,\tilde{i}}| \leq \mathcal{O}(t^\rho)$ for all $t$ and some $\rho \in [0,1]$. By choosing a learning rate sequence $\eta_t = \frac{1}{2d} t^{-\phi}$, for sufficiently large $T$ the last-iterate convergence rate of Algorithm 1 satisfies $\mathbb{E}\left[\|\hat{\boldsymbol{x}}_T - \boldsymbol{x}^*\|_2^2\right] \leq \max\left\{\mathcal{O}(dT^{\phi-1}), \mathcal{O}(dT^{-\phi}), \mathcal{O}(dT^{\rho-\frac{1}{2}(\phi+1)})\right\}$. Specifically, $\mathbb{E}\left[\|\hat{\boldsymbol{x}}_T - \boldsymbol{x}^*\|_2^2\right] \leq \mathcal{O}\left(dT^{\frac{2(\rho-1)}{3}}\right)$ can be achieved by choosing $\phi = \frac{2}{3}(\rho + \frac{1}{2})$.*

*Proof.* The proof outlined below is motivated by the observation that we can quantify the bias caused by adversarial attack in utility observation in the a Simultaneous Perturbation

Stochastic Approximation (SPSA) estimator, which shares a similar spirit to the gradient estimation under corruption lemma in Cheng et al. (2024). In particular, we can extend Lemma 3.5 in Ba et al. (2024) to incorporate adversarial attacks as follows.

**Lemma 4** (Extended Lemma 3.5 in (Ba et al., 2024))**.** *Suppose that $\mu_i$ is a concave function and $A_i \in \mathbb{R}^{d_i \times d_i}$ is an convertible matrix for each $i \in [n]$, we define the smoothed version of $\mu_i$ with respect to $A_i$ by $\hat{\mu}_i(\boldsymbol{x}) = \mathbb{E}_{w_i \sim \mathbb{B}^{d_i}} \mathbb{E}_{\boldsymbol{z}_{-i} \sim \prod_{j \neq i} \mathbb{S}^{d_j}} [\mu_i(\boldsymbol{x}_i + A_i w_i; \hat{\boldsymbol{x}}_{-i})]$ where $\mathbb{S}^{d_i}$ is a $d_i$ dimensional unit sphere, $\mathbb{B}^{d_i}$ is a $d_i$-dimensional ball and $\hat{\boldsymbol{x}}_i = \boldsymbol{x}_i + A_i \boldsymbol{z}_i$ for all $i \in [n]$. Then, the following statement hold true*

$$\nabla_i \hat{\mu}_i(\boldsymbol{x}) + \boldsymbol{b}_i = \mathbb{E}[d_i \mu_i(\hat{\boldsymbol{x}}_i; \hat{\boldsymbol{x}}_{-i})(A_i)^{-1} z_i | \boldsymbol{x}],$$

*where $\boldsymbol{b}_i = d_i \mathbb{E}[c_i(\hat{\boldsymbol{x}}_i; \hat{\boldsymbol{x}}_{-i})(A_i)^{-1} \boldsymbol{z}_i | \boldsymbol{x}]$.*

**Remark 2.** *The key difference between utility shifting attack and random noise is $\boldsymbol{b}_i \neq 0$. This is because the corruption $c_i(\hat{\boldsymbol{x}}_i; \hat{\boldsymbol{x}}_{-i})$ depends on the randomly sampled vector $\boldsymbol{z}_i$ through $\hat{\boldsymbol{x}}_i$, which makes*

$$\mathbb{E}[c_i(\hat{\boldsymbol{x}}_i; \hat{\boldsymbol{x}}_{-i})(A_i)^{-1} \boldsymbol{z}_i | \boldsymbol{x}] \neq \mathbb{E}[c_i(\hat{\boldsymbol{x}}_i; \hat{\boldsymbol{x}}_{-i})(A_i)^{-1} | \boldsymbol{x}] \mathbb{E}[\boldsymbol{z}_i | \boldsymbol{x}].$$

*Thus, we need to control the cumulative impact of the bias in gradient caused by adversarial attack.*

We then carry this bias $\boldsymbol{b}_i$ throughout the analysis, closely following the original proof developed by Ba et al. (2024). Finally, we derive a result demonstrating how the decreasing speed of the learning rate affects convergence and its associated robustness to adversarial attacks.

To prepare for the proof, we first extend the Lemma 2.12 and Lemma 3.10 in Ba et al. (2024) to incorporate adversarial attacks.

**Lemma 5** (Extended Lemma 2.12 in (Ba et al., 2024))**.** *Suppose that the iterate $\{\boldsymbol{x}^{(t)}\}_{t \geq 1}$ is generated by Algorithm 2 and let each corrupted utility value $\tilde{\mu}^t$ satisfy that $|\tilde{\mu}^t(\boldsymbol{x})| \leq 1$ for all $\boldsymbol{x} \in \mathcal{X}$ and $0 < \eta_t \leq \frac{1}{2d}$, we have*

$$D_{\mathcal{R}}(p, \boldsymbol{x}_{t+1}) + \frac{\eta_t \beta(t+1)}{2} \|x^{(t+1)} - p\|^2 \leq D_{\mathcal{R}}(p, \boldsymbol{x}_t) + \frac{\eta_t \beta(t+1)}{2} \|\boldsymbol{x}_t - p\|^2 + 2\eta_t^2 \|A^{(t)} \tilde{\boldsymbol{v}}^t\|^2 + \eta_t \langle \tilde{\boldsymbol{v}}^t, x^{(t)} - p \rangle,$$

*where $p \in \mathcal{X}$ and the sequence $\{\eta_t\}_{t \geq 1}$ is assumed to be non-increasing.*

---

**Algorithm 2:** Single Agent Mirror Descent Self-Concordant Barrier with Bandit Feedback under Utility Shifting Attack

---

**Input:** step size $\eta_t > 0$, module $\beta > 0$, and barrier $\mathcal{R} : \text{int}(\mathcal{X}) \to \mathbb{R}$

1 Initialize $\boldsymbol{x}^{(1)} = \arg\min_{a \in \mathcal{X}} \mathcal{R}(\boldsymbol{x})$
2 **for** $t \in [T]$ **do**
3      set $A^{(t)} \leftarrow (\nabla^2 R(\boldsymbol{x}^{(t)}) + \eta_t \beta(t+1) I_d)^{-1/2}$
4      draw $\boldsymbol{z}^{(t)} \sim \mathbb{S}^d$
5      play $\hat{\boldsymbol{x}}^{(t)} \leftarrow A^{(t)} \boldsymbol{z}^{(t)}$
6      receive $\tilde{\mu}^{(t)} \leftarrow \mu^{(t)}(\hat{\boldsymbol{x}}^{(t)}) + c_t(\hat{\boldsymbol{x}}^{(t)})$
7      set $\tilde{\boldsymbol{v}}^{(t)} \leftarrow \hat{\boldsymbol{v}}^{(t)} + dc_t(\hat{\boldsymbol{x}}^{(t)})(A^{(t)})^{-1} \boldsymbol{z}^{(t)}$, $\hat{v}^t = n\mu^{(t)}(\hat{\boldsymbol{x}}^{(t)})(A^{(t)})^{-1} \boldsymbol{z}^{(t)}$
8      update $\boldsymbol{x}^{(t+1)} \leftarrow \mathcal{P}_{\mathcal{R}}(\boldsymbol{x}^{(t)}, \tilde{\boldsymbol{v}}^{(t)}, \eta_t)$

---

**Remark 3.** *Algorithm 2 is the single agent version of Algorithm 1. The proof Lemma 5 is similar to the proof in Appendix A of the paperBa et al. (2024), with a simple modification for $\lambda(\boldsymbol{x}^{(t)}, g) = \eta_t \|\tilde{\boldsymbol{v}}^{(t)}\|_{\boldsymbol{x}^{(t)}, *} = d\eta_t |\tilde{\mu}^{(t)}(\hat{\boldsymbol{x}}^{(t)})| \|\boldsymbol{z}^{(t)}\|_2 \leq d\eta_t \leq \frac{1}{2}$, which still holds.*

**Lemma 6** (Extended Lemma 3.10 in Ba et al. (2024))**.** *Suppose that the iterate $\{\boldsymbol{x}^{(t)}\}_{t \geq 1}$ is generated by Algorithm 1 and let each corrupted utility value $\tilde{\mu}_i^t$ satisfy that $|\tilde{\mu}_i^{(t)}(\boldsymbol{x})| \leq 1$*

*for all $\boldsymbol{x} \in \mathcal{X}$ and $0 < \eta_t \leq \frac{1}{2d}$, we have*

$$\sum_{i \in [n]} \lambda_i D_{\mathcal{R}}(p, \boldsymbol{x}_i^{(t+1)}) + \frac{\eta_t \beta(t+1)}{2}\left(\sum_{i \in \mathcal{N}} \|\boldsymbol{x}_i^{(t+1)} - p_i\|^2\right) \leq \sum_{i \in [n]} \lambda_i D_{\mathcal{R}}(p_i, \boldsymbol{x}_i^{(t)})$$

$$+ \frac{\eta_t \beta(t+1)}{2}\left(\sum_{i \in [n]} \|\boldsymbol{x}_i^{(t)} - p_i\|^2\right) + 2\eta_t^2\left(\sum_{i \in [n]} \lambda_i \|A_i^{(t)} \tilde{\boldsymbol{v}}_i^{(t)}\|^2\right) + \eta_t\left(\sum_{i \in [n]} \lambda_i \langle \tilde{\boldsymbol{v}}_i^{(t)}, \boldsymbol{x}_i^{(t)} - p_i \rangle\right),$$

*where $p_i \in \mathcal{X}_i$ and the sequence $\{\eta_t\}_{t \geq 1}$ is assumed to be non-increasing.*

Now, we will prove Theorem 2. Taking the expectation of both sides for equation proved in Lemma 6 conditioned on $\boldsymbol{x}^{(t)}$, we have

$$\mathbb{E}\left[\sum_{i \in [n]} \lambda_i D_{\mathcal{R}}(p, \boldsymbol{x}_i^{(t+1)}) | \boldsymbol{x}_t\right] + \frac{\eta_t \beta(t+1)}{2} \mathbb{E}\left[\sum_{i \in [n]} \|\boldsymbol{x}_i^{(t+1)} - p_i\|^2 | \boldsymbol{x}_t\right] \leq$$

$$\sum_{i \in [n]} \lambda_i D_{\mathcal{R}}(p_i, \boldsymbol{x}_i^{(t)}) + \frac{\eta_t \beta(t+1)}{2}\left(\sum_{i \in [n]} \|\boldsymbol{x}_i^{(t)} - p_i\|^2\right)$$

$$+ 2\eta_t^2 \mathbb{E}\left(\sum_{i \in [n]} \lambda_i \|A_i^{(t)} \tilde{\boldsymbol{v}}_i^{(t)}\|^2 | \boldsymbol{x}_t\right) + \eta_t \mathbb{E}\left(\sum_{i \in [n]} \lambda_i \langle \tilde{\boldsymbol{v}}_i^{(t)}, \boldsymbol{x}_i^{(t)} - p_i \rangle | \boldsymbol{x}_t\right).$$

By the definition of $\tilde{\boldsymbol{v}}_i^{(t)}$ and using Lemma 3.5 in Ba et al. (2024), we have

$$\mathbb{E}(\|A_i^{(t)} \hat{\boldsymbol{v}}_i^{(t)}\|^2 | \boldsymbol{x}_t) \leq d_i^2.$$

In addition, using Lemma 3.5 in Ba et al. (2024) again and the Young's inequality, we have

$$\mathbb{E}\left(\langle \hat{\boldsymbol{v}}_i^{(t)}, \boldsymbol{x}_i^{(t)} - p_i \rangle | \boldsymbol{x}_t\right) \leq \langle \nabla_i \mu_i(\boldsymbol{x}_t), \boldsymbol{x}_i^{(t)} - p_i \rangle + \frac{\lambda_i l_i^2}{2\beta}\left(\sum_{i \in [n]} (\sigma_{\max}(A_i^{(t)}))^2\right) + \frac{\beta}{2\lambda_i}\|\boldsymbol{x}_i^{(t)} - p_i\|^2 + \boldsymbol{b}_i^{(t)\top}(\boldsymbol{x}_i^{(t)} - p_i),$$

where we have $\boldsymbol{b}_i^{(t)}$ equals the following $\boldsymbol{b}_i^{(t)} = n_i \mathbb{E}\left[c_{t,i}(\hat{\boldsymbol{x}}^{(t)})(A_i^{(t)})^{-1} \boldsymbol{z}_i^{(t)} | \boldsymbol{x}_t\right]$. $\boldsymbol{b}_i^{(t)} \neq 0$, this is because $c_{t,i}(\hat{\boldsymbol{x}})$ is a function of $\boldsymbol{z}_i^{(t)}$. Since $\nabla^2 \mathcal{R}_i(\boldsymbol{x})$ is positive definite for all $\boldsymbol{x} \in \mathcal{X}$ and

$$\mathbb{E}\left[\sum_{i \in [n]} \lambda_i \langle \tilde{\boldsymbol{v}}_i^{(t)}, \boldsymbol{x}_i^{(t)} - p_i \rangle | \boldsymbol{x}_t\right] \leq \sum_{i \in [n]} \lambda_i \langle \nabla_i \mu_i(\boldsymbol{x}^{(t)}), \boldsymbol{x}_i^{(t)} - p_i \rangle + \frac{1}{2\eta_t \beta^2(t+1)}\left(\sum_{i \in [n]} \lambda_i\right)\left(\sum_{i \in [n]} \lambda_i^2 l_i^2\right)$$

$$+ \frac{\beta}{2}\left(\|\boldsymbol{x}_i^{(t)} - p_i\|^2\right) + \sum_{i \in [n]} \lambda_i \boldsymbol{b}_i^{(t)\top}(\boldsymbol{x}_i^{(t)} - p_i)$$

$$\leq \frac{1}{2\eta_t \beta^2(t+1)}\left(\sum_{i \in [n]} \lambda_i\right)\left(\sum_{i \in [n]} \lambda_i^2 l_i^2\right) - \frac{\beta}{2}\left(\|\boldsymbol{x}_i^{(t)} - p_i\|^2\right) + \sum_{i \in [n]} \lambda_i \boldsymbol{b}_i^{(t)\top}(\boldsymbol{x}_i^{(t)} - p_i) + \sum_{i \in [n]} \lambda_i \langle \nabla_i \mu_i(p_i), \boldsymbol{x}_i^{(t)} - p_i \rangle.$$

In this way, we have

$$\mathbb{E}\left[\sum_{i \in [n]} \lambda_i D_{\mathcal{R}_i}(p_i, \boldsymbol{x}_i^{(t+1)})\right] + \frac{\eta_{t+1}\beta(t+1)}{2}\mathbb{E}\left[\sum_{i \in [n]} \|\boldsymbol{x}_i^{(t+1)} - p_i\|^2\right] \leq \eta_t \mathbb{E}\left[\sum_{i \in [n]} \lambda_i \langle \nabla_i u_i(p), \boldsymbol{x}_i^{(t)} - p_i \rangle\right]$$

$$+ \mathbb{E}\left[\sum_{i \in [n]} \lambda_i D_{\mathcal{R}_i}(p_i, \boldsymbol{x}_i^{(t)})\right] + \frac{\eta_t \beta t}{2}\mathbb{E}\left[\sum_{i \in [n]} \|\boldsymbol{x}_i^{(t)} - p_i\|^2\right] + 2\eta_t^2\left(\sum_{i \in [n]} d_i^2 \lambda_i\right)$$

$$+ \frac{1}{2\beta^2(t+1)}\left(\sum_{i \in [n]} \lambda_i\right)\left(\sum_{i \in [n]} \lambda_i^2 \ell_i^2\right) + \eta_t \mathbb{E}\left[\sum_{i \in [n]} \lambda_i \boldsymbol{b}_i^{(t)\top}(\boldsymbol{x}_i^{(t)} - p_i)\right].$$

Summing up the above inequality over $t = 1, 2, \ldots, T-1$, we have

$$
\mathbb{E}\left[\sum_{i\in[n]}\lambda_i D_{\mathcal{R}_i}(p_i, \boldsymbol{x}_i^{(t)})\right] + \frac{\eta_T\beta T}{2}\mathbb{E}\left[\sum_{i\in[n]}\|\boldsymbol{x}_i^{(t)} - p_i\|^2\right] \leq \sum_{t=1}^{T-1}\eta_t\mathbb{E}\left[\sum_{i\in[n]}\lambda_i\langle\nabla_i u_i(p), \boldsymbol{x}_i^{(t)} - p_i\rangle\right]
$$

$$
+ \sum_{i\in[n]}\lambda_i D_{\mathcal{R}_i}(p_i, \boldsymbol{x}_i^{(1)}) + \frac{\eta_1\beta}{2}\left(\sum_{i\in N}\|\boldsymbol{x}_i^{(1)} - p_i\|^2\right) + 2\left(\sum_{i\in[n]}d_i^2\lambda_i\right)\left(\sum_{t=1}^{T-1}\eta_t^2\right)
$$

$$
+ \frac{1}{2\beta^2}\left(\sum_{i\in[n]}\lambda_i\right)\left(\sum_{i\in[n]}\lambda_i^2\ell_i^2\right)\left(\sum_{t=1}^{T-1}\frac{1}{t+1}\right) + \sum_{t=1}^{T-1}\eta_t\mathbb{E}\left[\sum_{i\in[n]}\lambda_i\boldsymbol{b}_i^{(t)\top}(\boldsymbol{x}_i^{(t)} - p_i)\right]
$$

We should notice that for each round, there is freedom for the adversary to choose who to attack. To be consistent with the lower bound, in this upper bound, we assume that the adversary consistently choose $i := \tilde{i}$ to attack. Before proceeding afterwards, now let's analyze the cumulative impact of bias on the convergence of action profile. Notice that for $i \neq \tilde{i}, \boldsymbol{b}_i^{(t)} = 0$. Therefore we can use Cauchy Schwartz to bound the following inequality

$$
\sum_{t=1}^{T-1}\eta_t\mathbb{E}\left[\sum_{i\in[n]}\lambda_i\boldsymbol{b}_i^{(t)\top}(\boldsymbol{x}_i^{(t)} - p_i)\right] \leq \tilde{C}'\sqrt{\beta}\sum_{t=1}^{T-1}\eta_t^{3/2}\sqrt{t+1}|c_{t,\tilde{i}}(\hat{\boldsymbol{x}}_t)|, \tilde{C}' = \tilde{C}dB\max_{i\in[n]}\sqrt{\lambda_i},
$$

where $B = \max_{i\in[n]}B_i$, $B_i$ is the radius of $\mathcal{X}_i$. In particular, $\tilde{C}$ is the constant such that for all $t$, $\sigma_{\max}\left(\nabla^2\mathcal{R}_{\tilde{i}}(\boldsymbol{x}_{\tilde{i}}^{(t)}) + \frac{\eta_t\beta(t+1)}{\lambda_{\tilde{i}}}I_{d_{\tilde{i}}}\right) \leq \tilde{C}\frac{\eta_t\beta(t+1)}{\lambda_{\tilde{i}}}$. Therefore, we have

$$
\mathbb{E}\left[\sum_{i\in[n]}\lambda_i D_{\mathcal{R}_i}(p_i, \boldsymbol{x}_i^{(t)})\right] + \frac{\eta_T\beta T}{2}\mathbb{E}\left[\sum_{i\in[n]}\|\boldsymbol{x}_i^{(t)} - p_i\|^2\right] \leq \sum_{t=1}^{T-1}\eta_t\mathbb{E}\left[\sum_{i\in[n]}\lambda_i\langle\nabla_i u_i(p), \boldsymbol{x}_i^{(t)} - p_i\rangle\right]
$$

$$
+ \sum_{i\in[n]}\lambda_i D_{\mathcal{R}_i}(p_i, \boldsymbol{x}_i^{(1)}) + \frac{\eta_1\beta}{2}\left(\sum_{i\in[n]}\|\boldsymbol{x}_i^{(1)} - p_i\|^2\right) + 2\left(\sum_{i\in[n]}d_i^2\lambda_i\right)\left(\sum_{t=1}^{T-1}\eta_t^2\right)
$$

$$
+ \frac{1}{2\beta^2}\left(\sum_{i\in[n]}\lambda_i\right)\left(\sum_{i\in[n]}\lambda_i^2\ell_i^2\right)\left(\sum_{t=1}^{T-1}\frac{1}{t+1}\right) + \tilde{C}'\sqrt{\beta}\sum_{t=1}^{T-1}\eta_t^{3/2}\sqrt{t+1}|c_{t,\tilde{i}}(\hat{\boldsymbol{x}}_t)|.
$$

Since $\mathbb{E}\left[\sum_{i\in[n]}\lambda_i D_{\mathcal{R}_i}(p_i, \boldsymbol{x}_i^{(t)})\right] > 0$, we have the following equation

$$
\mathbb{E}\left[\sum_{i\in[n]}\|\boldsymbol{x}_i^{(t)} - p_i\|^2\right] \leq \frac{2}{\eta_T\beta T}\sum_{t=1}^{T-1}\eta_t\mathbb{E}\left[\sum_{i\in[n]}\lambda_i\langle\nabla_i u_i(p), \boldsymbol{x}_i^{(t)} - p_i\rangle\right]
$$

$$
+ \frac{2}{\eta_T\beta T}\sum_{i\in[n]}\lambda_i D_{\mathcal{R}_i}(p_i, \boldsymbol{x}_i^{(1)}) + \frac{\eta_1}{\eta_T T}\left(\sum_{i\in[n]}\|\boldsymbol{x}_i^{(1)} - p_i\|^2\right) + \frac{4}{\eta_T\beta T}\left(\sum_{i\in[n]}d_i^2\lambda_i\right)\left(\sum_{t=1}^{T-1}\eta_t^2\right)
$$

$$
+ \frac{2}{\eta_T\beta^3 T}\left(\sum_{i\in[n]}\lambda_i\right)\left(\sum_{i\in[n]}\lambda_i^2\ell_i^2\right)\left(\sum_{t=1}^{T-1}\frac{1}{t+1}\right) + \frac{2}{\eta_T\sqrt{\beta}T}\tilde{C}'\sum_{t=1}^{T-1}\eta_t^{3/2}\sqrt{t+1}|c_{t,\tilde{i}}(\hat{\boldsymbol{x}}_t)|.
$$

$$(57)$$

By the initialization $\boldsymbol{x}_i^{(1)} = \arg\min_{x\in\mathcal{X}}\mathcal{R}_i(x)$, we have $\nabla\mathcal{R}_i(\boldsymbol{x}_i^{(1)}) = 0$ which implies that $D_{\mathcal{R}_i}(p_i, \boldsymbol{x}_i^{(1)}) = \mathcal{R}_i(p_i) - \mathcal{R}(\boldsymbol{x}_i^{(1)})$. Then, let us inspect each coordinate of a unique Nash equilibrium $x^*$ and set $p$ coordinate by coordinate in 57; indeed, we consider the following two cases:

- A point $\boldsymbol{x}_i^* \in \mathcal{X}_i$ satisfies that $\pi_{x^1}(\boldsymbol{x}_i^*) \leq 1 - \frac{1}{\sqrt{T}}$. By Lemma 2.4 (Ba et al., 2024), we have $D_{\mathcal{R}_i}(\boldsymbol{x}_i^*, \boldsymbol{x}_i^{(1)}) = \mathcal{R}_i(\boldsymbol{x}_i^*) - \mathcal{R}_i(\boldsymbol{x}_i^{(1)}) \leq \nu_i\log(T)$. In this case, we set $p_i = \boldsymbol{x}_i^*$.

- A point $\boldsymbol{x}_i^* \in \mathcal{X}_i$ satisfies that $\pi_{x^1}(\boldsymbol{x}_i^*) > 1 - \frac{1}{\sqrt{T}}$. Thus, we can find $\bar{\boldsymbol{x}}_i \in \mathcal{X}_i$ such that $\|\bar{\boldsymbol{x}}_i - \boldsymbol{x}_i^*\| = O(\frac{1}{\sqrt{T}})$ and $\pi_{x^1}(\bar{\boldsymbol{x}}_i) \leq 1 - \frac{1}{\sqrt{T}}$. By Lemma 2.4 (Ba et al., 2024), we have $D_{\mathcal{R}_i}(\bar{\boldsymbol{x}}_i, \boldsymbol{x}_i^{(1)}) = \mathcal{R}_i(\bar{\boldsymbol{x}}_i) - \mathcal{R}_i(\boldsymbol{x}_i^{(1)}) \leq \nu_i \log(T)$. In this case, we set $p_i = \bar{\boldsymbol{x}}_i$.

Therefore, we have the following equation. By choosing $\eta_t = \frac{1}{2d}t^{-\phi}$, we have

$$\mathbb{E}\left[\sum_{i \in [n]} \|\boldsymbol{x}_i^{(t)} - \boldsymbol{x}_i^*\|^2\right] \leq \mathbb{E}\left[\sum_{i \in [n] \setminus \mathcal{I}} \|\boldsymbol{x}_i^{(t)} - \boldsymbol{x}_i^*\| \|\boldsymbol{x}_i^* - \bar{\boldsymbol{x}}_i\|\right] \tag{58}$$

$$+ \frac{4d}{\beta T^{1-\phi}} \sum_{t=1}^{T-1} \eta_t \left(\mathbb{E}\left[\left(\sum_{i \in [n]} \lambda_i \ell_i \|\boldsymbol{x}_i^{(t)} - \boldsymbol{x}_i^*\|\right)\left(\sum_{i \in [n] \setminus \mathcal{I}} \|\boldsymbol{x}_i^* - \bar{\boldsymbol{x}}_i\|\right) + \sum_{i \in [n]} \lambda_i \|\nabla_i u_i(\boldsymbol{x}_{\mathcal{I}}^{(t)}, \bar{\boldsymbol{x}}_{\mathcal{N} \setminus \mathcal{I}})\| \|\boldsymbol{x}_i^* - \bar{\boldsymbol{x}}_i\|\right]\right) \tag{59}$$

$$+ \frac{1}{T^{1-\phi}} \left(\sum_{i \in [n]} B_i^2\right) \tag{60}$$

$$+ \frac{2}{d\beta T^{1-\phi}} \left(\sum_{i \in [n]} d_i^2 \lambda_i\right) \sum_{t=1}^{T-1} t^{-2\phi} \tag{61}$$

$$+ \frac{4d \log T}{\beta T^{1-\phi}} \left(\sum_{i \in [n]} \lambda_i\right) \left(\sum_{i \in [n]} \lambda_i^2 \ell_i^2\right) \tag{62}$$

$$+ \frac{2\tilde{C}'}{d\sqrt{\beta} T^{1-\phi}} \sum_{t=1}^{T-1} t^{\frac{1}{2} - \frac{3}{2}\phi} |c_{t,\tilde{i}}(\hat{\boldsymbol{x}}_t)| \tag{63}$$

$$+ \frac{2\tilde{C}'}{d\sqrt{\beta} T^{1-\phi}} \sum_{t=1}^{T-1} t^{-\frac{3}{2}\phi} |c_{t,\tilde{i}}(\hat{\boldsymbol{x}}_t)| \tag{64}$$

$$+ \frac{4d}{\beta T^{1-\phi}} \sum_{i \in [n]} \nu_i \lambda_i \log T \tag{65}$$

We will analyze the convergence of $\mathbb{E}\left[\sum_{i \in [n]} \|\boldsymbol{x}_i^{(t)} - \boldsymbol{x}_i^*\|^2\right]$ by bounding the convergence of the right-hand size one by one. By Lemma 2.4 (Ba et al., 2024), we have Eq. (58) and Eq. (59) are upper bounded by $O\left(\frac{d}{\sqrt{T}}\right)$. Eq. (60) is upper bounded by $O(\frac{d}{T^{1-\phi}})$. Eq. (61) is upper bounded by $O(\frac{d}{T^\phi})$. Eq. (62) and Eq. (65) are upper bounded by $\tilde{O}(\frac{d}{T^{1-\phi}})$. Now it remains to analyze the convergence rate for Eq. (63), which we regard as the impact of observation error. Note that Eq. (65) can never be the dominant term, therefore, we drop it from the analysis. Since $t^{-\phi} - (t+1)^{-\phi} \leq t^{-(\phi+1)}$ for integer $t$ and $\phi > 0$ and if $\sum_{j=1}^t |c_{j,\tilde{i}}(\hat{\boldsymbol{x}}_j)| \leq t^\rho$ holds for all $t$, then we can bound

$$\frac{2\tilde{C}'}{d\sqrt{\beta} T^{1-\phi}} \sum_{t=1}^{T-1} t^{\frac{1}{2} - \frac{3}{2}\phi} |c_{\tilde{i}}^t(\hat{\boldsymbol{x}}^{(t)})| \leq O(T^{\rho - \frac{1}{2}(\phi+1)}).$$

Noticing that

$$\mathbb{E}\left[\sum_{i \in [n]} \|\hat{\boldsymbol{x}}_i^{(t)} - \boldsymbol{x}_i^{(t)}\|^2\right] \leq \mathbb{E}\left[\sum_{i \in [n]} (\sigma_{\max}(A_i^{(t)}))^2\right] \leq \frac{2d \sum_{i \in [n]} \lambda_i}{\beta T^{1-\phi}}$$

We have the following

$$\mathbb{E}\left[\sum_{i \in [n]} \|\hat{\boldsymbol{x}}_i^{(t)} - \boldsymbol{x}_i^*\|^2\right] \leq \max\left\{O(dT^{\phi-1}), O(dT^{-\phi}), O\left(T^{\rho - \frac{1}{2}(\phi+1)}\right)\right\},$$

which completes the proof. $\square$

### C.3 Essential Details for MAMD (Bravo et al., 2018)

At the beginning of this section, we provide necessary details for Muti-Agent Mirror Descent (MAMD), initially developed by Bravo et al. (2018) for multi-agent learning (see Algorithm 3). We attempt to evaluate the robustness of Algorithm 3 under utility shifting attack, highlighted by the red line (Line 7 of Algorithm 3).

---

**Algorithm 3:** MAMD (Bravo et al., 2018) under Utility Shifting Attack

---

**Input:** step-size $\eta_{t,1} = \gamma t^{-3\phi}$, query radius $\eta_{t,2} = \delta t^{-\phi}$, safety ball $\mathbb{B}_{r_i}(p_i) \subset \mathcal{X}_i$ for all agents

1 Initialize action $\boldsymbol{x}_i^{(0)} \in \mathcal{X}_i$ for each agent $i \in [n]$
2 **for** $t \in [T]$ **do**
3      **for** $i \in [n]$ **do**
4          Draw direction $\boldsymbol{z}_i^{(t)}$ uniformly from $\mathbb{S}^{d_i}$
5          Set $\boldsymbol{w}_i^{(t)} \leftarrow \boldsymbol{z}_i^{(t)} - r_i^{-1}(\boldsymbol{x}_i^{(t)} - p_i)$
6          play $\hat{\boldsymbol{x}}_i^{(t)} \leftarrow \boldsymbol{x}_i^{(t)} + \eta_{t,2}\boldsymbol{w}_i^{(t)}$
7      Receive $\tilde{\mu}_{\tilde{i}}^{(t)} \leftarrow \mu_{\tilde{i}}(\hat{\boldsymbol{x}}_t) + c_{t,\tilde{i}}(\hat{\boldsymbol{x}}_t)$ for $i \in \tilde{\mathcal{I}}_t$, $\tilde{\mu}_i^{(t)} \leftarrow \mu_i(\hat{\boldsymbol{x}}_t)$ for $i \in [n] \setminus \tilde{\mathcal{I}}_t$
8      **for** $i \in [n]$ **do**
9          Set $\hat{\boldsymbol{v}}_i^{(t)} \leftarrow (d_i/\eta_{t,2})\tilde{\mu}_i^{(t)}\boldsymbol{z}_i^{(t)}$
10          Update $\boldsymbol{x}_i^{(t+1)} \leftarrow \mathcal{P}_{\boldsymbol{x}_i^{(t)}}\left(\eta_{t,1}\hat{\boldsymbol{v}}_i^{(t)}\right)$ [a]

---

[a] The proximal mapping $\mathcal{P}_X(y) := \arg_{x' \in \mathcal{X}} \min\{y^\top(x - x') + D(x', x)\}$, where $D_\mathcal{R}(x', x)$ is the Bregman divergence, where $D_\mathcal{R}(x', x) = \mathcal{R}(x') - \mathcal{R}(x) - \nabla\mathcal{R}(x)^\top(x' - x)$, for some strongly convex function $\mathcal{R}$. Additionally, Algorithm 3 creates a safety ball $\mathbb{B}_{r_i}(p_i)$, with $\delta/r_i < 1$, and generate $\hat{\boldsymbol{x}}_i^{(t)}$ bases on the adjusted randomly sampled direction $\boldsymbol{z}_i^{(t)}$ to ensure the feasibility.

### C.4 Proof of Proposition 4

**Proposition 4.** *Consider an adversary that attacks a set of agents $\tilde{\mathcal{I}}_t$ at round $t$ with a total budget satisfying $\sum_{j=1}^t \sum_{\tilde{i} \in \tilde{\mathcal{I}}_j} |c_{j,\tilde{i}}| \leq \mathcal{O}(t^\rho)$ for all $t$ and some $\rho \in [0, 1]$. By choosing parameters $\eta_{t,1} := \gamma t^{-3\phi}$ and $\eta_{t,2} := \delta t^{-\phi}$, for constants $\gamma > \frac{1}{3\beta}$ and $\delta > 0$, $\frac{1}{4} < \phi \leq \frac{1}{3}$, running Algorithm 3 a sufficiently large $T$, the last-iterate convergence rate satisfies: $\mathbb{E}\left(\|\hat{\boldsymbol{x}}_T - \boldsymbol{x}^*\|^2\right) \leq \mathcal{O}(d^2 \max\{T^{-\phi}, T^{\rho-2\phi}\})$.*

*Proof.* Similar to the proof structure in Appendix C.2, at core of the proof of Proposition 4 is to quantify the bias, $\boldsymbol{b}_i$, caused by utility shifting attack in gradient estimation for a SPSA estimator developed by Bravo et al. (2018), resulting the following lemma.

**Lemma 7** (Extended Lemma C.1 in (Bravo et al., 2018)). *Consider $\hat{\boldsymbol{v}}_i^{(t)}$ (defined in Line 9 of Algorithm 3) and Let $\hat{\mu}_i(\boldsymbol{x}) := \mathbb{E}_{w_i \sim \mathbb{B}^{d_i}}\mathbb{E}_{\boldsymbol{z}_{-i} \sim \prod_{j \neq i} \mathbb{S}^{d_j}}[\mu_i(\boldsymbol{x}_i + \eta_{t,2}w_i; \hat{\boldsymbol{x}}_{-i})]$, then we have*

$$\mathbb{E}(\hat{\boldsymbol{v}}_i|\boldsymbol{x}) := \nabla_i\hat{\mu}_i(\boldsymbol{x}) + \boldsymbol{b}_i,$$

*where $\boldsymbol{b}_i := \frac{d_i}{\eta_{t,2}}\mathbb{E}\left(c_i(\hat{\boldsymbol{x}})\boldsymbol{z}_i|\boldsymbol{x}\right)$.*

Using Lemma 7, by choosing $\eta_{t,1} = \gamma t^{-p}, \eta_{t,2} = \delta t^{-q}$, we can extend Equation D.13 (see Eq. (66)) in Bravo et al. (2018) to incorporate bias (see term $C$ in Eq. (67)), where we use $\bar{D}_t$ to denote the average Bergman divergence $\mathbb{E}(D_t)$ between $\boldsymbol{x}^*$ and $\boldsymbol{x}_t$ induced by some $K$-strongly convex function $\mathcal{R}$, and $V^2 = \sum_i d_i^2$, we have

$$\bar{D}_{t+1} \leq (1 - \frac{\beta\gamma}{t^p})\bar{D}_t + \frac{\gamma}{t^p}\left(\frac{\delta}{t^q}\right) + \frac{V^2}{2K}\frac{\gamma^2\delta^2}{t^{2(p-q)}}, \tag{66}$$

$$\bar{D}_{t+1} \leq \underbrace{(1 - \frac{\beta\gamma}{t^p})\bar{D}_t}_{A} + \underbrace{\frac{\gamma}{t^p}\left(\frac{\delta}{t^q}\right)}_{B} + \underbrace{\eta_{t,1}\left(\sum_{i\in[n]} \boldsymbol{b}_i^{(t)}\right)}_{C} + \underbrace{\frac{V^2}{2K}\frac{\gamma^2\delta^2}{t^{2(p-q)}}}_{D}. \tag{67}$$

To analyze the convergence rate of $\bar{D}_{t+1}$, we need further extend the convergence lemma (Lemma D.2 in Bravo et al. (2018)) from smooth decaying $b_t \sim \mathcal{O}(\frac{1}{t^q})$ to only a sublinear summation bound (see Eq. (68)) to incorporate utility shifting attack.

**Lemma 8** (Extended Lemma D.2 in (Bravo et al., 2018)). *Let $a_n$ be a non-negative sequence satisfying that*

$$a_{t+1} \leq a_t \left(1 - \frac{P}{t^p}\right) + \frac{Q}{t^p} \cdot b_t,$$

$$\sum_{t=1}^{T} b_t < T^\alpha, \forall T > 0. \tag{68}$$

*where $P, Q > 0, p \in (0,1], \alpha \in [\frac{1}{2}, 1), b_t \in [0,1]$. Then, there exists a constant $C > 0$ such that for sufficiently large $t$, we have*

1. *there exists positive constants $C, T_0$ such that for any $t > T_0$,*

$$a_t \leq \frac{C}{t^{p-\alpha}},$$

2. *if the sequence $\{b_t\}_{t=0}^{\infty}$ is non-increasing starting from $t = t_0$, there exists positive constants $C, T_0$ such that for any $t > T_0$,*

$$a_t \leq \frac{C}{t^{1-\alpha}}.$$

*Proof.* For the first situation, it holds that

$$\begin{aligned}
a_{t+1} &\leq a_t \left(1 - \frac{P}{t^p}\right) + \frac{Q}{t^p} \cdot b_t \\
&\leq \left(a_{t-1}\left(1 - \frac{P}{(t-1)^p}\right) + \frac{Q}{(t-1)^p} \cdot b_{t-1}\right)\left(1 - \frac{P}{t^p}\right) + \frac{Q}{t^p} \cdot b_t \\
&\leq \cdots \\
&= Q \cdot \sum_{k=0}^{t-1}\left[\frac{b_{t-k}}{(t-k)^p}\prod_{i=1}^{k}\left(1 - \frac{P}{(t-i+1)^p}\right)\right] \tag{69} \\
&\leq Q \cdot \sum_{k=0}^{t^\alpha}\left[\frac{1}{(t-k)^p}\prod_{i=1}^{k}\left(1 - \frac{P}{(t-i+1)^p}\right)\right] \tag{70} \\
&\leq Q \cdot \sum_{k=0}^{t^\alpha}\left[\frac{1}{(t-k)^p}\right] \\
&\leq Qt^{\alpha-p},
\end{aligned}$$

where Eq. (70) holds because the sequence $\left\{\frac{1}{(t-k)^p}\prod_{i=1}^{k}\left(1 - \frac{P}{(t-i+1)^p}\right)\right\}_{k=0}^{\infty}$ is non-increasing when $p \in (0,1]$. Therefore, the maximum possible value of the RHS of Eq. (69) is achieved when $b_t = \cdots = b_{t-t^\alpha+1} = 1$. When $\{b_t\}$ is non-increasing, the RHS of Eq. (69) achieves its maximum possible value when $b_1 = \cdots = b_t = t^{\alpha-1}$. Therefore, it holds that

$$a_{t+1} \le Q \cdot \sum_{k=0}^{t-1} \left[ \frac{b_{t-k}}{(t-k)^p} \prod_{i=1}^{k} \left( 1 - \frac{P}{(t-i+1)^p} \right) \right]$$

$$\le t^{\alpha-1} Q \cdot \sum_{k=0}^{t-1} \left[ \frac{1}{(t-k)^p} \prod_{i=1}^{k} \left( 1 - \frac{P}{(t-i+1)^p} \right) \right]$$

$$\le O(t^{\alpha-1}).$$

□

Armed with Lemma 8, we can analyze the convergence of Eq. (67). Firstly, we note that $\eta_{t,1} \left( \sum_{i \in [n]} \boldsymbol{b}_i^{(t)} \right) \le \mathcal{O}(dT^{\rho-q})$. Second, we need to choose $p, q > 0$ such that the cumulative sum with respect to the term $A, B, C, D$ converge to a constant. To satisfy this constraint, we need $\frac{1}{4} < q \le \frac{1}{3}$ and we choose $q$ to make the order of term $B$ and $D$ equal, we get $p = 3q$. In this way, the convergence rate is determined by $\mathcal{O}(d^2 \max\{t^{-q}, t^{\rho-2q}\})$, completing the proof. □

## D   ADDITIONAL DISCUSSION ABOUT ABSOLUTE ROBUSTNESS

In Section 4, we explored the efficiency-robustness trade-off based on a specific concept of robustness—whether a utility poisoning adversary can induce NSA. However, this is not the only measure of robustness. In this section, we introduce a new concept called absolute robustness and establish a formal efficiency-robustness trade-off, analogous to our results in Section 4, under this broader notion of robustness.

To introduce the notion of absolute robustness, we need to first define a generalized concept of utility poisoning attack outcome, named $\gamma$-Successful Attack ($\gamma$-SA), as shown in Definition 8. Intuitively, $\gamma$-SA depicts a broader and milder goal (i.e., slowing down the convergence rate or causing non-convergence) compared to NSA.

**Definition 8** ($\gamma$-Successful Attack and Absolute Robustness). *We say an adversary successfully implements a $\gamma$-Successful Attack ($\gamma$-SA) for some $\gamma \in [0, 1]$ to an $(\alpha, p)$-MAL dynamics running a game if it induces a joint strategy sequence $\{\boldsymbol{x}^{(t)}\}_t^T$ such that for any sufficiently large $T$, the cumulative error*

$$\sum_{t=1}^{T} \mathbb{E}[\|\boldsymbol{x}^{(t)} - \boldsymbol{x}^*\|_2^p] = \Omega\left( T^{1-p\alpha(1-\gamma)} \right). \tag{71}$$

*In addition, we say an MAL dynamics $\mathcal{A}$ is absolutely robust to a utility poisoning adversary $\mathcal{C}$ with budget $\mathcal{O}(T^\rho)$ if $\mathcal{C}$ with budget $\mathcal{O}(T^\rho)$ cannot induce $\gamma$-SA for any $\gamma > 0$.*

Naturally, NSA is a form of 1-SA, as shifting the NE inevitably results in a linearly growing cumulative error. Therefore, $\gamma$-SA is a weaker concept in terms of the attack's impact, while absolute robustness represents a stronger defensive notion, since it implies that not only can a new convergence point not be imposed, but the original convergence rate remains unaffected. Our next result shown in Corollary 2 indicates that if an adversary aims to achieve $\gamma$-SA for some $\gamma \in (0, 1)$, it requires a smaller total budget, as expected. To accomplish this, the adversary can divide the time horizon into different phases, alternating between performing SUSA and doing nothing. By adjusting the lengths of these phases, the adversary can balance the attack budget and the desired level of regret.

**Corollary 2.** *Under the same conditions as stated in Theorem 1, an adversary can implement a $\gamma$-SA within a total budget*

$$\mathbb{E}\left[ \sum_{t=1}^{T} |c_t| \right] = \mathcal{O}\left( T^{\left(1 - \frac{p\alpha}{p+1}\right)(1 - p\alpha(1-\gamma))} \right). \tag{72}$$

*Proof.* Define the $p$-th moment regret $Reg_p(T) = \sum_{t=1}^{T} \mathbb{E}[\|\boldsymbol{x}_t - \boldsymbol{x}^*\|_2^p]$. Then from Theorem 1, we know that for any sufficiently large $T_0$, the adversary can use an expected total budget of $\mathcal{O}\left(T_0^{1-\frac{p\alpha}{p+1}}\right)$ to incur a regret $\Omega(T_0)$. Consequently, for any given $T$, setting $T_0 = T^{1-p\alpha(1-\gamma)}$, the adversary can use $\mathcal{O}\left(T^{(1-\frac{p\alpha}{p+1})(1-p\alpha(1-\gamma))}\right)$ total budget in expectation to incur a regret of the order $\Omega(T^{1-p\alpha(1-\gamma)})$ by performing SUSA for the initial $T_0$ rounds. For the remaining rounds, the adversary can refrain from injecting corruptions and allow the agents to run their MAL algorithm without interference. In this way, the adversary can incur at least $\Omega(T^{1-p\alpha(1-\gamma)})$ regret while using at most $\mathcal{O}\left(T^{(1-\frac{p\alpha}{p+1})(1-p\alpha(1-\gamma))}\right)$ total expected budget. $\square$

Theorem 1 and Corollary 2 together highlight an intriguing trade-off between the efficiency and robustness of MAL learning dynamics. That is, the faster an MAL algorithm converges (i.e., a larger $\alpha$), the smaller the total corruption budget required to implement a successful attack (either NSA or $\gamma$-SA). Next, we establish an analogous result to Corollary 1 under a new (and stronger) robustness concept, that is, whether an MAL dynamics is absolutely robust to a utility poisoning adversary with certain budget, as defined below:

$$\bar{\rho}(\mathcal{C}, \mathcal{A}) = \inf\{\rho \in [0,1]|\mathcal{C} \text{ with budget } \mathcal{O}(T^\rho) \text{ can achieve } \gamma\text{-SA against } \mathcal{A}.\},$$
$$= \sup\{\rho \in [0,1]|\mathcal{A} \text{ is } \textit{absolutely robust} \text{ to } \mathcal{C} \text{ with budget } \mathcal{O}(T^\rho).\},$$

where $\mathcal{C} \in ADV$ represents a generic utility poisoning adversary. Similarly, we fix $p = 2$ and define the following quantities:

$$\bar{\rho}_-(\alpha) = \inf_{\mathcal{C} \in ADV} \sup_{\mathcal{A} \in MAL(\alpha, 2)} \rho(\mathcal{C}, \mathcal{A}), \quad \bar{\rho}_+(\alpha) = \sup_{\mathcal{A} \in MAL(\alpha, 2)} \inf_{\mathcal{C} \in ADV} \rho(\mathcal{C}, \mathcal{A}). \tag{73}$$

Here, the function $\bar{\rho}_-(\alpha)$ defined in Eq. (73) represents the optimal strategy of a utility poisoning adversary. Specifically, for any MAL algorithm known to enjoy convergence rate $\alpha$, $\bar{\rho}_-(\alpha)$ denotes the minimum budget level $\rho$ required for a successful $\gamma$-SA by the adversary. In contrast, the function $\bar{\rho}_+(\alpha)$ characterizes the optimal defense from the perspective of the algorithm designer. Given a required convergence rate $\alpha$, $\bar{\rho}_+(\alpha)$ identifies the most robust algorithm that can withstand $\gamma$-SA of level $\rho$ from any utility poisoning adversary. From minimax theorem (v. Neumann, 1928; Sion, 1958) we still have $\bar{\rho}_-(\alpha) \geq \bar{\rho}_+(\alpha), \forall \alpha \in [0, \frac{1}{4}]$. Furthermore, Corollary 2 and Theorem 2 lead to the following result

**Corollary 3.** *Given the definitions of $\bar{\rho}_-(\alpha)$ and $\bar{\rho}_+(\alpha)$ in Eq. (73), for some particular MAL algorithm $\mathcal{A}_0$ and adversary $\mathcal{C}_0 \in ADV$, we further define*

$$\bar{\rho}(\mathcal{A}_0(\alpha)) = \inf_{\mathcal{C} \in ADV} \rho(\mathcal{C}, \mathcal{A} = \mathcal{A}_0(\alpha)), \quad \bar{\rho}(\alpha, \mathcal{C}_0) = \sup_{\mathcal{A} \in MAL(\alpha, 2)} \rho(\mathcal{C} = \mathcal{C}_0, \mathcal{A}),$$

*where $\mathcal{A}_0(\alpha)$ denotes algorithm $\mathcal{A}_0$ with a set of hyper-parameters that guarantees $MAL(\alpha, 2)$ convergence. Then for any $\alpha \in [0, \frac{1}{4}]$, it holds that*

$$1 - 3\alpha \leq \bar{\rho}(MD\text{-}SCB(\alpha)) \leq \bar{\rho}_+(\alpha) \leq \bar{\rho}_-(\alpha) \leq \bar{\rho}(\alpha; SUSA) \leq \left(1 - \frac{2\alpha}{3}\right)(1 - 2\alpha). \tag{74}$$

The proof of Corollary 3 follows exactly from the idea of Corollary 1, and thus we put them together in Appendix D.1.

Figure 3 illustrates this trade-off. The yellow dashed line represents $\bar{\rho}_-(\alpha)$; above it, a budget of $\mathcal{O}(T^\rho)$ suffices for a powerful adversary to induce $\gamma$-SA in any $MAL(\alpha, 2)$ dynamics. The red region, a subset of this area, corresponds to the actual budget required for $\gamma$-SA for the specific attacking strategy outlined in Corollary 2. Conversely, the blue dashed line represents $\bar{\rho}_+(\alpha)$; below it, the most robust $MAL(\alpha, 2)$ dynamics is absolutely robust to any adversary with a total corruption of $\mathcal{O}(T^\rho)$. For a specific MAL dynamics like MD-SCB, the green region shows the budget within which it remains absolutely robust. We only present $\rho \in [0.25, 1]$ because the MAL dynamics with the best achievable convergence rate in such a setting $(alpha = 0.25)$(Ba et al., 2024) can withstand a corruption level of $\rho = 0.25$ (see Theorem 2). The three gaps in Figure 3 correspond to three open problems similar to those we proposed in Figure 1.

## D.1 Proof of Corollary 1 and Corollary 3

*Proof.* By the definitions of $\rho_-(\alpha), \rho_+(\alpha), \bar{\rho}_-(\alpha), \bar{\rho}_+(\alpha)$ and minimax theorem, it holds that

$$\rho(\text{MD-SCB}(\alpha)) \leq \rho_+(\alpha) \leq \rho_-(\alpha) \leq \rho(\alpha; \text{SUSA}),$$
$$\bar{\rho}(\text{MD-SCB}(\alpha)) \leq \bar{\rho}_+(\alpha) \leq \bar{\rho}_-(\alpha) \leq \bar{\rho}(\alpha; \text{SUSA}).$$

Taking $p = 2$ in Theorem 1, we have $\rho(\alpha; \text{SUSA}) \leq 1 - \frac{2\alpha}{3}$, and taking $p = 2$ in Corollary 2, we have $\bar{\rho}(\alpha; \text{SUSA}) \leq \left(1 - \frac{2\alpha}{3}\right)(1 - 2\alpha)$.

Next, we show that $\rho(\text{MD-SCB}(\alpha)) \geq 1 - \alpha$ and $\bar{\rho}(\text{MD-SCB}(\alpha)) \geq 1 - 3\alpha$. By Theorem 2, we know that for MD-SCB, in order to incorporate a corruption $\mathcal{O}(T^\rho)$ with $\rho > 0$, we have to slow down the learning rate by increase $\phi$. Therefore, for any particular choice of $\phi$, the resultant convergence rate $\alpha = \frac{1-\phi}{2}$. Hence, only when $-\rho + \frac{\phi+1}{2} \geq 1 - \phi$, this choice does not affect the convergence rate, and we obtain $\rho \leq \frac{3}{2}\phi - \frac{1}{2} = 1 - 3\alpha$, meaning $\bar{\rho}(\text{MD-SCB}(\alpha)) \geq 1 - 3\alpha$. On the other hand, if $\rho \geq 1 - 3\alpha$, as long as $\rho - \frac{\phi+1}{2} < 0$ (i.e., $\rho < 1 - \alpha$), NSA is impossible to induce as MD-SCB will still converge although at a slower speed. This implies that $\rho(\text{MD-SCB}(\alpha)) \geq 1 - \alpha$. $\square$

Notably, the efficiency-robustness trade off also exists in Algorithm 3 for $\gamma$-SA. Eq. (67) suggest that as long as $1 - \rho - q > q$, $\frac{1}{t^{p+q}}$, will be the dominant order in deciding the convergence $\mathbb{E}(\|\boldsymbol{x}_T - \boldsymbol{x}^*\|_2^2)$. This implies that $\bar{\rho}(\text{MAMD}(\alpha)) \geq 1 - 4\alpha$, which also shows as long as $\rho < 1 - 4\alpha$, the convergence rate of Algorithm 3 will not be affected – robust to $\gamma$-SA. However, since $q$ is restricted to $\frac{1}{4} < q \leq \frac{1}{3}$ to ensure the convergence of $\bar{D}_t$, when $\rho > \frac{1}{2}$, the dominant order will be $t^{\rho-2q}$ and we can see optimal choice of $q = \frac{1}{3}$, implying $\rho(\text{MAMD}(\alpha)) \geq \frac{2}{3}$, which we do not observe such a efficiency-robustness trade-off of Algorithm 3 for SUSA, theoretically.

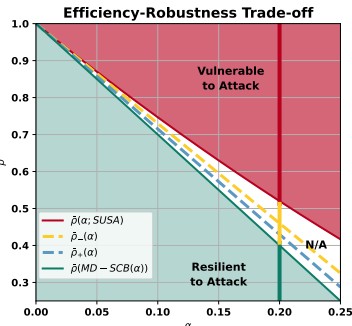

Figure 3: Illustration of the four quantities in Corollary 3, revealing the intrinsic efficiency-robustness trade-off under $\gamma$-SA.

## E Additional Experiments

### E.1 Experiment Result for MAMD

In this section, consider the same experimental setup as of Section 5 and evaluate the performance of Algorithm 3. We let each agent run MAMD with the game size $n \in \{10, 50, 100\}$. The learning rate schedule of MAMD is set to $\eta_{t,1} \propto t^{-3\phi}$, and $\eta_{t,2} \propto t^{-\phi}$, $\phi = 0.33, 0.30, 0.25$, corresponding to convergence rates $\alpha = 0.165, 0.15, 0.125, p = 2$ when no attack is present. We implement SUSA against agent 1, with a fixed $\delta = 10.0$. For each game instance specified by $n$ and the attacked algorithm specified by the convergence rate $\alpha$, we compare the dynamics' behavior both without attack and under SUSA, and report the actual total budget used.

**Result:** In the upper row of Figure 4, the left panel plots the convergence curve of the $L_2$ square error for $n = 10$, serving as a sanity check to confirm that different learning rate schedules induce different convergence rates for MAMD in absence of an adversary. The middle panel shows the outcome of SUSA against MAMD with $\alpha = 0.165$, with the $y$-axis representing $L_2$ distance between $\boldsymbol{x}_t$ and NE. The dashed lines display the convergence curve without the adversary, while the solid lines represent the dynamics under SUSA, with different colors indicating results for various game sizes $n$. As shown, for different values of $n$, the attacked dynamics exhibit divergence, as the solid lines saturate and stop decreasing, indicating that the dynamics are being steered to converge to a new point. However, the trend between induced NE deviation magnitude with respect to $n$ is unclear for MAMD, possibly because the lower bound proved in Eq. (10) of Theorem 1 is not tight. The right panel shows the cumulative attack budget $C_t = \sum_{\tau=1}^{t} |c_\tau|$ at each time step $t$ against MAMD

with different values of $\alpha$. As the results indicate, although all exhibit a sublinear trend, a faster-converging dynamics ($\alpha = 0.165$) requires a smaller attack budget, corroborating our findings in Theorem 1 and 2. The bottom row of Figure 4 evaluates the performance of MAMD under a different sets of parameters nevertheless conveys a similar message.

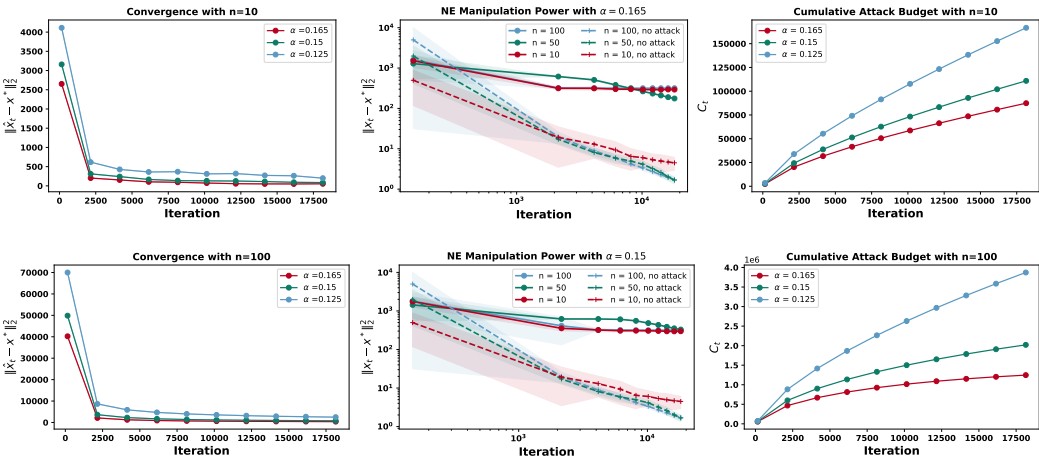

Figure 4: Left: square error of MAMD with varying convergence rates. Middle: square error of MAMD on different sizes of game instances against SUSA. Right: cumulative attack budget used by SUSA against MAMD with varying convergence rates. Error bars represent the 1-$\sigma$ region from 20 independent simulations.

## E.2 Additional Results for MD-SCB

Figure 5 evaluates the performance of MD-SCB under a different sets of parameters nevertheless conveys a similar message as of Figure 6.

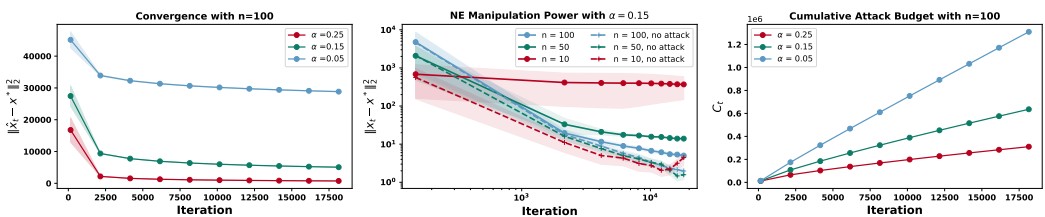

Figure 5: Left: square error of MD-SCB with varying convergence rates when $n = 100$. Middle: square error of MD-SCB on different sizes of game instances against SUSA with $\alpha = 0.15$. Right: cumulative attack budget used by SUSA against MD-SCB with varying convergence rates when $n = 100$. Error bars represent the 1-$\sigma$ region from 20 independent simulations.

## F Experiments

In this section, we validate our theoretical results from Theorem 1 and Theorem 2 by implementing SUSA against MD-SCB on $n$-person Cournot games, in which each player-$i$ has a utility function $u_i(x_i, x_{-i}) = x_i \left( a - b \sum_{j=1}^n x_j \right) - c_i x_i$. The formal definition is given in Appendix A. We also direct readers to Appendix E for additional results for attacks against another algorithm MAMD (see Algorithm 3). We run simulations on two types of Cournot game instances. In the first type (homogeneous), all players share the same marginal cost $c_i$, resulting in a symmetric game. In the second type (heterogeneous), players have varying costs. The results for the homogeneous case are presented in the main paper,

while those for the heterogeneous case are included in Appendix E.2, as they basically convey the same message.

**_Homogeneous_ $n$-person Cournot Game Experimental Setup:** We set $a = 10, b = 0.05$, and we consider the cost $c_i = 1$ for all agents $i \in [n]$. The action space is set to $\mathcal{X}_i = [0, 50]$ and the unique NE of such games can be verified as $\boldsymbol{x}_i^* = \frac{180}{n+1}, i \in [n]$. We let each agent run MD-SCB with the game size $n \in \{10, 50, 100\}$. The learning rate schedule of MD-SCB is set to $(\eta_t \propto t^{-\phi}, \phi = 0.5, 0.7, 0.9)$, corresponding to convergence rates $\alpha = 0.25, 0.15, 0.05, p = 2$ when no attack is present. We implement SUSA against agent 1, with a fixed $\delta = -10.0$. For each game instance specified by $n$ and the attacked algorithm specified by the convergence rate $\alpha$, we compare the dynamics' behavior both without attack and under SUSA, and report the actual total budget used.

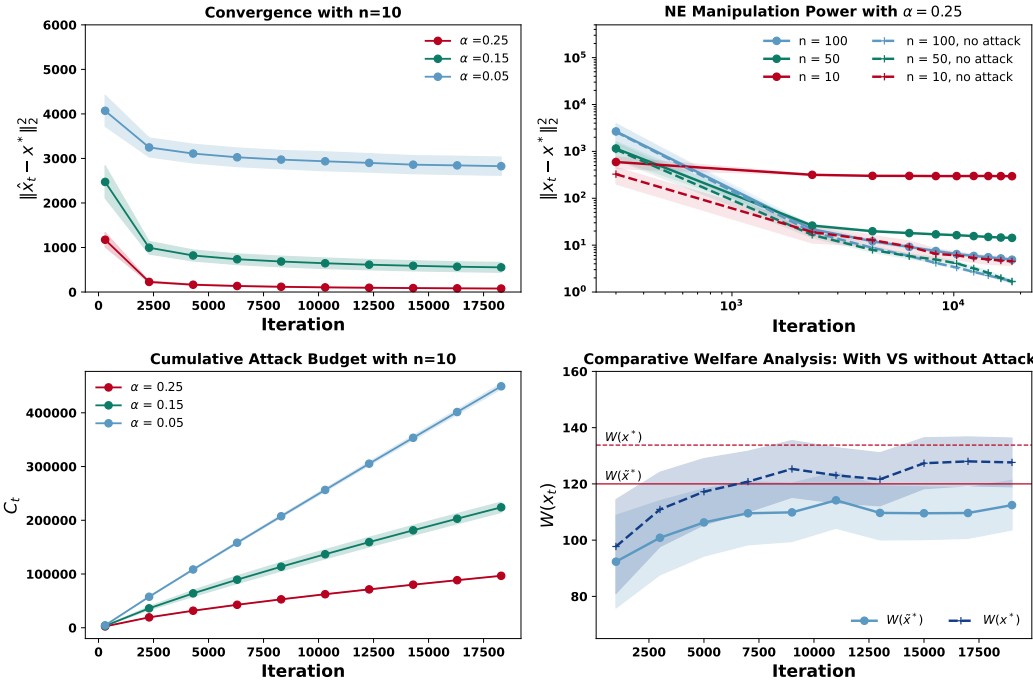

Figure 6: This figure displays results for _Homogeneous_ $n$-person Cournot Game. Top left: square error of MD-SCB with varying convergence rates. Top right: square error of MD-SCB on different sizes of game instances against SUSA. Bottom left: cumulative attack budget used by SUSA against MD-SCB with varying convergence rates. Bottom right: social welfare $W(\boldsymbol{x}_t)$ under SUSA and without attack for $n = 10$. The red dashed line denote the social welfare under NE without attack, $W(x^*)$, while the red solid line denote the social welfare under manipulated NE under SUSA, $W(\tilde{x}^*)$. We plot 0.1-$\sigma$ region from 20 independent simulations for social welfare (the bottom right figure) and 0.5-$\sigma$ region for the rest figures.

**Result:** The top left panel of Figure 6 plots the convergence curve of the $L_2$ square error for $n = 10$, serving as a sanity check to confirm that different learning rate schedules induce different convergence rates for MD-SCB in absence of an adversary. The top right panel shows the outcome of SUSA against MD-SCB with $\alpha = 0.25$, with the $y$-axis representing $L_2$ distance between $\boldsymbol{x}_t$ and NE. The dashed lines display the convergence curve without the adversary, while the solid lines represent the dynamics under SUSA, with different colors indicating results for various game sizes $n$. As shown, for different values of $n$, the attacked dynamics exhibit divergence, as the solid lines saturate and stop decreasing, indicating that the dynamics are being steered to converge to a new point. Additionally, we observe that the induced NE deviation decreases with respect to $n$, which aligns with our theoretical predictions in Theorem 1 and Remark 1. The bottom left panel shows the cumulative attack

budget $C_t = \sum_{\tau=1}^{t} |c_\tau|$ at each time step $t$ against MD-SCB with different values of $\alpha$. As the results indicate, although all exhibit a sublinear trend, a faster-converging dynamics ($\alpha = 0.25$) requires a smaller attack budget, corroborating our findings in Theorem 1 and 2. The bottom right panel plots the social welfare $W(\boldsymbol{x}_t)$ without attack (indicated by the dashed line) and under SUSA (indicated by the solid line). The panel shows that SUSA with $\delta = -10$ can lead the learning dynamic to a new NE with a lower social welfare, aligning with our theoretical results. These observations also hold for Cournot games with *heterogeneous* costs, and the corresponding results are presented in Appendix E.2.

**Heterogeneous $n$-person Cournot Game Experimental Setup:** Similar to the *homogeneous $n$-person* Cournot game, we set $a = 10, b = 0.05, n \in [10, 50, 100], \alpha = [0.25, 0.15, 0.05]$, and $\mathcal{X}_i \in [0, 50], \forall i \in [n]$. However, in the heterogeneous setting, we consider different configurations of $c_i$ based on the value of $n$. For $n = 10$, the costs are assigned as follows: $c_1 = 0.5$, $c_i = 0.6$ for $i \in [2, 7]$, and $c_i = 0.7$ for $i \in [8, 10]$. For larger values of $n$, specifically $n = 50$ and $n = 100$, we replicate the structure of the $n = 10$ case, dividing the sequence into 5 groups when $n = 50$ and into 10 groups when $n = 100$. Each group follows the same pattern of costs assignments as in the $n = 10$ case. We implement SUSA against agent 1 with $\delta = -15$.

**Result:** Figure 7 suggests that all results hold in the heterogeneous setting just as they do in the homogeneous setting.

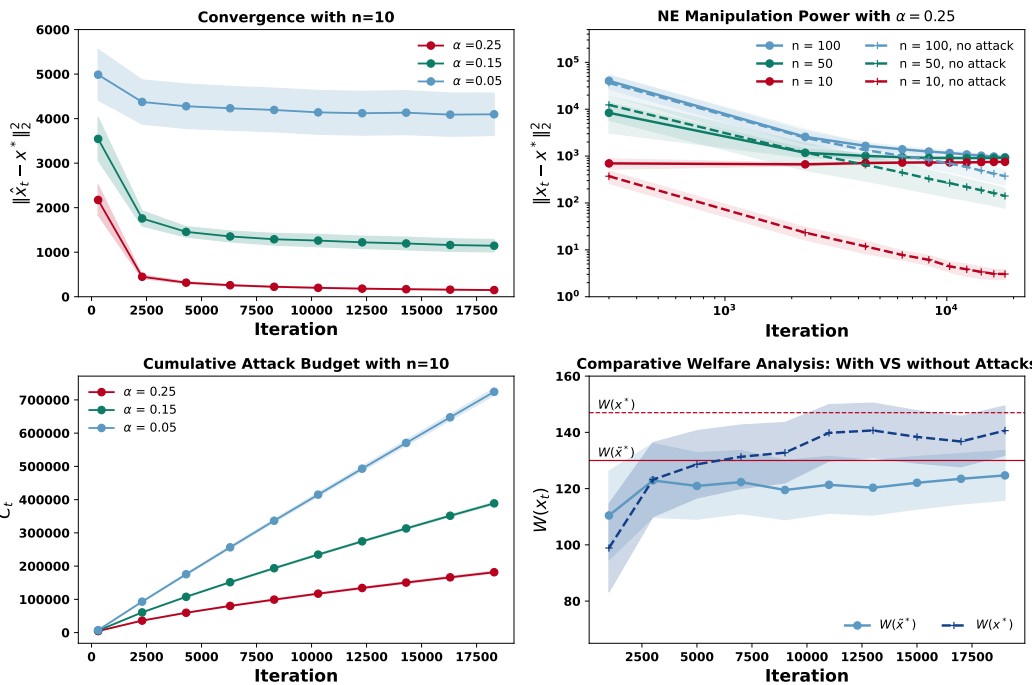

Figure 7: This figure displays results for *Heterogeneous $n$-person Cournot Game*.

