# OpenReview forum: "Single-agent Poisoning Attacks Suffice to Ruin Multi-Agent Learning"
_ICLR.cc/2025/Conference — ICLR 2025 Poster_

### Official Review · Reviewer_oyqK · 2024-10-18

**Soundness:** 4
**Presentation:** 4
**Contribution:** 3
**Rating:** 6
**Confidence:** 4

**Summary:**

This paper considers the poisoning attacks in multi-agent learning, targeting a single agent. The authors first propose the attack strategy which steers the game from the NE with sublinear attack cost. They then explore the robustness of learning algorithms, analyzing how the convergence rate can affect the algorithms' robustness. Finally, the experiments verify the theories by showing the derivative and the cost under different parameters.

**Strengths:**

This paper considers an interesting topic: Attacking a single agent in multi-agent learning. The two main contributions are both noteworthy:

* The first main result indicates that to shift the global equilibrium, an attacker only needs to target one agent with sublinear cost. This practical finding provides valuable insights into the potential risks of multi-agent learning.

* Second, the trade-off between efficiency and robustness is very interesting. I believe this will provide some heuristic for the design of robust algorithms.

Moreover, this paper is well-written.

**Weaknesses:**

I have several concerns about this paper:

1. The assumptions underlying the proposed attack are quite strong. (1) As the author admits,the adversary is assumed to have complete knowledge of the victim agent's utility function. This assumption, while useful for theoretical analysis, may not hold in practical scenarios, potentially limiting the contribution. (2) The agent is assumed to be unaware of the attacker's presence. If the agents do know, will the attack still work?

2. There’s no knowledge regarding the difficulty of the problem. In other words, can the authors show any lower bound on the cost, which will further indicate the efficiency of the attack? It is worth noting that in a similar topic which is also mentioned by this paper, $[1]$ has already closed such a gap. Furthermore, Figure 2 illustrates the significance of this concern, as in the last sub-figure, the cost increases at an almost linear rate. What’s more, this $\alpha$ is determined by the dynamic itself, therefore it cannot be well controlled. Thus, the cumulative cost is quite large to some degree. After all, both $\log(\log(T))$ and $T^{1-\alpha}$ are sublinear, but their outcomes are totally different.

3. The authors may have partially overstated their work. The problem they considered is a specific attack problem in monotone games, however, the title and introduction demonstrate greater ambition (in general games). Whether the results can be generalized remains unknown.

[1] Zuo, S. (2024, April). Near Optimal Adversarial Attacks on Stochastic Bandits and Defenses with Smoothed Responses. In International Conference on Artificial Intelligence and Statistics (pp. 2098-2106). PMLR.

**Questions:**

Please try to solve the problems in weaknesses. Besides, there are some extra questions:

* The trade-off between robustness and efficiency is very intriguing. I suggest authors further highlight this part in the next version.
* In Line 169-170, is the parameter $L$ known to the adversary? How can the adversary select the target agent “smartly”?
* In Line 295-301, the authors want to show that more information will lead to the ability to control the attack direction. However, without more knowledge (which means more assumptions), this is not addressed. Similarly, in Line 319-322, the same issue arises for the derivative distance.
* In Proposition 1, what will happen if the target agents are a subset of all agents? I.e., for example, if the attacker can manipulate the utilities of two agents, is there any related result?
* Minor typo: There is a missing "." in Line 390.

---

> ### Author Response · Authors · 2024-11-18
> **Response to Reviewer oyqK**
>
> We thank the reviewer for appreciating the representation and insights of our work. Below we address the raised concerns and questions in order.
>
> **W1: Assumption about attacker’s knowledge**
>
> For the first question, please refer to the second bullet point of our common response. For the second question, our Theorem 3 in Section 5 provides an answer. Specifically, if agents can estimate an upper bound $T^{\rho}$ of the total budget, they can adjust their learning rate accordingly to achieve a certain level of robustness. This insight is visually represented in the green region of Figure 1. To see this, observe that within the green region, drawing a horizontal line for any fixed $\rho$ results in a corresponding range of $\alpha$ on the $x$-axis. This range of $\alpha$ identifies the learning rates that ensure resilience against NSA.
>
> **W2: Lower bound on corruption budget**
>
> We kindly remind the reviewer that we do have comprehensive algorithm-dependent lower bounds in Section 5 (specifically, Corollary 2 and Proposition 4), similar to Theorem 3,4,5 in [Zuo, 2024]. We chose not to explicitly emphasize these aspects, as they are inherently embedded within our robustness-efficiency tradeoff analysis, which we believe offers deeper and more exciting insights. For detailed explanations, please refer to the last bullet points of our common response. That said, we sincerely appreciate the reviewer for bringing this relevant work to our attention, and we will add a discussion of [Zuo, 2024] in the revised version.
>
> Regarding the concern that our upper bound might be too large, we note that our polynomial-order lower bounds are reasonably tight. For example, in the case of MD-SCB, the lower bound is $\Omega(T^{1-\alpha})$, while the upper bound is $\Omega(T^{1-\frac{2\alpha}{3}})$. Therefore, such a seemingly large required budget is not a result of loose analysis but rather reflects the intrinsic difficulty of the attacking goal.
>
> This larger budget requirement in multi-agent settings starkly contrasts with [Zuo, 2024]’s single-agent learning setting, where the required budget has a matched bound of $O(\log T)$. We find this distinction particularly intriguing, as it may suggest a fundamental gap between single-agent and multi-agent learning environments, an area we believe is worthy of further exploration.
>
> **W3: Generalization to other game-theoretic context** Please refer to the third bullet point of our common response.
>
> **Q1/Q5:**  We will address your comments in revision.
>
> **Q2: Choosing victim agent**
>
> In many critical security applications, the adversary likely has limited options because being able to compromise some agent is likely the most difficult step in practice (which is why we believe our result is particularly valuable by showing it actually suffices to poison a single agent, if done properly). In such situations, the adversary will likely target the agent that it knows the best or a certain insider who has already been infiltrated and compromised.
>
> In other scenarios, where the adversary has greater freedom to choose the victim, it can strategically select agents to achieve specific goals. For instance, the adversary may choose the agent with the greatest potential to induce a desirable equilibrium deviation direction, as demonstrated in Proposition 1. Alternatively, it may target the agent whose compromise results in the most significant welfare reduction, as highlighted in Theorem 2 of the new draft (Section 4.3).
>
> **Q3: More knowledge requirement for manipulating attack direction**
>
> Your understanding is correct. First, as we clarified in the second bullet point of our common response, in many applications, the adversary does have access to additional knowledge, enabling more detailed control over the attack outcome.
>
> Second, we acknowledge that in cases where the adversary lacks additional knowledge about the environment, achieving refined control over the outcome is not guaranteed. However, this does not render the discussion meaningless. From a system or defender’s perspective, our discussion highlights potential threats to any MAL system by illustrating what an adversary could possibly achieve. We will add further discussion on this point in the revised version.
>
> **Q4: Target at a subset of agents**
>
> If the attacker manipulates $k$ agents, theoretically it can induce an NE shifting direction that lies in a $kd$-dimensional hyperplane, a sub-space of the $nd$-dimensional joint strategy space. This is why the ability to manipulate all agents’ utilities enables the attacker to achieve arbitrary NE deviation directions. However, characterizing this $kd$-dimensional hyperplane is highly nuanced, as it depends on a linear transformation induced by the inverse of the game’s Hessian (an $nd$-by-$nd$ matrix). Given the complexity and limited relevance of this characterization to our main contributions, we decided not to include a detailed discussion in the paper.

---

> > ### Comment · Reviewer_oyqK · 2024-11-21
> >
> > Thanks for your rebuttal. After reading the comments from other reviewers, I decide to increase my rating.
> >
> > I still believe that a more detailed consideration of the lower bound and the trade-off in the paper will reinforce this paper. I hope the authors to address these aspects in greater depth in the next version.

---

> ### Author Response · Authors · 2024-11-22
>
> We appreciate the reviewer's understanding and helpful thoughts and we will definitely integrate these comments into our next version.
>
> Regarding the trade-off and lower bound, we would like to clarify that we believe that these aspects have been emphasized in our section 5. Specifically:
>
> **Figure 1:** This figure summarizes and contrasts the upper and lower bound results, highlighting the gaps as open questions for further exploration.
>
> **Lower Bound:** In Section 5, we highlight the corresponding results in red text for better visibility. For instance, in Equation (17), $\rho_+(\alpha)$ denotes the algorithm-independent lower bound, while $\rho(MD-SCB(\alpha))$ corresponds to an algorithm-dependent lower bound. These elements and their interpretations are clearly presented in the following discussion below Corollary 2.
>
> That said, we would greatly appreciate it if the reviewer could provide more details on what considerations they feel are currently missing and suggest specific in-depth results or analyses regarding the trade-off that could further strengthen our paper. Thanks!

---

### Official Review · Reviewer_3kAq · 2024-10-30

**Soundness:** 3
**Presentation:** 3
**Contribution:** 3
**Rating:** 6
**Confidence:** 3

**Summary:**

This paper investigates the vulnerability of multi-agent learning (MAL) systems in strongly monotone games with bandit feedback. Specifically, the authors propose an attack strategy called Single-agent Utility Shifting Attack (SUSA), which can steer the learning dynamics away from the Nash equilibrium with a sublinear (w.r.t. T) budget. The authors also provide theoretical and empirical results to validate their points.

**Strengths:**

1. The paper introduces a novel type of poisoning attack that focuses on a single agent in multi-agent systems, a context underexplored in prior work. The proposed poisoning strategy can mislead any MAL dynamics in strongly monotone games away from the original NE, with a sublinear corruption budget.

2. The authors show a trade-off between convergence speed and the robustness to attacks.

3. The authors provide a thorough theoretical analysis to validate theoretical results.

**Weaknesses:**

In the attack model, the authors assume full knowledge of the victim agent’s utility function, which may not always be practical in real-world applications. Moreover, since the agent is unaware of their utilities and the adversary has such knowledge which further allows to compute more information such as gradient, misleading the system not to converge the original NE looks not surprising due to the information asymmetry. It could be more interesting to restrict the adversary. For example, the adversary only observes the noisy rewards.

**Questions:**

Can the authors discuss the attack strategy if the adversary only observes noisy rewards, e.g.,  sub-gaussian rewards?

Is there any lower bound on the budget that misleads the convergence?

---

> ### Author Response · Authors · 2024-11-18
> **Response to Reviewer 3kAq**
>
> We are happy to learn that the reviewer likes our results and insights. To address your concern regarding our assumption of the attacker's knowledge, please refer to the second bullet point of our common response.
>
> We address the reviewer’s two other questions below.
>
> **Q1: Noisy Rewards**
>
> Our attack can afford noisy $c_t$, i.e., the amount of utility corruption can only be estimated up to a Gausian noise (see the second Bullet point of our common response). However, we do not have a theoretical guarantee regarding what an adversary can achieve if it does not observe any agent’s utility but only has access to their noisy reward observations. We do appreciate the reviewer for bringing up this interesting direction that is worth future investigation. Additionally, we note that if the adversary possesses partial knowledge of the victim agent’s utility function $u_k$​, even noisy reward observations could enable it to refine its estimation of $u_k$ by probing the MAL system. Specifically, the adversary could implement trial attacks to gather information and subsequently solve an inverse problem to execute a successful attack. For a more detailed discussion, please refer to Bullet Point 2 in our common response.
>
> **Q2: Lower bound on corruption budget**
>
> We kindly point out that we did provide a set of lower bound results in Section 5 (specifically, Corollary 2 and Proposition 4). We chose not to explicitly emphasize these aspects, as they are inherently embedded within our robustness-efficiency tradeoff analysis, which we believe offers deeper and more exciting insights. Please refer to the last bullet point of our common response for more detailed discussion.

---

> > ### Comment · Reviewer_3kAq · 2024-11-27
> >
> > Thanks for your reply. I have no more questions. Overall, I believe it is a good paper.

---

> ### Author Response · Authors · 2024-11-27
> **Response to Reviewer 3kAq**
>
> We sincerely appreciate the reviewer's very positive feedback, and we are happy to address any further questions if the reviewer might have. If the reviewer thinks that we have fully addressed the concerns, we respectfully request the reviewer to reevaluate the rating of our paper (as our understanding is that a score of 6 means the paper is only “marginally” above the bar).
>
> In addition, we would like to further emphasize our contribution regarding the attacking scheme, SUSA. We respectfully disagree with the reviewer’s assessment that our result for SUSA is unsurprising in the full-information setting (actually we should call it “partial-information setting” since it only assumes that the attacker knows his attacked agent’s utility instead of the entire game) for the following two reasons.
>
> Firstly, we believe our assumption is “moderate” in the adversarial attack or “equilibrium steering” literature [1]. For example, [2] considers a game redesign problem by assuming the designer knows the **entire game** (equivalent to knowing every agents’ utility). [3,4] assume the attacker can completely control one of the learners in multi-agent learning.
>
> Secondly, while it is indeed expected that agents can be misled to converge to a new NE, our contribution lies in demonstrating that this can be accomplished with merely a **sublinear** budget on attacking previously already convergent algorithm, which is stronger than attacking **merely no regret learning algorithms** studied in other relevant literatures [5]. Therefore, we believe that the design of such an attacking strategy to achieve this outcome is non-trivial.
>
> However, we agree that it will be an intriguing open direction to further relax the attacker’s knowledge and design optimal attack strategies for the black-box setting (the attacker has no information of the game and has to learn it from past trajectories). **We mention it in the Limitation section and leave it as our future work. Please refer to the updated draft**.
>
> [1] A survey on the analysis and control of evolutionary matrix games
>
> [2] Game Redesign in No-regret Game Playing
>
> [3] Adversarial policies: Attacking deep reinforcement learning
>
> [4] Adversarial policy learning in two-player competitive games.
>
> [5] Steering No-Regret Learners to a Desired Equilibrium

---

### Official Review · Reviewer_vqqA · 2024-11-03

**Soundness:** 3
**Presentation:** 4
**Contribution:** 3
**Rating:** 8
**Confidence:** 3

**Summary:**

This paper designs the attack policy for multi-agent learning in monotone games. It shows that attacking a single agent is enough to diverge the convergence away from NE. The paper also studies the robustness of the MAL algorithm, presenting several interesting open problems and numerical simulations.

**Strengths:**

1. This paper's writing is clear and easy to follow. The figures (especially Figure 1) and remarks help readers understand the paper.
2. The contributions of the attack policy to a single agent and proving a sublinear attack is theoretically enough (Theorem 1) are novel in the literature.
3. The discussion of the robustness and efficiency trade-off is very insightful and opens the door for more interesting future works.

**Weaknesses:**

1. As the authors mentioned that the trade-off had been studied in a single agent, it would be helpful to discuss whether or not the three raised open problems are also present in the single-agent setting. If so, can we extend the single-agent results? If not, why?

**Questions:**

Minor comments:

1. Line 147, “will useful” → “will be useful”.
2. Line 242 is a “strong” attack, [Lykouris et al., 2018] is a “medium” attack before observing the action. For strong attack, the related literature has:
    1. Jun, Kwang-Sung, et al. "Adversarial attacks on stochastic bandits." *Advances in neural information processing systems* 31 (2018).
    2. Liu, Fang, and Ness Shroff. "Data poisoning attacks on stochastic bandits." *International Conference on Machine Learning*. PMLR, 2019

---

> ### Author Response · Authors · 2024-11-18
> **Response to Reviewer vqqA**
>
> We appreciate the reviewer’s positive feedback on our presentation and contributions. Regarding the minor comments, we find them very helpful and will surely address the typo and include relevant literature on strong attacks in our next revision. Below, we respond to the question raised under Weaknesses.
>
> **Trade-offs in single agent setting**
>
> The problem of closing the gap between budget lower and upper bounds has been addressed in some single-agent online learning settings, such as Stochastic Linear Bandits [Bogunovic et al., 2021]. However, to the best of our knowledge, previous work on reward poisoning against bandits does not study the efficiency-robustness trade-offs, as they mainly focus on the vulnerability of the most efficient learning algorithms with the best regret guarantees. [Cheng et al., 2024] discussed efficiency-robustness trade-offs in the context of reward poisoning in dueling bandits, as mentioned in Section 5 of our paper.
> Nonetheless, these results cannot be directly applied to multi-agent learning scenarios, where each agent’s utility observations are influenced by the actions of all agents. This interdependence makes the effects of adversarially "hacking" the utility of a single agent—whose observations may impact others—both complex and unpredictable. The fundamental barriers to adapting algorithmic designs from single-agent to multi-agent settings have also been discussed in prior works [Wu et al., 2021, Wang et al., 2022], implying the challenges inherent in studying the robustness of MAL algorithms, which is the focus of our work.
>
> [1] Bogunovic, Ilija, et al. "Stochastic linear bandits robust to adversarial attacks." International Conference on Artificial Intelligence and Statistics. PMLR, 2021.
>
> [2] Cheng, Yuwei, et al. "Learning from Imperfect Human Feedback: a Tale from Corruption-Robust Dueling." arXiv preprint arXiv:2405.11204 (2024).
>
> [3] Wu, Jibang, Haifeng Xu, and Fan Yao. "Multi-Agent Learning for Iterative Dominance Elimination: Formal Barriers and New Algorithms." COLT. 2022.
>
> [4] Wang, Yuanhao, et al. "Learning Rationalizable Equilibria in Multiplayer Games." arXiv preprint arXiv:2210.11402 (2022).

---

> > ### Comment · Reviewer_vqqA · 2024-11-18
> >
> > The reviewer thanks the authors for the follow-up discussion and will maintain their assessment.

---

### Official Review · Reviewer_mptu · 2024-11-06

**Soundness:** 3
**Presentation:** 3
**Contribution:** 3
**Rating:** 8
**Confidence:** 3

**Summary:**

This paper studies the robustness of multi-agent learning (MAL) algorithms in strongly monotone games with bandit feedback. The authors propose the Single-agent Utility Shifting Attack (SUSA) method to shift the Nash Equilibrium of monotone games by corrupting a single agent's utility feedback, using a sublinear corruption budget relative to the time horizon. Their analysis uncovers a trade-off between efficiency and robustness, showing that faster-converging MAL algorithms are more vulnerable to such attacks. They also validate their theoretical findings via numerical experiments.

**Strengths:**

1) The paper highlights a significant and underexplored vulnerability in multi-agent learning (MAL), specifically through single-agent utility poisoning attacks. It may stimulate further research into designing MAL algorithms that are robust to adversarial attacks.

2) The authors provide a rigorous theoretical analysis of the Single-agent Utility Shifting Attack (SUSA), clearly outlining the conditions under which SUSA can effectively alter the Nash Equilibrium (NE). The exploration of the efficiency-robustness trade-off is valuable, highlighting the increased vulnerability of faster-converging algorithms.

3) The authors conduct extensive empirical simulations, showcasing the practical impact and effectiveness of their proposed method

**Weaknesses:**

1) The current attack objective is focused on shifting the Nash Equilibrium (NE) with specific distance guarantees. While the paper briefly discusses steering the NE deviation in a desired direction (lines 295-300), it remains unclear if it is feasible to mislead agents toward specific, predefined strategies.
2) The study is primarily focused on monotone games, illustrated through Cournot competition and Tullock contests. It would be valuable to examine whether these insights hold in other game-theoretic contexts. For instance, in non-monotone games with multiple NEs, it would be interesting to explore alternative attack objectives, such as guiding agents toward an NE with low utility outcomes.
3) The paper evaluates the effectiveness of attacks based on NE shift and cumulative budget. Expanding the evaluation to include additional robustness metrics, such as stability and utility outcomes, would provide a more comprehensive understanding of the impact of attacks on MAL.

**Questions:**

1) Given that the attack model assumes full knowledge of the victim agent’s utility function, do the authors believe that SUSA could still be effective in a limited-information setting? Are there alternative attack strategies that might be feasible with only partial information?

2) The attack model relies on "strong" corruption, where the attacker observes the current round action. It would be valuable to investigate whether the results extend to scenarios where the attacker lacks this observational ability, as well as whether it becomes easier to design robust algorithms against such "weaker" attackers.

3) To what extent might the findings on NE shifting and efficiency-robustness trade-offs apply to non-monotone games or games with multiple NEs? Could the authors envision scenarios where the attack objective is to guide agents toward a low-utility NE?

---

> ### Author Response · Authors · 2024-11-18
> **Response to Reviewer mptu**
>
> We thank the reviewer for appreciating the value of our work. We respond to the reviewer’s criticisms mentioned in weaknesses and answer the raised questions in the following.
>
> **W1/W3: Mislead agents towards specific strategies / bad outcomes, and implications in stability and utility outcome**
>
> Please refer to the first bullet point of our common response and our updated draft.
>
> **W2: Extend to other game-theoretic context**
>
> Please refer to the third bullet point of our common response.
>
> **Q1: Attack under limited information**
>
> Please refer to the second bullet point of our common response.
>
> **Q2: Relax the strong attack assumption**
>
> First, we would like to clarify that our focus is on strong attack settings because, in many application scenarios, adversaries can indeed observe all players’ actions. For example, the adversary can be an incentive-misaligned platform designer, e.g., an online content platform might prioritize revenue at the expense of content providers' overall welfare. In this case, the adversary inherently has full knowledge of all agents' actions, and thus can easily apply SUSA to one agent in order to steer the entire MAL system (this actually happens in reality, known as platform subsidy to few selected agents).
> While we have justified our focus on the current setting, we do find the reviewer’s suggestion to investigate weak attacks very insightful. We agree that studying the implementation of and defense mechanisms against weak attacks are both practical and valuable, and we plan to explore this direction in future work.
>
> **Q3: Result for non-monotone games and towards a low-utility NE**
>
> For the extension to non-monotone games, please refer to the last bullet point of our common response. For the attack towards a low-utility NE, please refer to the first bullet point of our common response and our updated draft.

---

### Author Response · Authors · 2024-11-18
**Common Response-1**

We appreciate the reviewers' overall positive evaluations about our work, especially the acknowledgement of the significance of the problem we study and the new insight from our results. We are happy to integrate the reviewers’ suggestions for improving the current version.

In the following, we first respond to the common questions raised by reviewers, and then answer each reviewer’s questions separately.

1. **Additional results about attack guarantees beyond just shifting NE**

Some reviewers expressed interest in additional characterizations of the outcome of our attack beyond simply shifting the NE, such as its impact on social welfare or the utilities of other agents. **We have thoroughly addressed this problem** in our revised draft and included new theoretical results in Section 4.3 (theoretical guarantees with proofs provided in Appendix B.3) and additional experiments to support our new results in Appendix F (in the right bottom panel of Figure 6, 7), with updates marked in blue text.
The new draft with section 4.3 exceeds the page limit. However, should this work be accepted, we plan to address this by merging Sections 4.1 and 4.3 into a single subsection and combining Theorem 1 and Theorem 2. These adjustments will ensure the final camera-ready version complies within the page limit.

In summary, we prove in Theorem 2 and Corollary 1 that for almost any function $W$ defined w.r.t. agents’ joint strategies, the adversary can implement a SUSA that strictly decreases the value of this $W$ at the corrupted NE compared to its value at the original NE. This result is particularly compelling as $W$ can represent a wide range of metrics. For instance, $W$ could be a reasonable social welfare metric, such as the sum of all agents’ utilities, the sum of a targeted subset of agents’ utilities, or even a metric unrelated to any specific agent (e.g., an environmental or systemic metric). Surprisingly, the adversary can achieve this attack goal by targeting just a single agent while incurring an imperceptible total cost. This demonstrates a more significant implication than merely shifting the equilibrium, highlighting the broader vulnerabilities of such systems.

2. **Full knowledge of the victim agent’s utility**

Reviewers raised a common concern that the information required by the adversary to implement SUSA appears too strong. We argue that this assumption is not as stringent as it may first appear, for the following reasons:

* First, our attack success only requires the adversary to know the utility function of any single agent, not every agent's. In practice, it is plausible for an adversary to target an agent that it knows the best or one that can be easily infiltrated or probed (e.g., an insider).
* To implement SUSA, the corruption $c_t$ does not need to be calculated with perfect precision. We can show that even if $c_t$​ is computed with zero-mean sub-Gaussian noise, the same success and budget guarantees still hold. This relaxed requirement broadens the applicability of the attack, as it allows the adversary to probe the victim agent’s utility over time within a tolerable margin of error. We will include this discussion in the revision.
* Finally, in some application scenarios (e.g., e-commerce platforms), the “adversary” may not be an external entity but rather an incentive-*misaligned* platform designer. For instance, an online content platform might prioritize revenue at the expense of content providers' overall welfare. In these scenarios, the adversary inherently has full knowledge of all agents’ utility functions since these payoffs are designed and assigned by the adversary itself. Hence the adversary can easily apply SUSA to one agent in order to steer the entire MAL system (this actually happens in reality, known as platform subsidy to few selected agents).

Moreover, as this is the first work to explore the vulnerability of MAL systems, our primary objective is to reveal the potential risks to MAL algorithms. The development of more nuanced attack strategies under various partial information structures is an interesting and important open problem, which we intend to explore in future work. Nonetheless, our work reveals the feasibility of attacks under partial knowledge, uncovering the vulnerability of MAL.

---

> ### Author Response · Authors · 2024-11-18
> **Common Response-2**
>
> (Continued from Common Response-1)
> --- ---
> We also provide a proposal for relaxing the full-knowledge utility assumption for future studies. Here we outline a method for learning the victim agent's utility function using a preliminary probing phase. Suppose the victim’s utility $u_k(\boldsymbol{x};\theta)$ is parameterized by $\theta$, where the adversary knows the function form of $u_k$ but not the parameter $\theta$. During the probing phase, the adversary can design and apply a sequence of corruptions $c_t$​, fixing each $c_t$​ for a sufficient duration to observe the induced corrupted equilibrium. By doing so, the adversary can estimate the victim’s best-response function at various points, gradually refining its knowledge of the utility parameter $\theta$. To ensure the success of the attack while maintaining a sublinear corruption budget, the adversary must carefully balance the length of the probing phase. A longer probing phase improves the accuracy of the utility estimation but incurs higher overall costs, as probing involves linear budget growth. While a detailed discussion of this process is beyond the scope of this work, we intend to address it in future research.
>
> 3. **Extensions to more general game classes**
>
> Some reviewers pointed out that our results are established within the context of monotone games and inquired whether they can be extended to other settings. **The answer is Yes**; however, before discussing the generalizability of our results, it is important to emphasize that our attack is not targeted at a specific class of games $\mathcal{G}$, but rather at a class of MAL algorithms $\mathcal{A}$ running on an environment defined by a class of games $\mathcal{G}$.
>
> In fact, the fundamental requirements for $\mathcal{G}$ and $\mathcal{A}$ in Theorem 1 to establish the same attack success guarantees are quite broad:
>
> - (1) condition for $\mathcal{G}$: Any game such that: 1. each agent has a compact, continuous action set, and a well-defined best-response mapping; 2. it has a unique PNE, and if one player’s utility $u_k(x_k,x_{-k})$ changes to $u_k(x_k+\delta,x_{-k})$, the new game also has a unique PNE.
> - (2) condition for $\mathcal{A}$: MAL algorithms that convergence to the PNE for any $G\in\mathcal{G}$.
>
> In our current draft, we chose to present our results in the context of monotone games and refrained from stating Theorem 1 in this more abstract and general form for a practical reason: to the best of our knowledge, *the only well-established class of MAL algorithms with rigorous guarantees of convergence to a unique PNE currently exists is within the monotone game context*. That said, should future work develop classes of new MAL algorithms for other types of games that satisfy the above requirements (1) and (2), our SUSA attack would remain a valid threat to such systems. If this more general description clarifies the broader applicability of our results, we are happy to revise the draft to incorporate this general setting.
>
> 4. **Lower bounds on corruption budget**
>
> We appreciate the reviewers’ inquiry regarding lower bound results. In fact, our paper includes a comprehensive discussion of algorithm-dependent lower bounds, as explained below. They are implicit within our robustness-efficiency tradeoff analysis in Section 5, which we are more excited about since we believe robustness-efficiency tradeoff conveys deeper and more novel insights. Meanwhile, we also acknowledge that: 1. We do not have algorithm-independent lower bounds, 2. There are still gaps between our lower and upper bounds. As discussed in Section 5, addressing these gaps presents novel and intriguing challenges due to the inherent complexity of multi-agent learning in game-theoretic environments. We have outlined these challenges explicitly as Open Questions 1, 2, and 3 (OP 1,2,3) in Section 5, and we plan to explore them in future work.
>
> - **Algorithm-dependent budget lower bounds for achieving NE Shifting Attack (NSA)** We provide budget lower bounds for achieving NSA against MD-SCB and MAMD algorithms.
>
>   + a. **MD-SCB**. Corollary 2 presents the lower bound for MD-SCB. The LHS inequality $\rho(MD-SCB(\alpha))\geq 1-\alpha$ in Eq. (14) demonstrates that a total budget of $\Omega(T^{1-\alpha})$ is necessary to achieve NSA for MD-SCB with a convergence rate $(\alpha, 2)$. This result complements the upper bound in the right-side inequality of Eq. (14) — a restatement of Theorem 1— indicating that SUSA with a budget of $O(T^{1-\frac{2\alpha}{3}})$ suffices to achieve NSA for any MAL algorithm (see the green solid line in Figure 1).
>
>   + b. **MAMD**. A similar lower bound of $\Omega(T^{2/3})$ for MAMD is derived from Proposition 4 (Appendix C.4). This result is further discussed in the final paragraph of Appendix D.1. While this bound aligns with our theoretical insights from Theorem 3 and Corollary 2, it is relegated to the appendix due to its relatively limited standalone value in the main narrative.

---

> > ### Author Response · Authors · 2024-11-18
> > **Common Response-3**
> >
> > (Continued from Common Response-2)
> > --- ---
> >
> > - **Algorithm-dependent budget lower bounds for achieving $\gamma$-SA** We also addressed a weaker attack goal of slowing down the convergence rate of the target MAL algorithm (without necessarily misleading convergence). This discussion is detailed in Appendix D due to space constraints. The corresponding budget lower bounds are:
> >
> >   + a. **MD-SCB**. $\Omega(T^{1-3\alpha})$ (see Corollary 3).
> >   + b. **MAMD**. $\Omega(T^{1-4\alpha})$ (see the discussion in the last paragraph of Appendix D.1).

---

### Meta-Review · Area_Chair_qkrz · 2024-12-19

**Metareview:**

The paper studies how easy it would be to disrupt learning dynamics in strongly-monotone games under bandit feedback.

Overall, the reviewers agreed that this paper is a significant contribution to the field, providing theoretical and practical insights into the vulnerabilities of MAL systems. In the words of Reviewer oyqK, “The two main contributions are noteworthy [..] This practical finding provides valuable insights into the potential risks of multi-agent learning.”

Reviewer mptu highlighted the value of the work in exposing vulnerabilities in MAL: “The paper highlights a significant and underexplored vulnerability in multi-agent learning [..] showing that faster-converging algorithms are more vulnerable to such attacks.” This insight was echoed by Reviewer oyqK: “to shift the global equilibrium, an attacker only needs to target one agent with sublinear cost,” a practical finding with implications for understanding MAL system resilience.

All the reviewers agreed that the paper deserves presentation at ICLR.

**Additional Comments On Reviewer Discussion:**

In the first round of responses, the authors addressed several concerns and expanded on key points. For instance, in response to concerns about the strong knowledge assumptions of the attacker, the authors clarified that their framework is applicable to scenarios where the adversary has partial or noisy observations. Reviewer oyqK appreciated the response and decided to increase their score.

---

### Decision · Program_Chairs · 2025-01-22

Accept (Poster)